# Conditional Distributional Treatment Effects: Doubly Robust Estimation and Testing

**Saksham Jain**[1]  **Alex Luedtke**[2]

## Abstract

Beyond conditional average treatment effects, treatments may impact the entire outcome distribution in covariate-dependent ways, for example, by altering the variance or tail risks for specific subpopulations. We propose a novel estimand to capture such conditional distributional treatment effects, and develop a doubly robust estimator that is minimax optimal in the local asymptotic sense. Using this, we develop a test for the global homogeneity of conditional potential outcome distributions that accommodates discrepancies beyond the maximum mean discrepancy (MMD), has provably valid type 1 error, and is consistent against fixed alternatives—the first test, to our knowledge, with such guarantees in this setting. We then provide a test that aggregates evidence across a grid of kernel-bandwidth choices. Furthermore, we derive exact closed-form expressions for two natural discrepancies (including the MMD), and provide a computationally efficient, permutation-free algorithm for our test.

## 1. Introduction

Causal inference for mean effects is well-studied for both marginal (Rosenbaum & Rubin, 1983; Robins et al., 1994) and conditional (Abrevaya et al., 2015; Wager & Athey, 2018; Künzel et al., 2019) estimands, as is their doubly robust estimation (Van der Laan et al., 2011; Kurz, 2022).

However, treatments may impact the entire outcome distribution, a fact that has spurred interest in distributional treatment effects (DTEs) (Bitler et al., 2006; Chernozhukov et al., 2013; Muandet et al., 2021; Fawkes et al., 2024). Further, these distributional impacts may differ across sub-

[1]Department of Statistics, University of Washington, Seattle, WA, USA [2]Department of Health Care Policy, Harvard Medical School, Boston, MA, USA. Correspondence to: Saksham Jain <sj305@uw.edu>.

*Proceedings of the 43rd International Conference on Machine Learning*, Seoul, South Korea. PMLR 306, 2026. Copyright 2026 by the author(s).

populations, as illustrated in Fig. 1. Understanding how potential outcome distributions $P_{Y(a)\,|\,X}$ differ given covariates is of significant interest (Chang et al., 2015; Hohberg et al., 2020; Chernozhukov et al., 2024).

Kernel methods offer a rigorous framework for analyzing DTEs by embedding distributions into reproducing kernel Hilbert spaces (RKHSs) (Song et al., 2009; Gretton et al., 2012) and comparing these embeddings via measures of statistical discrepancy such as the MMD, which is zero if and only if the distributions are equal, provided a characteristic kernel is used (Sriperumbudur et al., 2011).

While inference for marginal DTEs has advanced significantly (Martinez Taboada et al., 2023; Luedtke & Chung, 2024), it remains underdeveloped in the conditional setting. Park et al. (2021) presented a test based on the conditional distributional treatment effect associated with the MMD (henceforth referred to as the CoDiTE function) defined as

$$\mathrm{CoDiTE}_P(x) = \left\| \mu_{P_{Y(1)\,|\,X}}(x) - \mu_{P_{Y(0)\,|\,X}}(x) \right\|_{\mathcal{H}_{\mathcal{Y}}}, \quad (1)$$

where $\mu_{P_{Y(a)\,|\,X}}(x)$ is the conditional mean embedding (Park & Muandet, 2020) of $P_{Y(a)\,|\,X}(\cdot\,|\,x)$ in an RKHS $\mathcal{H}_{\mathcal{Y}}$. However, their estimator for this function is not doubly robust, and they rely on permutation tests that lack validity guarantees. Moreover, other current approaches are either limited to best linear projections (Kallus & Oprescu, 2023) or study testing of pointwise equivalence (Näf & Susmann, 2024). In this work, we instead focus on globally testing the null of equal conditional potential outcome distributions,

$$H_0 : P_{Y(1)\,|\,X}(\cdot\,|\,x) = P_{Y(0)\,|\,X}(\cdot\,|\,x) \quad P_X\text{-a.e.}, \quad (2)$$

against the complementary alternative. Further discussion of related work is provided in App. A.

**Our Contributions.**

1. We propose, to our knowledge, the first provably valid kernel-based test for the (global) homogeneity of conditional potential outcome distributions based on a doubly robust estimator. We also extend it to a test that aggregates evidence across kernel bandwidths.

2. Our test uses the bootstrap to determine a rejection region. In contrast to permutation tests, it does not

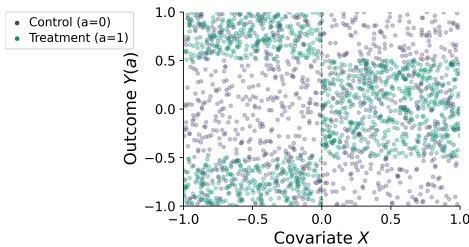 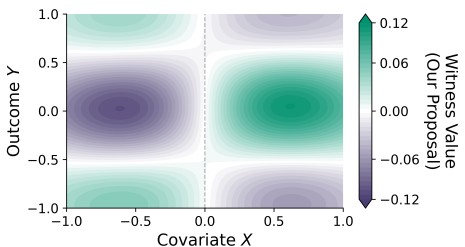

*Figure 1.* Simple setting where the **conditional average treatment effect is null** (left) even though **there is DTE heterogeneity** (right). *In more detail: (left) Scatter plot of $X$ and $Y(a)$, $a \in \{0, 1\}$, with: $X, Y(0) \sim \mathrm{Unif}[-1, 1]$ independently and $Y(1) \mid X$ a $\mathrm{Unif}[-.5, .5]$ distribution if $X > 0$ and a $\mathrm{Unif}([-1, -.5] \cup [.5, 1])$ if $X < 0$. (right) Proposed witness function for conditional DTE.*

refit the nuisances across replicates, thereby amortizing computational costs.

3. We derive exact closed-form expressions for MMD and Wald-type test statistics, enabling the construction of Wald-type confidence sets for conditional DTEs. We also give Nyström approximations for these statistics to reduce the computational cost.

4. We construct asymptotically valid uniform confidence bands to help identify regions of heterogeneous distributional treatment effects.

5. We demonstrate the finite-sample performance of our methods using both simulations and real-world data.

## 2. Preliminaries

### 2.1. Problem Setup

We observe an i.i.d. sample $\mathscr{D} := \{Z_i\}_{i=1}^n$ from a distribution $P$ in a statistical model $\mathcal{P}$ on $\mathcal{Z} := \mathcal{X} \times \{0, 1\} \times \mathcal{Y}$, where $Z_i := (X_i, A_i, Y_i)$ comprises pre-treatment covariates, a treatment assignment, and outcomes. We define the propensity score as $\pi_P(x) := P(A = 1 \mid X = x)$. Along with the standard causal assumptions (consistency, unconfoundedness, overlap) (Stone, 1993; Mealli & Rubin, 2003), we also assume strong positivity: there exists $\eta > 0$ such that for all $P \in \mathcal{P}$, $\eta \leq \pi_P(x) \leq 1 - \eta$ $P_X$-a.e.

We assume $\mathcal{P}$ is dominated and locally nonparametric. The latter means that the tangent space at each $P \in \mathcal{P}$ is the Hilbert space $L_0^2(P) := \{h \in L^2(P) : \mathbb{E}_P[h(Z)] = 0\}$ (van der Vaart, 2000).

We let $k$ and $\ell$ be bounded characteristic kernels on $\mathcal{X}$ and $\mathcal{Y}$ with feature maps $K_x := k(x, \cdot)$ and $L_y := \ell(y, \cdot)$, respectively. We operate in the real, separable tensor product RKHS $\mathcal{H} := \mathcal{H}_\mathcal{X} \otimes \mathcal{H}_\mathcal{Y}$ associated with the kernel $\lambda((x, y), (x', y')) := k(x, x')\ell(y, y')$. As the product of bounded characteristic kernels, $\lambda$ is also bounded and characteristic. App. B gives an extended discussion of the full theoretical setup, including formal statements of the causal assumptions and the definition of the tangent space through quadratic mean differentiability (QMD).

For readability, we suppress the explicit dependence of functionals on $P$ (e.g., writing $\mathrm{CoDiTE}(x)$ instead of $\mathrm{CoDiTE}_P(x)$) when the value of $P$ is clear from context.

### 2.2. Conditional Distributional Treatment Effects

Under the standard causal assumptions, the conditional mean embedding $\mu_{P_{Y(a)} \mid X}(x)$ from (1) is identified from the observed data by the following $\mathcal{H}_\mathcal{Y}$-valued function:

$$\nu_{P,a}(x) := \mathbb{E}_P\left[L_Y \mid A = a, X = x\right]. \quad (3)$$

Let $U_{P|x} := \nu_{P,1}(x) - \nu_{P,0}(x)$. Then, $\mathrm{CoDiTE}(x)$ from (1) can be expressed as

$$\|\nu_{P,1}(x) - \nu_{P,0}(x)\|_{\mathcal{H}_\mathcal{Y}} = \|U_{P|x}\|_{\mathcal{H}_\mathcal{Y}}. \quad (4)$$

Park et al. (2021) use this to develop a test for $H_0$ as in (2) against the complementary alternative. We show an equivalent null can be formulated using *joint* potential outcome and covariate distributions instead of *conditional* distributions. The key argument used to establish this is intuitive: since $X$ precedes treatment, the marginal distribution $P_X$ must be the same on either side of Eq. 2—see App. C.1.

**Proposition 2.1** (Equivalent null). *For any $P \in \mathcal{P}$, $H_0$ holds if and only if $P_{Y(1),X} = P_{Y(0),X}$.*

The conditional mean embedding of the joint potential outcome and covariate distribution $P_{Y(a),X}$ is identified under the standard causal assumptions as the following $\mathcal{H}$-valued function of $x \in \mathcal{X}$ defined for each $a \in \{0, 1\}$:

$$\theta_{P,a}(x) := K_x \otimes \nu_{P,a}(x). \quad (5)$$

This motivates our definition of the 'Smoothed' Conditional Distributional Treatment Effect (SCoDiTE) as the Hilbert-valued parameter $\psi : \mathcal{P} \to \mathcal{H}$ given by

$$\psi(P) := \mathbb{E}_P\left[\theta_{P,1}(X) - \theta_{P,0}(X)\right]. \quad (6)$$

We use the shorthand $\psi_P := \psi(P)$ throughout. Consequently, for $U_{P|x}(y)$ the witness function for $\mathrm{CoDiTE}(x)$ from (4), the SCoDiTE witness function writes as

$\psi_P(x, y) = \int k(x', x) U_{P \mid x'}(y) P_X(dx')$. Thus, $\psi_P(\cdot, y)$ is an $\mathcal{H}_{\mathcal{X}}$-kernel smoothing of $U_{P \mid \cdot}(y)$.

Park et al. (2021) test $H_0$ through a criterion they call the kernel conditional discrepancy:

$$\text{KCD} := \mathbb{E}_P \left[ \text{CoDiTE}^2(X) \right] = \mathbb{E}_P \left[ \| U_{P \mid X} \|_{\mathcal{H}_{\mathcal{Y}}}^2 \right]. \quad (7)$$

However, the resulting test statistic is a degenerate two-sample U-statistic under the null, requiring nonparametric estimates of the conditional mean embeddings; it does not admit weak convergence to a known distribution in general, preventing analytical computation of critical values, leading them to use permutation resampling. In contrast, the squared MMD associated with the SCoDiTE is

$$\| \psi_P \|_{\mathcal{H}}^2 = \mathbb{E}_{P_X} \mathbb{E}_{P_X} \left[ k(X, X') \langle U_{P \mid X}, U_{P \mid X'} \rangle_{\mathcal{H}_{\mathcal{Y}}} \right]. \quad (8)$$

By cross-correlating the $\mathcal{H}_{\mathcal{Y}}$ discrepancies rather than squaring them pointwise, it allows us to recast $H_0$ as the *linear* moment condition $\psi_P = 0 \in \mathcal{H}$, expressed in terms of the identified joint distributions $P_{Y(a), X}$. This enables statistically and computationally efficient inference, as we establish rigorously in the following sections.

### 2.3. Efficient, Doubly-Robust Estimation of the SCoDiTE

The classic one-step estimation procedure for a finite-dimensional parameter (Pfanzagl, 1982) involves 'correcting' an initial (plug-in) estimate using the so-called efficient influence function (EIF) of that parameter (Bickel et al., 1993). However, as the SCoDiTE is Hilbert-valued, classic one-step estimation is not directly applicable. Consequently, we take inspiration from Luedtke & Chung (2024) to develop a one-step estimator for $\psi_P$. The subsequent lemma is key in accomplishing this, as it proves the existence of, and exhibits the form taken by, the EIF of $\psi_P$ at each $P \in \mathcal{P}$.

Before presenting the result, we highlight the main technical challenge underpinning it. Namely, establishing that $P \mapsto \psi_P$ is *pathwise differentiable* relative to the statistical model $\mathcal{P}$. We refer the reader to App. D.1 for the formal presentation of this concept and the subsequent proof. Recall from Sec. 2.1 that $\pi_P(x)$ is the propensity to receive treatment given $X = x$ and $\theta_{P,a}(x)$ is (a $\lambda$-kernelized version of) the outcome model for $X = x$ corresponding to group $A = a$, both under $P$. We now present the EIF.

**Lemma 2.2** (Existence and form of the EIF). *The parameter $\psi$ defined as in Eq. 6 is pathwise differentiable at every $P \in \mathcal{P}$, and has an EIF at each $P$ that takes the form*

$$\phi_P(x, a, y) = \left( \frac{a}{\pi_P(x)} - \frac{1-a}{1 - \pi_P(x)} \right) \left( \Lambda_{x,y} - \theta_{P,a}(x) \right)$$
$$+ \theta_{P,1}(x) - \theta_{P,0}(x) - \psi_P.$$

*Moreover, $0 < \int \| \phi_P(z) \|_{\mathcal{H}}^2 P(dz) < \infty$ for all $P \in \mathcal{P}$.*

The proof is provided in App. D.2. Constructing a one-step estimator with the above EIF yields (a $\lambda$-kernelized version of) an augmented inverse propensity weighted (AIPW) estimator (Glynn & Quinn, 2010; Hines et al., 2022). To see this, note that $\mathbb{E}_P[\phi(Z)] = 0$ by definition. Let $P_n$ be the empirical distribution induced by the i.i.d. dataset $\mathscr{D}$, and let $\widehat{P}_n$ be an independent (not based on $\mathscr{D}$) plug-in estimate of $P$. The one-step estimator is then given by

$$\widehat{\psi}_n := \psi_{\widehat{P}_n} + \mathbb{E}_{P_n} \left[ \phi_{\widehat{P}_n}(Z) \right].$$

In practice, the nuisances $\pi_{\widehat{P}_n}$ and $\{ \theta_{\widehat{P}_n, 1}, \theta_{\widehat{P}_n, 0} \}$ must be estimated from data. To avoid overfitting while maintaining statistical efficiency, we employ cross-fitting (Schick, 1986).

Specifically, let $r \in \{1, 2\}$ denote a data split and fix the complement $s := 3 - r$. Let $\widehat{P}_n^r$ be an initial estimate of the data-generating distribution based on the data split $\mathscr{D}_r$ and $P_n^s$ be the empirical distribution induced by the complementary split $\mathscr{D}_s$. We set the notational convention of using $[\cdot]_n^r$ instead of $[\cdot]_{\widehat{P}_n^r}$. For instance, we let $\psi_n^r$ denote the plug-in parameter estimate $\psi_{\widehat{P}_n^r}$ and $\phi_n^r$ the $\mathcal{H}$-valued EIF estimate $\phi_{\widehat{P}_n^r}$, both of whose nuisances are fitted using $\mathscr{D}_r$. Our cross-fitted one-step estimator is then

$$\bar{\psi}_n := \frac{1}{2} \sum_{r=1}^{2} \left( \psi_n^r + \mathbb{E}_{P_n^s} [\phi_n^r(Z)] \right). \quad (9)$$

We emphasize that Lem. 2.2 provides the theoretical basis for establishing the optimality of $\bar{\psi}_n$. Indeed, we show in Sec. 3.1 that, under suitable conditions, $\bar{\psi}_n$ is asymptotically linear. Intuitively, this means that $\bar{\psi}_n$ behaves almost like an empirical mean: it converges to a tight $\mathcal{H}$-valued Gaussian random variable at the $n^{-1/2}$ rate (see Thm. 3.1 and the discussion surrounding it).

A key property of our estimator, arising from the form of the EIF, is double robustness. Specifically, $\bar{\psi}_n$ remains consistent if either the propensity score models $\{\pi_n^r\}$ or the outcome models $\{\theta_{n,a}^r\}$, but not necessarily both, are correctly specified. We formalize this property in Sec. 3.1.

### 2.4. Permutation-Free, Variance-Aware Inference

The KCD test of $H_0$ (2) uses $M$ permutations to find the empirical p-value (Park et al., 2021). Each permutation involves refitting the outcome models for both treatment groups. The worst-case computational complexity of their algorithm is $\mathcal{O}(Mn^3)$. Since $M$ has been shown to often vary between $10^2$ and $10^3$ for performant permutation-based inference (Davison & Hinkley, 1997), this can quickly become impractical for even moderate datasets.

We propose the 'smoothed' kernel conditional discrepancy (SKCD) to test the reformulated null $H_0 : P_{Y(1), X} = P_{Y(0), X}$ against the complementary alternative. This statis-

tic takes the following quadratic form:

$$\text{SKCD} := \langle \Omega_P(\psi_P), \psi_P \rangle_{\mathcal{H}}, \tag{10}$$

where $\Omega_P$ (to denote potential dependence on $P$) is a continuous self-adjoint positive-definite linear $\mathcal{H} \to \mathcal{H}$ operator. It is evident that when $\Omega_P$ is the identity operator, SKCD reduces to a squared MMD (8). However, this formulation enables richer discrepancies beyond the MMD.

For instance, suppose the appropriately scaled $(\bar{\psi}_n - \psi_P)$ converges weakly to some $\mathcal{H}$-valued limiting distribution with covariance operator $\Sigma_P$. Taking $\Omega_P := [(1 - \varepsilon)\Sigma_P + \varepsilon I]^{-1}$ yields a kernelized Hotelling-type two-sample $T^2$ statistic in the spirit of the two-sample test in Eric et al. (2007), but for a cross-fitted one-step estimator in the more complex counterfactual setting. This Wald-type formulation offers higher power when the true effect lies in a low-variance subspace of $\mathcal{H}$. To our knowledge, this paper is the first to study this class of discrepancies for a conditional distributional causal estimand.

A compelling reason to use SKCD to test $H_0$ is that it circumvents the need to analytically compute or numerically approximate the asymptotic null distribution of degenerate two-sample U-statistics like the KCD. To see this, first note that $\psi_P = 0$ under the null. Let

$$\widehat{\text{SKCD}} := \langle \Omega_n(\bar{\psi}_n), \bar{\psi}_n \rangle_{\mathcal{H}}, \tag{11}$$

where $\Omega_n$ is an appropriate estimator of $\Omega$. Now, if an appropriately scaled $\bar{\psi}_n$ converges weakly to some $\mathcal{H}$-valued limiting distribution under the null that can be analytically derived, then the continuous mapping theorem for Hilbert random elements immediately yields the limiting null distribution $\mathbb{L}$ of the appropriately scaled $\widehat{\text{SKCD}}$. This leads to a simple testing procedure: reject $H_0$ at level $\alpha$ when the scaled $\widehat{\text{SKCD}}$ exceeds the $(1 - \alpha)$-quantile of $\mathbb{L}$. Sec. 3.1 details how the quantile can be estimated *without refitting the nuisance models*, drastically reducing the computational cost of resampling for inference.

In the following section, we establish that the appropriate scaling is $n \cdot \widehat{\text{SKCD}}$. We proceed by rigorously showing that we can (i) analytically derive the root-$n$ rate limiting distribution of $\bar{\psi}_n$, which is optimal in the semiparametric efficiency (in Hilbert spaces) sense, and (ii) efficiently compute both natural SKCD variants, the MMD and Wald-type formulations, in closed form for use as test statistics with known limiting distributions under the null.

## 3. Main Results

### 3.1. Theoretical Guarantees

We henceforth distinguish the true data-generating distribution, denoted by $P_\star \in \mathcal{P}$, from an arbitrary distribution

$P \in \mathcal{P}$. We set the notational convention of using $[\cdot]_\star$ instead of $[\cdot]_{P_\star}$, e.g., $\psi_\star$ denotes the true parameter (6) under $P_\star$, and $\pi_\star$, $\theta_{\star,0}$, and $\theta_{\star,1}$ denote the respective nuisance parameters under $P_\star$, and so on. Let $\widehat{P}_n^r \in \mathcal{P}$ be an initial estimate of $P_\star$ computed using the data split $\mathscr{D}_r$. The goal in this section is to establish the asymptotic normality of $\bar{\psi}_n$ (9) and use it to construct a test of the null, $\psi_\star = 0$.

**Estimation.** The analysis hinges on showing that $\bar{\psi}_n$ is asymptotically linear. This property holds if the estimator's error, $\bar{\psi}_n - \psi_\star$, can be written as an empirical average, with any remaining terms vanishing at a faster than $n^{-1/2}$ rate. Adding zero to $\bar{\psi}_n - \psi_\star$ and rearranging terms yields

$$\bar{\psi}_n - \psi_\star = \frac{1}{2} \sum_{r=1}^{2} \mathbb{E}_{P_n^s} [\phi_\star(Z)] + \frac{1}{2} \sum_{r=1}^{2} (\mathcal{R}_n^r + \mathcal{D}_n^r), \tag{12}$$

where $\mathcal{R}_n^r := \psi_n^r + \mathbb{E}_\star[\phi_n^r(Z)] - \psi_\star$ and $\mathcal{D}_n^r := \mathbb{E}_{P_n^s}[\phi_n^r(Z) - \phi_\star(Z)] - \mathbb{E}_\star[\phi_n^r(Z) - \phi_\star(Z)]$. The following theorem provides sufficient conditions on the convergence rates of the nuisance estimators to ensure that both $\max_r \|\mathcal{R}_n^r\|_{\mathcal{H}}$ and $\max_r \|\mathcal{D}_n^r\|_{\mathcal{H}}$ vanish at the required rate. Slutsky's lemma and a Hilbert central limit theorem consequently imply the weak convergence of $\sqrt{n}(\bar{\psi}_n - \psi_\star)$.

**Theorem 3.1** (Weak convergence). *Let $\phi_\star$ be the EIF of $\psi$ at $P_\star$. For $r \in \{1, 2\}$, suppose $\widehat{P}_n^r$ is such that:*

*(i)* $\|\pi_n^r - \pi_\star\|_{L^2(P_\star, X)} = O_p(n^{-\tau_r})$ *for scalar $\tau_r > 0$,*

*(ii)* $\|\theta_{n,a}^r - \theta_{\star,a}\|_{L^2(P_\star, X; \mathcal{H})} = O_p(n^{-\gamma_{a,r}})$ *for scalar $\gamma_{a,r} > 0$ for each $a \in \{0, 1\}$, and*

*(iii)* $\tau_r + \min\{\gamma_{0,r}, \gamma_{1,r}\} > 1/2$.

*Then, letting '$\rightsquigarrow$' denote weak convergence in $\mathcal{H}$, we have*

1. $\bar{\psi}_n - \psi_\star = \frac{1}{n} \sum_{i=1}^{n} \phi_\star(Z_i) + o_p(n^{-1/2})$,
2. $\sqrt{n}(\bar{\psi}_n - \psi_\star) \rightsquigarrow \mathbb{H}$,

*where $\mathbb{H}$ is a tight $\mathcal{H}$-valued random variable such that $\langle \mathbb{H}, h \rangle_{\mathcal{H}} \sim \mathcal{N}\left(0, \mathbb{E}_\star\left[\langle \phi_\star(Z), h \rangle_{\mathcal{H}}^2\right]\right)$ for every $h \in \mathcal{H}$.*

The proof is provided in App. E.2. Condition (iii) is a double robustness condition that ensures the remainder $\mathcal{R}_n^r$ converges to zero if the product of the nuisance estimation rates goes to zero faster than $n^{-1/2}$. The empirical process term $\mathcal{D}_n^r$ is controlled using the consistency of the EIF estimate, which we show holds under conditions (i) and (ii).

Now we discuss the statistical efficiency of our estimator. Since a direct Cramér-Rao lower bound does not always exist in such RKHS settings, we analyze this in a more general framework. As we establish in the following theorem, the proposed cross-fitted one-step estimator is asymptotically efficient under the conditions of Thm. 3.1. Intuitively, this means that among the limiting distributions of estimators

of $\psi_\star$, the weak limit $\mathbb{H}$ of our estimator is optimal in the 'smallest spread' sense. We use the shorthand $\psi_{s,\epsilon}$ to mean $\psi_{P_{s,\epsilon}}$.

**Theorem 3.2** (Local asymptotic minimax optimality). *For any score $s \in L_0^2(P_\star)$, let $\{P_{s,\epsilon}\} \subset \mathcal{P}$ be a QMD submodel such that $P_{s,0} = P_\star$. Define the local asymptotic minimax risk for an estimator sequence $(\check{\psi}_n)_{n=1}^\infty$ as*

$$\mathrm{LAMRisk}_\rho(\check{\psi}_n; P_\star) :=$$
$$\sup_I \liminf_{n\to\infty} \sup_{s\in I} \mathbb{E}_{s,\frac{1}{\sqrt{n}}}\left[\rho\left(\sqrt{n}\left[\check{\psi}_n - \psi_{s,\frac{1}{\sqrt{n}}}\right]\right)\right],$$

*where $\rho : \mathcal{H} \to \mathbb{R}$ is a nonnegative map, the first supremum is over all finite subsets of $L_0^2(P_\star)$, and the expectation is under the product measure $P_{s,1/\sqrt{n}}^n$. Suppose the conditions of Thm. 3.1 hold. Further, let $(\widetilde{\psi}_n)_{n=1}^\infty$ be any Borel-measurable estimator sequence and $\rho$ be any subconvex function that is continuous a.s. under the law of $\mathbb{H}$. Provided that the sequence $\rho(\sqrt{n}(\bar{\psi}_n - \psi_{s,1/\sqrt{n}}))$ is asymptotically uniformly integrable under $P_{s,1/\sqrt{n}}$, we have:*

$$\mathrm{LAMRisk}_\rho(\widetilde{\psi}_n; P_\star) \geq \mathbb{E}_\star[\rho(\mathbb{H})] = \mathrm{LAMRisk}_\rho(\bar{\psi}_n; P_\star).$$

The proof, presented in App. E.3, follows from the pathwise differentiability of $P \mapsto \psi_P$ and the convolution and minimax theorems for Hilbert-valued estimators (van der Vaart & Wellner, 2023, Theorems 3.12.2 and 3.12.5). The final equality is achieved via the convergence of means for asymptotically uniformly integrable sequences (van der Vaart & Wellner, 2023, Theorem 1.11.3).

We highlight the relationship between our estimator and existing kernel-based procedures for *marginal* DTEs. Martinez Taboada et al. (2023) present a "cross-U-statistic" estimator that relies on a single data split. While it attains the $\sqrt{n}$ rate, it is asymptotically linear on only half the sample; this results in an effective sample size of $n/2$, precluding local asymptotic minimax optimality (Kim & Ramdas, 2024). In contrast, in a work concurrent to Martinez Taboada et al. (2023), Luedtke & Chung (2024) construct a doubly robust cross-fitted estimator for marginal DTEs that is asymptotically linear over the entire sample, thereby attaining optimality. Our estimator $\bar{\psi}_n$ is a nontrivial extension of this full-sample one-step construction: under the conditions of Theorem 3.1, it is asymptotically linear in $\mathcal{H}$ over all $n$ observations, thereby achieving local asymptotic minimax optimality for conditional DTEs.

**Inference.** We now propose a test for the sharp null hypothesis $H_0 : \psi_\star = \psi_0$ (e.g., $\psi_0 = 0$ for the null $H_0$ in Eq. 2). Let $\mathcal{W}$ denote the set of continuous self-adjoint positive-definite linear operators on $\mathcal{H}$. We define our test statistic as

$$T_n := n\langle\Omega_n(\bar{\psi}_n - \psi_0), \bar{\psi}_n - \psi_0\rangle_\mathcal{H}, \qquad (13)$$

where $\Omega_n \in \mathcal{W}$ is a consistent estimator for a possibly-$P_\star$-dependent operator $\Omega_\star \in \mathcal{W}$ (e.g., the identity, or a regularized inverse covariance operator as discussed in Sec. 2.4). Note that when $\psi_0 = 0$, $T_n$ corresponds to $n \cdot \widehat{\mathrm{SKCD}}$ (11). We let submodel $\mathcal{P}_0 \subseteq \mathcal{P}$ denote the set of all distributions for which the null hypothesis $H_0$ holds.

Under $H_0$ and consistent estimation of $\Omega_\star$, Thm. 3.1 and the continuous mapping theorem imply that $T_n \rightsquigarrow \langle\Omega_\star(\mathbb{H}), \mathbb{H}\rangle_\mathcal{H}$. This limiting distribution depends on $P_\star$, which is generally unknown. Therefore, a valid test requires a consistent estimate of the $(1-\alpha)$-quantile of this limit, $q_\alpha$. Alg. 1 bootstraps the empirical mean of the influence function to compute this estimate, $\widehat{q}_{n,\alpha}$. Our SKCD test rejects $H_0$ at level $\alpha$ if $T_n > \widehat{q}_{n,\alpha}$.

---

**Algorithm 1** SKCD test via bootstrapping the EIF

---

**Input:** Data $\mathscr{D} = \{Z_i\}_{i=1}^n$, null $\psi_0$ (default 0), level $\alpha$, bootstrap samples $B$, estimate $\Omega_n$ of operator $\Omega_\star$.
  Split $\{1,\ldots,n\}$ into index sets $\mathcal{I}_1, \mathcal{I}_2$ for cross-fitting;
  Let $\bar{\psi}_n = n^{-1}\sum_{i=1}^n \varphi_i$; Compute $\bar{\psi}_n$ (9) and $T_n$ (13);
  **for** $b = 1$ to $B$ **do**
    Draw $\xi_j \sim \mathrm{Multinomial}(n_r, 1/n_r, \ldots, 1/n_r) - 1$ for $j \in \mathcal{I}_r, r \in \{1,2\}$; Compute $\Delta_n^{(b)} = n^{-1}\sum_{i=1}^n \xi_i\varphi_i$ and $T_n^{(b)} = n\langle\Omega_n(\Delta_n^{(b)}), \Delta_n^{(b)}\rangle_\mathcal{H}$;
  **end for**
  Set $\widehat{q}_{n,\alpha}$ as $(1-\alpha)$-quantile of $\{T_n^{(b)}\}_{b=1}^B$;
  **Return:** $\mathbb{I}(T_n > \widehat{q}_{n,\alpha})$.

---

Importantly, unlike permutation tests (as in Park et al., 2021) that require refitting nuisance models $\theta_{n,a}$ in every permutation, our approach computes the EIF estimates only once. In the bootstrap loop, we simply re-weight these fixed estimates using random, zero-centered multinomial draws $\xi_i$ to simulate the limit distribution $\mathbb{H}$. In fact, for specific closed-forms of $T_n$ (see Sec. 3.2), we can amortize the most expensive operations, achieving a worst-case complexity of $\mathcal{O}(n^3 + Bn^2)$. Compared to a cross-MMD based test (as in Martinez Taboada et al., 2023), our test achieves optimal asymptotic power while maintaining equivalent computational complexity, provided that nuisance estimation is super-quadratic.

**Theorem 3.3** (Validity of the SKCD test in Alg. 1). *If the conditions of Thm. 3.1 hold, $\Omega_\star \in \mathcal{W}$, and $\Omega_n \in \mathcal{W}$ satisfies $\|\Omega_n - \Omega_\star\|_{\mathrm{op}} = o_p(1)$, then*

1. *(type 1 error control)* $\lim_{n\to\infty} P_\star^n\{T_n > \widehat{q}_{n,\alpha}\} = \alpha$ *for all $P_\star \in \mathcal{P}_0$, and*

2. *(test consistency)* $\lim_{n\to\infty} P_\star^n\{T_n > \widehat{q}_{n,\alpha}\} = 1$ *for any fixed $P_\star \in \mathcal{P} \setminus \mathcal{P}_0$.*

The proof hinges on bootstrap consistency, and is deferred to App. F.1.1. While our test provides a decision rule for

rejecting the null hypothesis of no global conditional distributional effect, it does not immediately reveal the nature of the possible heterogeneity upon rejecting the null. To enable finer interpretation of the SCoDiTE, we can construct a uniform confidence band for the witness function $(x, y) \mapsto \psi_\star(x, y)$ by simply inverting our testing procedure, i.e., by evaluating the support function of the $(1 - \alpha)$-confidence ellipsoid implied by the test. This guarantees uniform coverage over the entire domain $\mathcal{Z}$. When using the Wald-type formulation, our approach adapts the width of the band to the local geometry of the operator $\Omega_n$, allowing for tighter bands in regions of the covariate space with lower variance. Let $\mathscr{W}_{\mathrm{inv}} \subset \mathscr{W}$ consist of all $\Omega \in \mathscr{W}$ that are boundedly invertible.

**Theorem 3.4** (Uniform confidence band for the SCoDiTE). *Suppose the conditions of Thm. 3.1 hold, $\Omega_\star \in \mathscr{W}_{\mathrm{inv}}$, and the bootstrap quantile $\widehat{q}_{n,\alpha}$ is constructed (Alg. 1) using $\Omega_n \in \mathscr{W}_{\mathrm{inv}}$ such that $\|\Omega_n - \Omega_\star\|_{\mathrm{op}} = o_p(1)$. Define $w_n : \mathcal{X} \times \mathcal{Y} \to \mathbb{R}$ that satisfies $w_n^2(x, y) := \langle \Lambda_{x,y}, \Omega_n^{-1} \Lambda_{x,y} \rangle_{\mathcal{H}} \, \widehat{q}_{n,\alpha}/n$, and let $B_n(x, y) := [\bar{\psi}_n(x, y) - w_n(x, y), \; \bar{\psi}_n(x, y) + w_n(x, y)]$. Then,*

$$\lim_{n \to \infty} P_\star^n \left( \psi_\star(x, y) \in B_n(x, y) \quad \text{for all } x, y \right) \geq 1 - \alpha.$$

The proof, provided in App. F.1.2, relies on the Cauchy-Schwarz inequality in RKHSs. The band $B_n$ allows practitioners to visualize the SCoDiTE and helps identify regions of covariates and outcomes where the effect is statistically significant. We can also construct tighter pointwise-in-$x$ uniform-in-$y$ confidence bands by restricting the test statistic in Alg. 1 to $\mathcal{H}_x = \{h(x, \cdot) : h \in \mathcal{H}\}$. We demonstrate this utility in Sec. 4.2, where we use these bands to localize wealth impacts for distinct household profiles.

**Max-aggregated test.** Although the global null $H_0$ (2) is unchanged for any bounded characteristic kernel $\lambda$, the finite-sample power of the SKCD test can depend on how well the chosen bandwidth matches the scale of the effect: smaller bandwidths can more easily detect localized discrepancies, but may be noisier, whereas larger bandwidths borrow strength over broader covariate regions but may oversmooth fine-scale effects. The median heuristic (Fukumizu et al., 2009) is a popular choice of bandwidth, though not necessarily optimal or stable for power. To reduce possible sensitivity to the choice of bandwidth, we propose SKCDAgg, a multiple testing procedure inspired by Schrab et al. (2023) that combines evidence across a fixed finite grid of bandwidths.

Let $J < \infty$ be fixed. For each bandwidth choice indexed by $j \in [J]$, let $\lambda^j = k^j \otimes \ell^j$ be a bounded characteristic product kernel with RKHS $\mathcal{H}^j$. Let $\psi_\star^j$, $\bar{\psi}_n^j$, $\Omega_\star^j$ and $\Omega_n^j$ denote the bandwidth-specific SCoDiTE, one-step estimator, and

the operator and its estimate, respectively. Let $T_n^j$ be the test statistic (13) with explicit dependence on the bandwidth indexed by $j$. We consider $H_0$, i.e., $\psi_\star^j = 0 \; \forall j \in [J]$. Under $H_0$ and consistent estimation of $\Omega_\star^j$, Thm. 3.1 and the continuous mapping theorem imply that $T_n^j \rightsquigarrow \left\langle \Omega_\star^j(\mathbb{H}^j), \mathbb{H}^j \right\rangle_{\mathcal{H}^j}$.

For a fixed $\tau \in (0, 1)$, let $q_\tau^j$ denote the $(1 - \tau)$-quantile of the above limit distribution, and let $\widehat{q}_{n,\tau}^j$ estimate it, e.g., via Alg. 1. Define the max-aggregated test statistic

$$S_n^{\mathrm{Agg}} := \max_{j \in [J]} T_n^j / \widehat{q}_{n,\tau}^j. \tag{14}$$

The standardization by $\widehat{q}_{n,\tau}^j$ calibrates each statistic $T_n^j$ to the scale of its own marginal null distribution, ensuring that the aggregation compares evidence for different bandwidths on a common scale. Note that $S_n^{\mathrm{Agg}}$ can be obtained by repeating Alg. 1 with the same bootstrap replicates for each $j \in [J]$. We can then compute the $(1 - \alpha)$-quantile of $\{S_n^{\mathrm{Agg},(b)}\}_{b=1}^B$, where $S_n^{\mathrm{Agg},(b)} := \max_{j \in [J]} T_n^{j,(b)} / \widehat{q}_{n,\tau}^j$; we denote this by $\widehat{c}_{n,\alpha}^{\mathrm{Agg}}$. Our proposed SKCDAgg test rejects $H_0$ at level $\alpha$ if $S_n^{\mathrm{Agg}} > \widehat{c}_{n,\alpha}^{\mathrm{Agg}}$.

**Proposition 3.5** (Validity of the SKCDAgg test). *Suppose that the conditions of Thm. 3.1 hold for each $j \in [J]$, and $\Omega_n^j, \Omega_\star^j \in \mathscr{W}$ with $\left\| \Omega_n^j - \Omega_\star^j \right\|_{\mathrm{op}} = o_p(1)$ for each $j \in [J]$. Assume $q_\tau^j > 0$ for every $j \in [J]$. Then,*

1. *$\lim_{n \to \infty} P_\star^n \left( S_n^{\mathrm{Agg}} > \widehat{c}_{n,\alpha}^{\mathrm{Agg}} \right) = \alpha$ under $H_0$, and*
2. *$\lim_{n \to \infty} P_\star^n \left( S_n^{\mathrm{Agg}} > \widehat{c}_{n,\alpha}^{\mathrm{Agg}} \right) = 1$ if $\psi_\star^j \neq 0$ for some $j \in [J]$.*

The proof, deferred to App. F.2, proceeds by proving a joint weak convergence result for the finite collection of the bandwidth-specific one-step estimators in the direct-sum Hilbert space $\bigoplus_{j=1}^J \mathcal{H}^j$, and then applying the same bootstrap arguments jointly across the fixed finite bandwidth grid.

### 3.2. Closed-Form Estimators for the SKCD Statistic

We now derive exact computable expressions for the SKCD test statistic. Our constructions are agnostic to the choice of propensity models $\pi_n^r$ and accommodate a range of outcome models $\theta_{n,a}^r$, including kernel ridge regression, distributional random forests (Näf et al., 2023), and deep kernel methods (Shimizu et al., 2024), provided these estimates lie in the finite-dimensional subspace $\mathcal{F}_n := \mathrm{span}\{\Lambda_{x_i,y_j}\}_{i,j=1}^n$. Under this condition, $\bar{\psi}_n$ lies in $\mathcal{F}_n$, allowing the SKCD to be estimated in closed-form using only Gram matrices $[\mathbf{K}]_{ij} = k(x_i, x_j)$ and $[\mathbf{L}]_{ij} = \ell(y_i, y_j)$.

First, for the MMD formulation ($\Omega_\star = \Omega_n = I$), we construct a weight matrix. Let $\boldsymbol{\beta}_a^r(x) \in \mathbb{R}^n$ denote the vector of coefficients for the outcome model $\theta_{n,a}^r(x) = \sum_j [\boldsymbol{\beta}_a^r(x)]_j \Lambda_{x,y_j}$ such that $[\boldsymbol{\beta}_a^r(x)]_j = 0$ for any observation where $j \notin \mathcal{I}^r$ or $a_j \neq a$. For any index $i$, let

$s(i) \in \{1, 2\}$ be the split containing $i$, and $r(i) = 3 - s(i)$ be the complement. Define $\pi_n^{r(i)}(x_i) := w_i$. We construct $\mathbf{C} \in \mathbb{R}^{n \times n}$ entry-wise as:

$$[\mathbf{C}]_{ij} := \begin{cases} \frac{1}{2n_{s(i)}} \left( \frac{a_i}{w_i} - \frac{1-a_i}{1-w_i} \right) & \text{if } j = i \\ \frac{1}{2n_{s(i)}} \left[ \left( 1 - \frac{a_i}{w_i} \right) [\boldsymbol{\beta}_1^{r(i)}(x_i)]_j \\ + \left( \frac{1-a_i}{1-w_i} - 1 \right) [\boldsymbol{\beta}_0^{r(i)}(x_i)]_j \right] & \text{if } j \neq i \end{cases} \quad (15)$$

The diagonal terms of $\mathbf{C}$ hold inverse propensity weights, while the off-diagonal block terms capture the augmentation corrections. With this representation, the squared RKHS norm of our estimator reduces to a trace operation, as established in the following result.

**Proposition 3.6** (Closed-form MMD statistic from Alg. 1). *If $\Omega_n = I$ and $\mathbf{C}$ is as constructed using* (15)*, then the squared MMD test statistic from Alg. 1 takes the form $T_n^{\mathrm{MMD}} := n \left\| \bar{\psi}_n \right\|_{\mathcal{H}}^2 = n \left\langle \mathbf{C}, \mathbf{KCL} \right\rangle_{\mathrm{F}}$.*

We prove this result in App. G.1. The MMD statistic can thus be evaluated with the standard $\mathcal{O}(n^3)$ worst-case complexity for kernel methods, ensuring that our test does not incur an extra prohibitive computational overhead. For computational scalability, we also provide a low-rank approximation using the Nyström method, which reduces the cost of evaluating this statistic to $\mathcal{O}(n^2 r)$ for $r$ inducing points (see App. H.2). While our statistical guarantees from Sec. 3.1 are established for the exact closed-form statistics, we expect that they can be extended to the Nyström versions under standard conditions on kernel spectral decay by appropriately controlling the induced approximation errors, as suggested by existing low-rank approximation theory (Bach, 2013; Rudi et al., 2015; Chatalic et al., 2022).

We next turn to the Wald-type statistic, which incorporates the covariance structure of the estimator. Let $\Sigma_\star(h) := \mathbb{E}_\star[\langle \phi_\star(Z), h \rangle_{\mathcal{H}} \phi_\star(Z)]$ denote the covariance operator of $\mathbb{H}$ and let $\Sigma_n(h) := \frac{1}{2} \sum_{r=1}^2 \mathbb{E}_{P_n^s}[\langle \phi_n^r(Z), h \rangle_{\mathcal{H}} \phi_n^r(Z)]$ be a finite-dimensional estimator. The choice of $\Omega_\star$ corresponds to the regularized inverse of $\Sigma_\star$, and so we consider the finite-dimensional operator

$$\Omega_n := ((1 - \varepsilon)\Sigma_n + \varepsilon I)^{-1} \quad (16)$$

to compute $T_n^{\mathrm{Wald}}$. A naïve inversion on the tensor product space would involve an $n^2 \times n^2$ matrix, incurring a prohibitive $\mathcal{O}(n^6)$ worst-case complexity. To avoid this, we exploit the fact that the empirical covariance $\Sigma_n$ has rank $\leq n$, constructing auxiliary matrices that capture the cross-fitting structure and the low-rank factors. Let $[\mathbf{E}]_{ij}$ represent the pure outcome model coefficients (case 2 of Eq. 15 *without* the propensity weights). Then, define:

$$[\mathbf{D}^s]_{ij} := \mathbb{1}_{\{i \in \mathcal{I}^s\}} \sqrt{2n_s} [\mathbf{C}]_{ij} \quad (17)$$
$$[\mathbf{V}^s]_{ij} := \mathbb{1}_{\{i \in \mathcal{I}^s\}} \sqrt{2/n_s} [\mathbf{E}]_{ij}, \quad \mathbf{W}^s := \mathbf{D}^s - n_s \mathbf{V}^s.$$

Let $\mathbf{d}^s, \mathbf{v}^s, \mathbf{w}^s$ be the row-wise vectorizations of these matrices respectively. Define $\mathbf{G} := \mathbf{K} \otimes \mathbf{L}$, and $\mathbf{S}^s := (I_n \bullet \mathbf{D}^s)^\top$, where '$\bullet$' denotes the row-wise Kronecker product. We stack these components into two block matrices $\mathbf{T}, \mathbf{U} \in \mathbb{R}^{n^2 \times (2n+4)}$ as follows:

$$\mathbf{T} := \begin{bmatrix} \mathbf{GS}^1 & \mathbf{GS}^2 & \mathbf{Gv}^1 & \mathbf{Gv}^2 & \mathbf{Gw}^1 & \mathbf{Gw}^2 \end{bmatrix},$$
$$\mathbf{U} := \begin{bmatrix} \mathbf{S}^1 & \mathbf{S}^2 & -\mathbf{d}^1 & -\mathbf{d}^2 & -\mathbf{v}^1 & -\mathbf{v}^2 \end{bmatrix}. \quad (18)$$

With this, the Wald-type statistic reduces to a low-rank correction of the MMD statistic, as established below.

**Proposition 3.7** (Closed-form Wald-type statistic from Alg. 1). *If $\Omega_n$ is as in* (16) *and $\mathbf{c} := \mathrm{vec}(\mathbf{C}^\top)$ is constructed from* (15)*, then the Wald-type statistic from Alg. 1 can be computed in $\mathcal{O}(n^3)$ operations as*

$$T_n^{\mathrm{Wald}} := n \left\langle \Omega_n(\bar{\psi}_n), \bar{\psi}_n \right\rangle_{\mathcal{H}} = \frac{n}{\varepsilon} \left\langle \mathbf{C}, \mathbf{KCL} \right\rangle_F$$
$$- \frac{n(1-\varepsilon)}{\varepsilon} \mathbf{c}^\top \mathbf{T} \left( \varepsilon \mathbf{I} + (1-\varepsilon)\mathbf{U}^\top \mathbf{T} \right)^{-1} \mathbf{U}^\top \mathbf{Gc}.$$

The proof of this result is provided in App. G.2. The operator $\Sigma_n$ estimates the covariance of the EIF in a tensor product RKHS. Consequently, deriving $\mathbf{U}^\top \mathbf{T}$ tractably requires applications of identities involving face-splitting and Khatri-Rao products. This reduces the dominating computation of $T_n^{\mathrm{Wald}}$ to inverting a $(2n + 4) \times (2n + 4)$ matrix, yielding the same computational complexity as $T_n^{\mathrm{MMD}}$. The subcubic Nyström approximation is provided in App. H.2.

Notably, while Luedtke & Chung (2024) suggest a test for marginal DTEs using their one-step estimator, they do not derive closed-form expressions for the resulting test statistic; in contrast, our derivations enable testing of conditional DTEs while avoiding approximation error. Crucially, these expressions allow us to further exploit the bilinearity of the inner product in Eq. 13 to pre-compute all objects requiring $\mathcal{O}(n^3)$ operations in the SKCD test. Evaluations within the bootstrap loop then simply project the random multipliers onto these pre-computed objects, with each resampling requiring only $\mathcal{O}(n^2)$ operations (see App. H.1). Moreover, employing our Nyström implementations with $r^2 < n$ further reduces the computational complexity of the test for both MMD and Wald-type variants to $\mathcal{O}(n^2 r + Bnr^2)$. Under a polylogarithmically growing $r$, this is near-quadratic (in $n$) in the pre-computation phase and near-linear in the bootstrap phase (see App. H.2 for details).

## 4. Experiments

### 4.1. Simulation: Distribution Shift in Images

We investigate the finite-sample size and power of our SKCD test at level $\alpha = 0.05$. Our simulation design uses the MNIST dataset (Deng, 2012) to create scenarios where

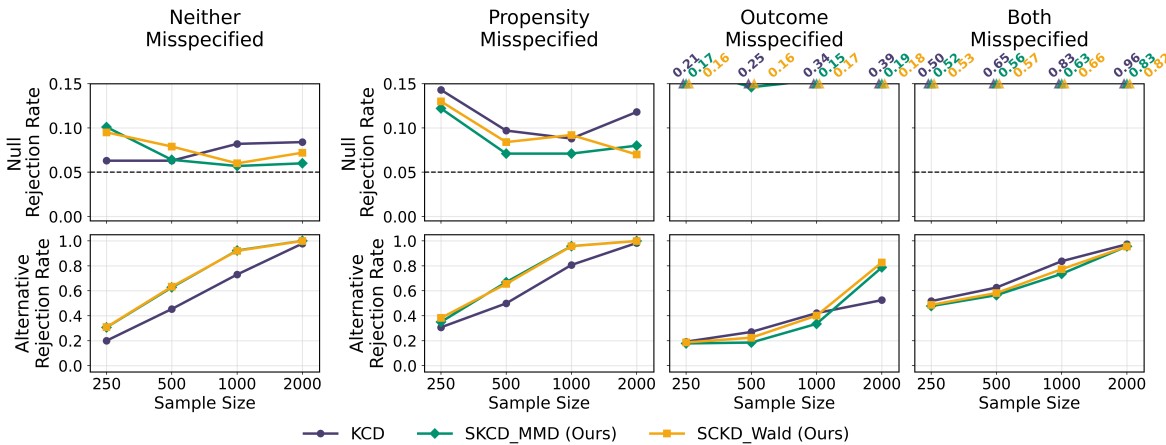

*Figure 2.* Type 1 error and power at $\alpha = 0.05$ across sample sizes and nuisance misspecification regimes. **(Left)** Scenario satisfying asymptotic guarantees (the product of nuisance estimation errors is $o_p(n^{-1/2})$). **(Right)** Robustness checks under model misspecification. The proposed tests benefit from double robustness of the estimator; type 1 error is closer than baseline to the nominal level under propensity misspecification; under outcome misspecification, type 1 error is inflated but stable, while power increases with sample size.

treatment effects manifest as distribution shifts that are challenging to detect. We let both covariates $X$ and outcomes $Y$ be PCA embeddings of learned image representations (in $\mathbb{R}^5$) using a ResNet-18-based encoder. Treatment $A$ is assigned via a Bernoulli draw parameterized by a non-linear function of the covariates, designed to maintain overlap. We provide all experimental specifications and implementation details in App. I.1.

Under the null, outcomes are generated after the images for both groups undergo random intensity changes, ignoring treatment. Under the alternative, the treated group images undergo an additional rotation whose angle depends non-linearly on $X$. Thus, the treatment induces a multivariate distributional effect that is not limited to the mean and varies with the covariates.

We compare our proposed SKCD test, using both MMD and Wald-type statistics (referred to as SKCD_MMD and SKCD_Wald respectively), against the baseline KCD test (Park et al., 2021). We employ Gaussian kernels for both covariate and outcome spaces, using the median heuristic (Fukumizu et al., 2009) for the bandwidth. All methods use kernel ridge regression for the outcome models and gradient-boosted decision trees for the propensity model. We evaluate robustness across four regimes: (1) *Neither Misspecified*; (2) *Propensity Misspecified*; (3) *Outcome Misspecified*; and (4) *Both Misspecified*. Misspecification is achieved by withholding the principal components that drive treatment assignment and effect heterogeneity.

We sample a subset of size $n \in \{250, 500, 1000, 2000\}$ from the simulated data $\{X_i, A_i, Y_i\}_{i=1}^{45k}$ with replacement. The plots in Fig. 2 report the rejection rates at level $\alpha = 0.05$ for 1000 Monte Carlo (MC) replicates of each experimental configuration for all three tests under consideration. For

SKCD_Wald, $\varepsilon$ is set using $\varepsilon/(1-\varepsilon) = \gamma \operatorname{tr}(\mathbf{U}^\top \mathbf{T})$ with $\gamma = 1/3$ (see App. G.2.3 for the relevant discussion).

In the *Neither Misspecified* regime, the proposed SKCD test variants show slightly inflated type 1 error at smaller sample sizes that—consistent with our theory—approaches nominal as sample size grows. SKCD_MMD achieves type 1 error quite close to nominal even in the *Propensity Misspecified* regime, while the baseline KCD suffers significant inflation. Even in the *Outcome Misspecified* setting, where type 1 error control is challenging, both SKCD variants prove notably more stable than KCD, which diverges sharply as $n$ increases. Under the alternative hypothesis, power increases with sample size across all (valid) configurations; however, our proposed methods consistently outperform KCD. This is most visible in the *Outcome Misspecified* regime, where (though under inflated type 1 error) both SKCD_MMD and SKCD_Wald achieve $\sim 80\%$ power at $n = 2000$ while the baseline plateaus. Additional experiments in App. J (Fig. 4) using known propensity scores show that all methods achieve nominal type I error control when outcome is correctly specified, while the observed advantages of both SKCD variants over KCD under outcome misspecification become even more pronounced.

To assess our double robustness guarantee for the estimator $\bar{\psi}_n$ (9), we analyze its convergence under the null ($\psi_\star = 0$). In App. J (Fig. 5), we plot its empirical mean squared error (MSE) in the RKHS norm, i.e., the average of $\|\bar{\psi}_n\|_{\mathcal{H}}^2 = n^{-1} T_n^{\text{MMD}}$ across MC replicates. We observe that the MSE decreases sharply as $n$ increases if even one nuisance model is correctly specified, consistent with our theory.

**Additional experiments.** We examine our implementation choices in further detail in the appendix. In App. J.1,

we study the statistical performance of `SKCD_Wald` over a range of $\gamma$ values. Fig. 6 shows that $\gamma = 1/3$ is a reasonable default in finite samples, although smaller $\gamma$ can further increase power by relying on more covariance directions, provided the covariance is estimated well.

We also empirically study the sensitivity to bandwidth choice, with details provided in App. J.2. We vary the covariate kernel bandwidth and compare the individual SKCD tests with the max-aggregated SKCDAgg test. Figs. 7 and 8 show that while the median heuristic is a reasonable default, the SKCDAgg test is competitive for both size and power, and is generally more stable across sample sizes.

Finally, we repeat the original experiments with Nyström approximations of the MMD and Wald-type statistics under varying sublinear (in $n$) rank schedules, using the implementation detailed in App. H.2. Empirically (see App. J.3, Figs. 9 and 10), the low-rank variants largely preserve the qualitative type 1 error and power behavior, and we observe large runtime gains for `SKCD_Wald` (up to 5.4×), where low-rank structure substantially reduces the matrix inversion cost.

### 4.2. Real Data: Impact of 401(k) Eligibility on Household Wealth

We apply our methods to Wave 4 ($9,915$ households) of the 1990 Survey of Income and Program Participation (Chernozhukov & Hansen, 2004; Benjamin, 2003; Gelber, 2011; Kallus & Oprescu, 2023) to study the effect of 401(k) eligibility ($A$) on household wealth. All experimental specifications and implementation details are provided in App. I.2.

Following recent work (Näf & Susmann, 2024), we analyze a multivariate outcome $Y \in \mathbb{R}^3$ comprising Net Financial Assets (TFA), Net non-401(k) Assets (NIFA), and Total Wealth (TW). The pre-treatment covariates $X$ comprise four continuous features—age, income, family size, education, and five categorical—defined-benefit plan, marital status, dual earner, IRA participation, and home ownership.

The proposed SKCD test rejects the global null $H_0$ : $P_{Y(1),X} = P_{Y(0),X}$ at level $\alpha = 0.05$. Extending the analysis, we construct 95% uniform-in-$y$ confidence bands for the SCoDiTE witness function $\psi_\star(x, \cdot)$ by adapting the construction from Thm. 3.4 to the RKHS slice $\{h(x, \cdot) : h \in \mathcal{H}\}$. Due to the infeasibility of visualizing the full 3D witness function surface over $\mathcal{Y}$, we compute 1D cross-sections by varying each wealth component $Y_j$ over its support while fixing the other two at their sample means. This allows us to localize the detectable effect to specific regions of the outcome space for household profiles characterized by $x$.

Fig. 3 displays these witness function cross-sections for two distinct households that illustrate the effect heterogeneity. Individual 1 (top) is a 58-year-old individual with moderate income ($30.3k), in a family of size 1, with high education (18 years), possessing an IRA and a defined-benefit plan. Individual 2 (bottom) is a 36-year-old individual with similar income ($34k) but a large family (size 13), low education (4 years), and no other retirement plans.

For Individual 1, the confidence band along the first wealth measure excludes zero over significant regions. In particular, the estimated witness function for Net Financial Assets exhibits a negative-to-positive swing. This suggests that, holding other assets at their average levels, 401(k) eligibility shifts the distribution of financial assets for this demographic: reducing the density of low asset values and increasing the density of high asset values. For Individual 2, the estimated witness function cross-sections are essentially flat, and the confidence bands contain zero across the entire domain of each wealth measure, providing no evidence of wealth impact from 401(k) eligibility.

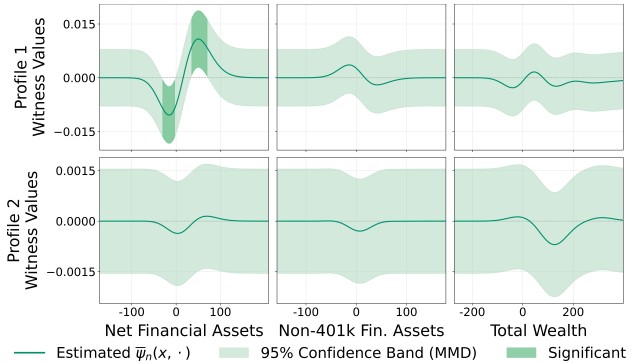

*Figure 3.* Estimated witness functions with 95% uniform confidence bands for two household profiles (rows) across three wealth outcomes (columns; in $1k units). Shaded regions indicate statistical significance. While Profile 1 exhibits a distributional shift along the first axis, Profile 2 shows no detectable effect.

## 5. Discussion

We introduce the SCoDiTE framework, bridging kernel mean embeddings and semiparametric efficiency theory to rigorously test for conditional distributional treatment effects. We provide the first doubly robust, asymptotically optimal estimator for this setting, along with a permutation-free test for valid inference, for which we derive MMD and Wald-type test statistics in closed form.

Future work could focus on extending this framework to continuous treatments or instrumental variable settings. Furthermore, while our Wald-type statistic improves power, data-driven selection of the regularization parameter $\varepsilon$ remains an open problem. Finally, formally extending our guarantees under Nyström approximation is a natural theoretical direction.

## Acknowledgments

This work was supported by the Patient-Centered Outcomes Research Institute (PCORI, ME-2024C2-39990). The content is solely the responsibility of the authors and does not necessarily represent the official views of the funding agency.

## Impact Statement

This paper advances causal inference methodology by developing a valid statistical test for conditional distributional treatment effects. While our methods have broad applicability in healthcare, economics, and policy evaluation, the ethical implications depend on the specific application domain. Practitioners should ensure valid causal assumptions and consider fairness and equity implications when applying these methods to inform consequential decisions.

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

# Appendix

## A. Extended Related Work

Muandet et al. (2021) introduced kernel-based marginal DTE estimators. Fawkes et al. (2024) developed MMD-based doubly robust test statistics for marginal DTEs though they did not provide complete theoretical guarantees such as type 1 error control for inference. Martinez Taboada et al. (2023) provided a test based on a doubly robust estimator to test marginal DTEs but their estimator incurs a loss in asymptotic efficiency relative to an optimal estimator by a factor of $\sqrt{2}$ due to their sample splitting-based approach. Luedtke & Chung (2024) developed a one-step estimator for testing marginal DTEs that avoids this penalty, but did not derive closed-form test statistics or consider conditional DTEs.

Eric et al. (2007) proposed a kernelized Hotelling's $T^2$ statistic using a plug-in regularized inverse covariance operator for standard two-sample testing. More recently, this framework has been studied for goodness-of-fit testing (Balasubramanian et al., 2021) and distribution shifts (Kübler et al., 2022; Mukherjee & Sriperumbudur, 2025). However, these approaches are restricted to non-causal settings: Kübler et al. (2022) use a two-stage (train/test split) procedure to construct a precision-weighted witness, while Mukherjee & Sriperumbudur (2025) use random features to achieve minimax optimality in the standard two-sample problem. To the best of our knowledge, such Wald-type discrepancies have not been extended to the conditional distributional causal setting

## B. Extended Problem Setup

Let $\mathcal{P}$ be the statistical model, a collection of distributions on a space $\mathcal{Z}$. We assume $\mathcal{Z}$ is a Polish space defined as $\mathcal{Z} := \mathcal{X} \times \mathcal{A} \times \mathcal{Y}$, where $\mathcal{A} := \{0, 1\}$, equipped with its Borel $\sigma$-algebra $\mathcal{B}_{\mathcal{Z}} \equiv \mathcal{B}_{\mathcal{X}} \otimes \mathcal{B}_{\mathcal{A}} \otimes \mathcal{B}_{\mathcal{Y}}$. We observe an i.i.d. sample

$$\mathscr{D} := \{Z_i\}_{i=1}^n, \quad Z_i := (X_i, A_i, Y_i) \sim P \in \mathcal{P},$$

where $X_i \in \mathcal{X}$ are covariates, $A_i \in \{0, 1\}$ is the treatment, and $Y_i \in \mathcal{Y}$ is the outcome. For a given $P \in \mathcal{P}$, we denote the marginal distribution of $X$ by $P_X$ and the conditional distribution of $Y$ given $(A, X)$ by $P_{Y \mid A, X}$. We assume $P_{Y \mid A, X}$ is non-degenerate. We denote the conditional probability mass function of the treatment $A$ given $X = x$ by $g_P(\cdot \mid x)$, and define the propensity score as $\pi_P(x) := g_P(1 \mid x) = P(A = 1 \mid X = x)$.

We assume the model $\mathcal{P}$ is dominated by a $\sigma$-finite measure $\mu$. For each $P \in \mathcal{P}$, we let $L^2(P)$ denote the usual Hilbert space of $P$-square-integrable real-valued functions on $\mathcal{Z}$ with inner product $\langle h_1, h_2 \rangle_{L^2(P)} := \int h_1 h_2 dP$.

We now state a smoothness assumption required for the model to support semiparametric efficiency theory. A submodel $\{P_\epsilon \in \mathcal{P} : \epsilon > 0\}$ is called quadratic mean differentiable (QMD) at $P \in \mathcal{P}$ if there exists a score function $s \in L^2(P)$ such that $\mathbb{E}_P[s(Z)] = 0$ and

$$\left\| \sqrt{p_\epsilon} - \sqrt{p} - \frac{1}{2} \epsilon s \sqrt{p} \right\|_{L^2(\mu)} = o(\epsilon), \tag{19}$$

where $p_\epsilon := dP_\epsilon/d\mu$ and $p := dP/d\mu$. The set of all such scores $s$, taken over all possible QMD submodels at $P$, forms the tangent set at $P$. Its closed linear span is the tangent space.

Finally, we assume that $\mathcal{P}$ is locally nonparametric. Specifically, for each $P \in \mathcal{P}$, that means the tangent space is the entire

set of centered square-integrable functions: $L_0^2(P) := \{h \in L^2(P) : \mathbb{E}_P[h(Z)] = 0\}$. Throughout, we assume that the tangent set is equal to the tangent space.

**Causal identification assumptions.** The conditional mean embedding $\mu_{P_{Y(a)} \mid X}$ is identified with $\nu_{P,a}$ as defined in (3) under the following standard assumptions (Stone, 1993; Mealli & Rubin, 2003):

1. *Consistency:* $Y = AY(1) + (1 - A)Y(0)$.
2. *Unconfoundedness:* $Y(a) \perp\!\!\!\perp A \mid X$ for $a \in \{0, 1\}$.
3. *Overlap:* $0 < \pi_P(x) < 1$ for all $x \in \mathcal{X}$.

In addition to the identification conditions above, we impose *strong positivity*: the propensity scores $\pi_P$ are $P_X$-a.e. bounded away from 0 and 1 uniformly over $P \in \mathcal{P}$. Specifically, there exists $\eta > 0$ such that for all $P \in \mathcal{P}$, $\eta \leq \pi_P(x) \leq 1 - \eta$ for $P_X$-almost all $x$.

**RKHS structure.** We utilize the following RKHSs:

- $\mathcal{H}_{\mathcal{Y}}$: Associated with a bounded characteristic kernel $\ell : \mathcal{Y} \times \mathcal{Y} \to \mathbb{R}$ and feature map $L_y := \ell(y, \cdot)$.

- $\mathcal{H}_{\mathcal{X}}$: Associated with a bounded characteristic kernel $k : \mathcal{X} \times \mathcal{X} \to \mathbb{R}$ and feature map $K_x := k(x, \cdot)$.

- $\mathcal{H}$: The tensor product RKHS $\mathcal{H} := \mathcal{H}_{\mathcal{X}} \otimes \mathcal{H}_{\mathcal{Y}}$ (Park & Muandet, 2020). This space is associated with the product kernel $\lambda((x, y), (x', y')) := k(x, x')\ell(y, y')$ and has the feature map $\Lambda_{x,y} = K_x \otimes L_y$.

Since $k$ and $\ell$ are bounded and characteristic, the product kernel $\lambda$ is also bounded and characteristic. We assume throughout that $\mathcal{H}$ is real and separable.

**Notational remark for proofs.** We will frequently omit the tensor product notation $\otimes$ to declutter math displays. For instance, we will often write the feature map $\Lambda_{x,y} = K_x \otimes L_y$ as $K_x L_y$. Similarly, the kernel $\lambda((x, y), (x', y'))$ may appear as $k(x, x')\ell(y, y')$. When manipulating terms involving our one-step estimator, we will often rely on the bilinearity of the tensor product to factorize expressions, and will then omit $\otimes$. For example, the term $\frac{a}{\pi_P(x)}(K_x \otimes L_y - K_x \otimes \nu_{P,a}(x))$ may be written more compactly as $\frac{a}{\pi_P(x)} K_x (L_y - \nu_{P,a}(x))$. This should be interpreted strictly as the tensor product of the element $K_x \in \mathcal{H}_{\mathcal{X}}$ with the element $(L_y - \nu_{P,a}(x)) \in \mathcal{H}_{\mathcal{Y}}$.

## C. Formulating the Estimand

In this appendix, we justify the formulation of our estimand, the SCoDiTE, by showing that testing for conditional distributional invariance is equivalent to testing for the equality of joint distributions $P_{Y(a),X}$ for $a \in \{0, 1\}$.

### C.1. Proof of Proposition 2.1

In the potential outcomes framework, $X$ is *pre*-treatment. This is key for the following proof.

**Proposition 2.1** (Equivalent null). *For any $P \in \mathcal{P}$, $H_0$ holds if and only if $P_{Y(1),X} = P_{Y(0),X}$.*

*Proof.* Suppose $H_0$ holds. For all $a \in \{0, 1\}$ and Borel-measurable $B \subseteq \mathcal{Y}$ and $C \subseteq \mathcal{X}$, it holds that

$$P_{Y(a),X}(B \times C) = \int_C \int_B P_{Y(a) \mid X}(dy \mid x) P_X(dx).$$

Note that the disintegration theorem guarantees the uniqueness of conditional distributions in every Polish space equipped with its Borel $\sigma$-algebra. Now, if $P_{Y(1) \mid X}(\cdot \mid x) = P_{Y(0) \mid X}(\cdot \mid x)$ $P_X$-a.e., then direct substitution of the above yields

$$P_{Y(1),X}(B \times C) = \int_C \int_B P_{Y(1) \mid X}(dy \mid x) P_X(dx) = \int_C \int_B P_{Y(0) \mid X}(dy \mid x) P_X(dx) = P_{Y(0),X}(B \times C).$$

Since this holds for all measurable Cartesian products $B \times C$, which form a $\pi$-system that generates the product $\sigma$-algebra, we conclude by the $\pi$-$\lambda$ theorem that $P_{Y(1),X} = P_{Y(0),X}$.

We now establish the other direction. Suppose $P_{Y(1),X} = P_{Y(0),X}$. For any Borel-measurable sets $B \subseteq \mathcal{Y}$ and $C \subseteq \mathcal{X}$, $P_{Y(1),X} = P_{Y(0),X}$ and the law of total expectation together yield

$$0 = P_{Y(1),X}(B \times C) - P_{Y(0),X}(B \times C) = \int_C \left[ P_{Y(1)\,|\,X}(B\,|\,x) - P_{Y(0)\,|\,X}(B\,|\,x) \right] P_X(dx).$$

Since this must hold for any measurable set $C \subseteq \mathcal{X}$, we have that $P_{Y(1)\,|\,X}(B\,|\,x) = P_{Y(0)\,|\,X}(B\,|\,x)$ $P_X$-almost all $x$. Thus, for any fixed set $B$, there exists a null set $N_B \subseteq \mathcal{X}$ such that:

$$P_{Y(1)\,|\,X}(B\,|\,x) = P_{Y(0)\,|\,X}(B\,|\,x) \text{ for all } x \notin N_B.$$

Let $\mathcal{G}$ be a countable $\pi$-system that generates the Borel $\sigma$-algebra on $\mathcal{Y}$. Let $N = \bigcup_{B' \in \mathcal{G}} N_{B'}$, and note that as a countable union of null sets, $N$ is also a null set. Now, for any $x \notin N$, it holds that $P_{Y(1)\,|\,X}(B'\,|\,x) = P_{Y(0)\,|\,X}(B'\,|\,x)$ for all $B' \in \mathcal{G}$. Thus, we conclude that $P_{Y(1)\,|\,X}(\cdot\,|\,x) = P_{Y(0)\,|\,X}(\cdot\,|\,x)$ for $P_X$-almost all $x$ by appealing again to the $\pi$-$\lambda$ theorem. $\qquad\square$

## D. Derivation of the EIF

We use a one-step estimator of SCoDiTE, based on its nonparametric EIF. Here we establish the existence and functional form of that object.

### D.1. Pathwise Differentiability of $\psi_P$

The RKHS-valued SCoDiTE parameter $P \mapsto \psi_P$ (6) will have an EIF if it is pathwise differentiable and a moment condition is satisfied. We begin by establishing pathwise differentaibility. Let $\{P_\epsilon : \epsilon \in [0, \delta)\} \subset \mathcal{P}$ be a QMD submodel passing through $P \in \mathcal{P}$ at $\epsilon = 0$ with score function $s \in L_0^2(P)$. Let $\mathscr{P}(P, \mathcal{P}, s)$ be the set of all such submodels. A parameter $\psi : \mathcal{P} \to \mathcal{H}$ is pathwise differentiable at $P$ relative to the locally nonparametric model $\mathcal{P}$ if and only if there exists a continuous linear operator $\dot{\psi}_P : L_0^2(P) \to \mathcal{H}$ such that for all $s \in L_0^2(P)$ and every $\{P_\epsilon\} \in \mathscr{P}(P, \mathcal{P}, s)$,

$$\left\| \psi_{P_\epsilon} - \psi_P - \epsilon \dot{\psi}_P(s) \right\|_{\mathcal{H}} = o(\epsilon). \tag{20}$$

The operator $\dot{\psi}_P$ is referred to as the local parameter or pathwise derivative of $\psi$ at $P$.

Let $\psi_{P,a} := \mathbb{E}_P \mathbb{E}_P[\Lambda_{X,Y}\,|\,A = a, X]$. Then, by linearity of expectation, our estimand $\psi_P$ (6) decomposes as

$$\psi_P = \psi_{P,1} - \psi_{P,0}. \tag{21}$$

To establish the pathwise differentiability of $\psi_P$, we can first establish it for $\psi_{P,1}$, appeal to symmetry of the binary treatment, and then use the triangle inequality to conclude the argument. To this end, we leverage an existing result for the counterfactual kernel mean embedding (CKME) of a generic distribution $Q$ on $\mathcal{X} \times \{0,1\} \times \mathcal{W}$; in our subsequent arguments, $Q$ will be the distribution of $(X, A, W := (X, Y))$ under sampling $(X, A, Y) \sim P$.

**Lemma D.1** (Pathwise differentiability of the CKME). *Let $\mathcal{Q}$ be a locally nonparametric model comprising distributions on $\mathcal{Z} := \mathcal{X} \times \{0,1\} \times \mathcal{W}$ satisfying strong positivity, where $\mathcal{W}$ is a Polish space equipped with a bounded characteristic kernel $k_{\mathcal{W}}$ and associated RKHS $\mathcal{H}_{\mathcal{W}}$ with feature map $\Phi_w := k_{\mathcal{W}}(w, \cdot)$.*

*The parameter $\mu_a : \mathcal{Q} \to \mathcal{H}_{\mathcal{W}}$ defined by $\mu_a(Q) := \mathbb{E}_Q[\mathbb{E}_Q[\Phi_W\,|\,A = a, X]]$ is pathwise differentiable at any $Q \in \mathcal{Q}$. Its local parameter at score $s \in L_0^2(Q)$ is given by*

$$\dot{\mu}_{Q,a}(s) := \iint \Phi_w \left( s_{W\,|\,A,X}(w\,|\,a,x) + s_X(x) \right) Q_{W\,|\,A,X}(dw\,|\,a,x) Q_X(dx), \tag{22}$$

*where $s_{W\,|\,A,X}$ and $s_X$ are the conditional and marginal score components defined as $s_{W\,|\,A,X}(w|a,x) := s(z) - \mathbb{E}_Q[s(Z)\,|\,A = a, X = x]$ and $s_X(x) := \mathbb{E}_Q[s(Z)\,|\,X = x]$.*

*Proof.* See Appendix B.4.1 of Luedtke & Chung (2024), specifically the derivation of Eq. 19 and the verification of conditions for Lemma 2 therein. Their proof ultimately relies on the boundedness of the kernel and strong positivity, which are both satisfied here. $\qquad\square$

Next, we establish that quadratic mean differentiability (QMD) and pathwise differentiability are preserved when pushed forward through an injective map.

**Lemma D.2** (Invariance under injective pushforwards). *Let $(\mathcal{Z}, \mathcal{B}_{\mathcal{Z}})$ and $(\tilde{\mathcal{Z}}, \mathcal{B}_{\tilde{\mathcal{Z}}})$ be Polish spaces. Let $T : \mathcal{Z} \to \tilde{\mathcal{Z}}$ be a measurable injection such that $T^{-1}$ is measurable on the range $T(\mathcal{Z})$. For a locally nonparametric model $\mathcal{P}$ on $\mathcal{Z}$, define the induced model $\mathcal{Q} := \{P \circ T^{-1} : P \in \mathcal{P}\}$ on $\tilde{\mathcal{Z}}$, noting that each $Q \in \mathcal{Q}$ is supported on $T(\mathcal{Z})$.*

(i) *If a submodel $\{P_\epsilon\} \subset \mathcal{P}$ is QMD at $P$ with score $s \in L_0^2(P)$, then the induced submodel $\{Q_\epsilon := P_\epsilon \circ T^{-1}\} \subset \mathcal{Q}$ is QMD at $Q := P \circ T^{-1}$ with score $\tilde{s} := s \circ T^{-1} \in L_0^2(Q)$.*

(ii) *Let $\mathcal{H}$ be the action space and let $\tilde{\psi} : \mathcal{Q} \to \mathcal{H}$ be a parameter. Define $\psi : \mathcal{P} \to \mathcal{H}$ by $\psi(P) := \tilde{\psi}(P \circ T^{-1})$. If $\tilde{\psi}$ is pathwise differentiable at $Q$ with local parameter $\dot{\tilde{\psi}}_Q$, then $\psi$ is pathwise differentiable at $P$ with local parameter $\dot{\psi}_P(s) := \dot{\tilde{\psi}}_{P \circ T^{-1}}(s \circ T^{-1})$.*

*Proof.* Let $\mu$ be a $\sigma$-finite measure dominating the model $\mathcal{P}$. Define the pushforward measure on $\tilde{\mathcal{Z}}$ by $\tilde{\mu} := \mu \circ T^{-1}$. We claim that $\tilde{\mu}$ dominates $\mathcal{Q}$. To see why, note that for any $Q = P \circ T^{-1} \in \mathcal{Q}$, if $\tilde{\mu}(C) = 0$, then $\mu(T^{-1}(C)) = 0$, which implies $P(T^{-1}(C)) = 0$, and thus $Q(C) = 0$.

*Statement (i)*: Let $p_\epsilon = dP_\epsilon/d\mu$ and $q_\epsilon = dQ_\epsilon/d\tilde{\mu}$. We first establish the following pointwise relationship between these densities: $q_\epsilon(T(z)) = p_\epsilon(z)$ for $\mu$-a.e. $z$ and $q_\epsilon(t) = p_\epsilon(T^{-1}(t))$ for $\tilde{\mu}$-a.e. $t$. Indeed, for any measurable set $C \in \mathcal{B}_{\tilde{\mathcal{Z}}}$, the change of variables theorem for integrals yields

$$\int_{T^{-1}(C)} q_\epsilon(T(z))d\mu(z) = \int_C q_\epsilon(t)d\tilde{\mu}(t) = Q_\epsilon(C) = P_\epsilon(T^{-1}(C)) = \int_{T^{-1}(C)} p_\epsilon(z)d\mu(z),$$

establishing the desired pointwise relationships.

Now, we examine the quadratic mean differentiability of $Q_\epsilon$ at $Q = P \circ T^{-1}$ using the candidate score $\tilde{s} = s \circ T^{-1}$. Observe that

$$\left\| \sqrt{q_\epsilon} - \sqrt{q} - \frac{\epsilon}{2}\tilde{s}\sqrt{q} \right\|_{L^2(\tilde{\mu})}^2 = \int_{\tilde{\mathcal{Z}}} \left( \sqrt{q_\epsilon(t)} - \sqrt{q(t)} - \frac{\epsilon}{2}\tilde{s}(t)\sqrt{q(t)} \right)^2 d\tilde{\mu}(t).$$

With the change of variables $t = T(z)$, the above display becomes

$$= \int_{\mathcal{Z}} \left( \sqrt{q_\epsilon(T(z))} - \sqrt{q(T(z))} - \frac{\epsilon}{2}\tilde{s}(T(z))\sqrt{q(T(z))} \right)^2 d\mu(z)$$

$$= \int_{\mathcal{Z}} \left( \sqrt{p_\epsilon(z)} - \sqrt{p(z)} - \frac{\epsilon}{2}s(z)\sqrt{p(z)} \right)^2 d\mu(z)$$

$$= \left\| \sqrt{p_\epsilon} - \sqrt{p} - \frac{\epsilon}{2}s\sqrt{p} \right\|_{L^2(\mu)}^2 = o(\epsilon^2),$$

where the last equality holds by the quadratic mean differentiability of $P_\epsilon$ at $P$ with score $s$. This establishes QMD with score $\tilde{s}$ provided $\tilde{s} \in L_0^2(Q)$. This indeed holds since $s \in L_0^2(P)$ yields that

$$\int \tilde{s}dQ = \int (s \circ T^{-1})d(P \circ T^{-1}) = \int sdP = 0,$$

and similarly $\|\tilde{s}\|_{L^2(Q)}^2 = \|s\|_{L^2(P)}^2$.

*Statement (ii)*: Assume $\tilde{\psi}$ is pathwise differentiable at $Q$. Then, by definition, there exists a continuous linear map $\dot{\tilde{\psi}}_Q : L_0^2(Q) \to \mathcal{H}$ such that for any QMD submodel $\{Q_\epsilon\} \subset \mathcal{Q}$ with score $\tilde{s} \in L_0^2(Q)$, we have:

$$\left\| \tilde{\psi}(Q_\epsilon) - \tilde{\psi}(Q) - \epsilon\dot{\tilde{\psi}}_Q(\tilde{s}) \right\|_{\mathcal{H}} = o(\epsilon).$$

Now, consider an arbitrary submodel $\{P_\epsilon\} \subset \mathcal{P}$ that is QMD at $P$ with score $s \in L_0^2(P)$. From Part (i), the induced submodel $\{Q_\epsilon := P_\epsilon \circ T^{-1}\}$ is QMD at $Q$ with score $\tilde{s} = s \circ T^{-1} \in L_0^2(Q)$. By definition,

$$\left\| \tilde{\psi}(Q_\epsilon) - \tilde{\psi}(Q) - \epsilon\dot{\tilde{\psi}}_Q(\tilde{s}) \right\|_{\mathcal{H}} = \left\| \tilde{\psi}(Q_\epsilon) - \tilde{\psi}(Q) - \epsilon\dot{\tilde{\psi}}_Q(s \circ T^{-1}) \right\|_{\mathcal{H}} = o(\epsilon).$$

Recognizing that $\psi(P) = \tilde{\psi}(P \circ T^{-1})$ yields

$$\left\| \psi\left(P_\epsilon\right) - \psi(P) - \epsilon \dot{\tilde{\psi}}_Q(\tilde{s}) \right\|_{\mathcal{H}} = o(\epsilon).$$

Hence, we will have established pathwise differentiability of $\psi$ with local parameter $\dot{\psi}_P := \eta_P$ for $\eta_P(s) := \dot{\tilde{\psi}}_{P \circ T^{-1}}(s \circ T^{-1})$ provided we can show that $\eta_P$ is bounded and linear.

Linearity follows from the fact that $\eta_P$ is a composition of the linear map $\dot{\tilde{\psi}}_{P \circ T^{-1}}$ and the composition operator. For $\| \cdot \|_{\mathrm{op}}$ the usual operator norm, boundedness follows by the fact that, for any $s$ with $\|s\|_{L^2(P)} \leq 1$,

$$\|\eta_P(s)\|_{\mathcal{H}} = \left\| \dot{\tilde{\psi}}_Q(s \circ T^{-1}) \right\|_{\mathcal{H}} \leq \left\| \dot{\tilde{\psi}}_Q \right\|_{\mathrm{op}} \|s \circ T^{-1}\|_{L^2(Q)}$$

$$= \left\| \dot{\tilde{\psi}}_Q \right\|_{\mathrm{op}} \left[ \int \left(s \circ T^{-1}\right)^2 dQ \right]^{1/2} = \left\| \dot{\tilde{\psi}}_Q \right\|_{\mathrm{op}} \|s\|_{L^2(P)} \leq \left\| \dot{\tilde{\psi}}_Q \right\|_{\mathrm{op}},$$

where the right-hand side does not depend on $s$ and is finite since $\dot{\tilde{\psi}}_Q$ is the local parameter of $\tilde{\psi}$. $\qquad \square$

We now establish the pathwise differentiability of SCoDiTE by identifying it as a linear combination (with respect to $a \in \{0, 1\}$) of CKMEs (21) on a reparameterized outcome space.

**Proposition D.3** (Pathwise differentiability of the SCoDiTE). *$\psi$ is pathwise differentiable relative to the locally nonparametric model $\mathcal{P}$. For an arbitrary score $s \in L_0^2(P)$, the local parameter $\dot{\psi}_P$ takes the form*

$$\dot{\psi}_P(s)(\cdot) := \iint \Lambda_{x,y}(\cdot) \left[ s_{Y \mid A,X}(y \mid 1, x) + s_X(x) \right] P_{Y \mid A,X}(dy \mid 1, x) P_X(dx) \tag{23}$$

$$- \iint \Lambda_{x,y}(\cdot) \left[ s_{Y \mid A,X}(y \mid 0, x) + s_X(x) \right] P_{Y \mid A,X}(dy \mid 0, x) P_X(dx),$$

*where $s_{Y \mid A,X}(y|a, x) := s(z) - \mathbb{E}_P[s(Z) \mid A = a, X = x]$ and $s_X(x) := \mathbb{E}_P[s(Z) \mid X = x]$.*

*Proof.* From the problem setup, we know that $Z = (X, A, Y)$ takes values in the Polish space $(\mathcal{Z}, \mathcal{B}_{\mathcal{Z}}) \equiv (\mathcal{X} \times \mathcal{A} \times \mathcal{Y}, \mathcal{B}_{\mathcal{X}} \otimes \mathcal{B}_{\mathcal{A}} \otimes \mathcal{B}_{\mathcal{Y}})$ for each probability measure $P \in \mathcal{P}$. Define the reparameterization map as the measurable embedding $g : \mathcal{X} \times \mathcal{A} \times \mathcal{Y} \to \mathcal{X} \times \mathcal{A} \times (\mathcal{X} \times \mathcal{Y})$ given by $g(x, a, y) := (x, a, (x, y))$.

For each $P \in \mathcal{P}$, let the pushforward of $P$ by $g$ be the measure $Q$ on the space $(\mathcal{X} \times \mathcal{A} \times (\mathcal{X} \times \mathcal{Y}), \mathcal{B}_{\mathcal{X}} \otimes \mathcal{B}_{\mathcal{A}} \otimes (\mathcal{B}_{\mathcal{X}} \otimes \mathcal{B}_{\mathcal{Y}}))$, given by

$$Q(B) := P(g^{-1}(B)) \text{ for all measurable sets } B.$$

Let $\mathcal{Q} := \{P \circ g^{-1} : P \in \mathcal{P}\}$ be the collection of these pushforward measures. Note that every $Q \in \mathcal{Q}$ is a singular measure on the product space, supported entirely on the set $\{(x, a, (x', y)) : x = x'\}$. Consider an arbitrary measure $Q \in \mathcal{Q}$. By the disintegration theorem, $Q$ can be characterized by its conditional and marginal distributions. Crucially, $Q$ is a strict reparameterization of $P$ in the sense that its components satisfy:

1. $Q_{X,Y \mid A,X}(dx', dy \mid a, x) = \delta_x(dx') \times P_{Y \mid A,X}(dy \mid a, x)$ for $P$-almost all $(a, x)$.
2. $Q_{A \mid X}(\cdot \mid x) = P_{A \mid X}(\cdot \mid x)$ for $P_X$-almost all $x$.
3. $Q_X = P_X$.

Now, define a parameter $\tilde{\psi}_1 : \mathcal{Q} \to \mathcal{H}_{\mathcal{X}} \otimes \mathcal{H}_{\mathcal{Y}}$ such that $\tilde{\psi}_1(Q) := \mathbb{E}_Q[\mathbb{E}_Q[\Lambda_{X,Y} \mid A = 1, X]]$, which simplifies as follows:

$$\tilde{\psi}_1(Q) = \int \left( \int_{\mathcal{X} \times \mathcal{Y}} \Lambda_{x',y} Q_{X,Y \mid A,X}(d(x', y) \mid 1, x) \right) Q_X(dx)$$

$$= \int \left( \int_{\mathcal{X} \times \mathcal{Y}} \Lambda_{x',y}(\delta_x \times P_{Y \mid A,X})(d(x', y) \mid 1, x) \right) Q_X(dx)$$

$$= \int \left( \iint \Lambda_{x',y} \delta_x(dx') P_{Y \mid A,X}(dy \mid 1, x) \right) Q_X(dx)$$

$$= \iint \Lambda_{x,y} P_{Y \mid A,X}(dy \mid 1, x) P_X(dx) = \mathbb{E}_P[\theta_{P,1}(X)],$$

where $\theta_{P,1}(x) = \mathbb{E}_P[\Lambda_{x,Y} \mid A=1, X=x] = K_x \mathbb{E}_P[L_Y \mid A=1, X=x] = K_x \nu_{P,1}(x)$, matching its definition in (5). It is evident that $\tilde{\psi}_1(Q) = \tilde{\psi}_1(P \circ g^{-1}) = \psi_{P,1}$ from the decomposition in Eq. 21.

Although $Q_{W \mid A,X}(\cdot \mid a, x)$ is a.e. degenerate (supported only on the slice $\{x\} \times \mathcal{Y}$), the assumptions on $\mathcal{P}$ ensure the induced model $\mathcal{Q}$ satisfies the conditions of Lemma D.1. Further, we know that $\tilde{\psi}_1$ is precisely the CKME parameter with outcome space $\mathcal{W} := \mathcal{X} \times \mathcal{Y}$ and associated RKHS $\mathcal{H}_\mathcal{W} = \mathcal{H}_\mathcal{X} \otimes \mathcal{H}_\mathcal{Y}$ with feature map $\Phi_{(x,y)} \equiv \Lambda_{x,y}$. Thus, by Lemma D.1, $\tilde{\psi}_1$ is pathwise differentiable at $Q$, with local parameter $\dot{\tilde{\psi}}_{Q,1}(\tilde{s})$ for score $\tilde{s} \in L_0^2(Q)$ given by

$$\dot{\tilde{\psi}}_{Q,1}(\tilde{s}) = \iiint \Lambda_{x,y} \left[ \tilde{s}_{X,Y \mid A,X}(x,y \mid 1, x) + \tilde{s}_X(x) \right] Q_{X,Y \mid A,X}(dx', dy \mid 1, x) Q_X(dx),$$

where $\tilde{s}_{X,Y \mid A,X}(x,y|a,x) := \tilde{s}(\tilde{z}) - \mathbb{E}_Q[\tilde{s}(\tilde{Z}) \mid A=a, X=x]$ and $\tilde{s}_X(x) := \mathbb{E}_Q[\tilde{s}(\tilde{Z}) \mid X=x]$.

Consequently, Lemma D.2 yields that $\psi_{P,1}$ is pathwise differentiable at $P = Q \circ g$ with local parameter $\dot{\psi}_{P,1}(s) = \dot{\tilde{\psi}}_{Q,1}(s \circ g^{-1})$. Plugging in $\tilde{s} = s \circ g^{-1}$ and $\tilde{z} = g(z)$ yields that for any score $s \circ g^{-1} \in L_0^2(Q)$ and corresponding $s \in L_0^2(P)$,

$$\tilde{s}_{X,Y \mid A,X}(x,y \mid a, x) = \tilde{s}(g(z)) - \int \tilde{s}(\tilde{z}) [P \circ g^{-1}]_{X,Y \mid A,X}(dx', dy \mid a, x)$$

$$= s \circ g^{-1}(g(z)) - \int (\tilde{s} \circ g)(z) P_{Y \mid A,X}(dy \mid a, x)$$

$$= s(z) - \int s(z) P_{Y \mid A,X}(dy \mid a, x) = s_{Y \mid A,X}(y \mid a, x),$$

and similarly $\tilde{s}_X(x) = \mathbb{E}_P[s(Z) \mid X=x] = s_X(x)$. It follows that

$$\dot{\psi}_{P,1}(s) = \dot{\tilde{\psi}}_{Q,1}(\tilde{s})$$

$$= \iiint \Lambda_{x,y} \left[ s_{Y \mid A,X}(y \mid 1, x) + s_X(x) \right] \left( \delta_x(dx') \times P_{Y \mid A,X}(dy \mid 1, x) \right) P_X(dx)$$

$$= \iint K_x L_y \left[ s_{Y \mid A,X}(y \mid 1, x) + s_X(x) \right] P_{Y \mid A,X}(dy \mid 1, x) P_X(dx).$$

An analogous argument holds for $\psi_{P,0}$, showing that, for score $s \in L_0^2(P)$,

$$\dot{\psi}_{P,0}(s) := \iint K_x L_y \left[ s_{Y \mid A,X}(y \mid 0, x) + s_X(x) \right] P_{Y \mid A,X}(dy \mid 0, x) P_X(dx).$$

By the triangle inequality, the fact that $\psi_P = \psi_{P,1} - \psi_{P,0}$ shows that $P \mapsto \psi_P$ is pathwise differentiable with local parameter $\dot{\psi}_P = \dot{\psi}_{P,1} - \dot{\psi}_{P,0}$. Since $Q \in \mathcal{Q}$ (and thus $P \in \mathcal{P}$) was arbitrary, we have that $\psi$ is pathwise differentiable at each $P \in \mathcal{P}$. $\qquad \square$

### D.2. Proof of Lemma 2.2

To derive the form of the EIF of our parameter, we first introduce the efficient influence operator (EIO). Let $\dot{\psi}_P : L_0^2(P) \to \mathcal{H}$ be the local parameter. Note that its image is a closed subspace of $\mathcal{H}$, denoted by $\dot{\mathcal{H}}_P$ and referred to as the local parameter space. As $\mathcal{H}$ is a real separable RKHS in our setting, $\dot{\mathcal{H}}_P$ inherits this structure. The efficient influence operator is the adjoint of the local parameter, $\dot{\psi}_P^* : \mathcal{H} \to L_0^2(P)$, i.e. the continuous linear operator uniquely defined by the duality condition:

$$\left\langle h, \dot{\psi}_P(s) \right\rangle_\mathcal{H} = \left\langle \dot{\psi}_P^*(h), s \right\rangle_{L^2(P)} \qquad \text{for all } h \in \mathcal{H} \text{ and } s \in L_0^2(P). \tag{24}$$

Unlike finite-dimensional calculus where gradients are vectors, here the EIF $\phi_P$ is an $\mathcal{H}$-valued random variable. As detailed in Theorem 1 of Luedtke & Chung (2024), the EIF can be constructed via the Riesz representation of the EIO applied to the RKHS feature map. We use this to prove that our proposed form for the EIF of $\psi_P$ is correct.

**Lemma 2.2** (Existence and form of the EIF). *The parameter $\psi$ defined as in Eq. 6 is pathwise differentiable at every $P \in \mathcal{P}$, and has an EIF at each $P$ that takes the form*

$$\phi_P(x, a, y) = \left( \frac{a}{\pi_P(x)} - \frac{1 - a}{1 - \pi_P(x)} \right) (\Lambda_{x,y} - \theta_{P,a}(x))$$
$$+ \theta_{P,1}(x) - \theta_{P,0}(x) - \psi_P.$$

*Moreover, $0 < \int \|\phi_P(z)\|_{\mathcal{H}}^2 P(dz) < \infty$ for all $P \in \mathcal{P}$.*

*Proof. Part 1: deriving the EIO, $\dot{\psi}_P^*$:* Fix any $P \in \mathcal{P}$ and $s \in L_0^2(P)$, and let $s_{Y \mid A,X}(y \mid a, x)$ and $s_X(x)$ be as defined in Proposition D.3. Recall the definition of $\psi_{P,1}$ from (21), and recall from Proposition D.3 that we have the corresponding local parameter as follows:

$$\dot{\psi}_{P,1}(s) = \underbrace{\iint \Lambda_{x,y} s_{Y \mid A,X}(y \mid 1, x) P_{Y \mid A,X}(dy \mid 1, x) P_X(dx)}_{\text{I}} + \underbrace{\iint \Lambda_{x,y} s_X(x) P_{Y \mid A,X}(dy \mid 1, x) P_X(dx)}_{\text{II}}.$$

By the law of total expectation, Term I rewrites as

$$\iint \Lambda_{x,y} s_{Y \mid A,X}(y \mid 1, x) P_{Y \mid A,X}(dy \mid 1, x) P_X(dx)$$
$$= \int \frac{1}{g_P(1 \mid x)} \int a \int \Lambda_{x,y} \left\{ s(x, a, y) - \mathbb{E}_P \left[ s(x, a, Y) \mid A = a, X = x \right] \right\} P_{Y \mid A,X}(dy \mid a, x) g_P(a \mid x) P_X(dx).$$

We distribute the integral and recognize that $P_{Y \mid A,X}(dy \mid a, x) g_P(a \mid x) P_X(dx) = P(dz)$ is the joint distribution, so that

$$\text{I} = \int \frac{a}{g_P(1 \mid x)} K_x L_y s(z) P(dz)$$
$$- \iint \frac{a}{g_P(1 \mid x)} K_x \left( \int L_y P_{Y \mid A,X}(dy \mid a, x) \right) \left( \int s(x, a, y) P_{Y \mid A,X}(dy \mid a, x) \right) g_P(a \mid x) P_X(dx)$$

Applying the law of total expectation (conditioning on $A, X$) to the second term and recognizing the inner integral as the conditional expectation $\mathbb{E}_P[L_Y \mid A = a, X = x]$ yields

$$\text{I} = \int \frac{a}{\pi_P(x)} K_x L_y s(z) P(dz) - \int \frac{a}{\pi_P(x)} K_x \mathbb{E}_P[L_Y \mid A = a, X = x] s(z) P(dz)$$
$$= \int \frac{a}{g_P(1 \mid x)} \left\{ K_x L_y - K_x \mathbb{E}_P[L_Y \mid A = a, X = x] \right\} s(z) P(dz).$$

Next, we rewrite Term II as

$$\iint \Lambda_{x,y} s_X(x) P_{Y \mid A,X}(dy \mid 1, x) P_X(dx)$$
$$= \iint \Lambda_{x,y} P_{Y \mid A,X}(dy \mid 1, x) \mathbb{E}_P[s(x, A, Y) \mid X = x] P_X(dx)$$
$$= \int K_x \left( \int L_y P_{Y \mid A,X}(dy \mid 1, x) \right) \left( \iint s(x, a, y) P_{Y \mid A,X}(dy \mid a, x) g_P(a \mid x) \right) P_X(dx).$$

The first parenthesis is $\mathbb{E}_P[L_Y \mid A = 1, X = x]$ and the second parenthesis is $\mathbb{E}_P[s(Z) \mid X = x]$. Recall that $\int s(z) P(dz) = 0$ by definition. Thus, applying the law of total expectation (conditioning on $X$) to the above display and subtracting zero from it yields

$$\text{II} = \int K_x \mathbb{E}_P[L_Y \mid A = 1, X = x] s(z) P(dz) - \mathbb{E}_P \mathbb{E}_P[K_X L_Y \mid A = 1, X] \int s(z) P(dz)$$
$$= \int (K_x \mathbb{E}_P[L_Y \mid A = 1, X = x] - \mathbb{E}_P \mathbb{E}_P[\Lambda_{X,Y} \mid A = 1, X]) s(z) P(dz),$$

Combining terms I and II yields

$$\dot{\psi}_{P,1}(s) = \int \frac{a}{g_P(1\,|\,x)} \left\{ K_x L_y - K_x \mathbb{E}_P\left[L_Y\,|\,A = a, X = x\right] \right\} s(z)P(dz)$$
$$+ \int \left( K_x \mathbb{E}_P[L_Y\,|\,A = 1, X = x] - \mathbb{E}_P \mathbb{E}_P[\Lambda_{X,Y}\,|\,A = 1, X] \right) s(z)P(dz).$$

By an analogous argument,

$$\dot{\psi}_{P,0}(s) = \int \frac{1-a}{g_P(0\,|\,x)} \left\{ K_x L_y - K_x \mathbb{E}_P\left[L_Y\,|\,A = a, X = x\right] \right\} s(z)P(dz)$$
$$+ \int \left( K_x \mathbb{E}_P[L_Y\,|\,A = 0, X = x] - \mathbb{E}_P \mathbb{E}_P[\Lambda_{X,Y}\,|\,A = 0, X] \right) s(z)P(dz).$$

Therefore (recalling that $g_P(1\,|\,x) = \pi_P(x)$),

$$\dot{\psi}_P(s) = \dot{\psi}_{P,1}(s) - \dot{\psi}_{P,0}(s)$$
$$= \int \frac{a}{\pi_P(x)} \left\{ K_x L_y - K_x \mathbb{E}_P\left[L_Y\,|\,A = a, X = x\right] \right\} s(z)P(dz)$$
$$+ \int \left( K_x \mathbb{E}_P[L_Y\,|\,A = 1, X = x] - \mathbb{E}_P \mathbb{E}_P[\Lambda_{X,Y}\,|\,A = 1, X] \right) s(z)P(dz)$$
$$- \int \frac{1-a}{1-\pi_P(x)} \left\{ K_x L_y - K_x \mathbb{E}_P\left[L_Y\,|\,A = a, X = x\right] \right\} s(z)P(dz)$$
$$- \int \left( K_x \mathbb{E}_P[L_Y\,|\,A = 0, X = x] - \mathbb{E}_P \mathbb{E}_P[\Lambda_{X,Y}\,|\,A = 0, X] \right) s(z)P(dz)$$
$$= \int \left\{ \frac{a}{\pi_P(x)} - \frac{1-a}{1-\pi_P(x)} \right\} \left\{ K_x L_y - K_x \mathbb{E}_P\left[L_Y\,|\,A = a, X = x\right] \right\} s(z)P(dz)$$
$$+ \int \left\{ K_x \mathbb{E}_P[L_Y\,|\,A = 1, X = x] - K_x \mathbb{E}_P[L_Y\,|\,A = 0, X = x] \right.$$
$$\left. - \left( \mathbb{E}_P \mathbb{E}_P[\Lambda_{X,Y}\,|\,A = 1, X] - \mathbb{E}_P \mathbb{E}_P[\Lambda_{X,Y}\,|\,A = 0, X] \right) \right\} s(z)P(dz).$$

Consequently, for any $s \in L_0^2(P)$ and $h \in \mathcal{H}$, we have that

$$\left\langle \dot{\psi}_P(s), h \right\rangle_{\mathcal{H}} = \int \left\{ \frac{a}{\pi_P(x)} - \frac{1-a}{1-\pi_P(x)} \right\} \left\{ h(x,y) - \mathbb{E}_P\left[h(x,Y)\,|\,A = a, X = x\right] \right\} s(z)P(dz)$$
$$+ \int \left\{ \mathbb{E}_P[h(x,Y)\,|\,A = 1, X = x] - \mathbb{E}_P[h(x,Y)\,|\,A = 0, X = x] \right.$$
$$\left. - \left( \mathbb{E}_P \mathbb{E}_P[h(X,Y)\,|\,A = 1, X] - \mathbb{E}_P \mathbb{E}_P[h(X,Y)\,|\,A = 0, X] \right) \right\} s(z)P(dz).$$

The Hermitian adjoint $\dot{\psi}_P^*$ is identified from the integrand multiplying $s(z)$, and is given by

$$\dot{\psi}_P^*(h)(z) = \left\{ \frac{a}{\pi_P(x)} - \frac{1-a}{1-\pi_P(x)} \right\} \left\{ h(x,y) - \mathbb{E}_P\left[h(x,Y)\,|\,A = a, X = x\right] \right\}$$
$$+ \mathbb{E}_P[h(x,Y)\,|\,A = 1, X = x] - \mathbb{E}_P[h(x,Y)\,|\,A = 0, X = x]$$
$$- \mathbb{E}_P \mathbb{E}_P[h(X,Y)\,|\,A = 1, X] + \mathbb{E}_P \mathbb{E}_P[h(X,Y)\,|\,A = 0, X].$$

_Part 2: deriving the EIF, $\phi_P$_: Now, for each $(\tilde{x}, \tilde{y}) \in \mathcal{X} \times \mathcal{Y}$, define $\tilde{\phi}_P : \mathcal{Z} \to \mathcal{H}$ as $\tilde{\phi}_P(z)(\tilde{x}, \tilde{y}) := \dot{\psi}_P^*(\Lambda_{\tilde{x},\tilde{y}})(z)$ $P$-a.s.

$z$, which takes the form

$$\tilde{\phi}_P(z)(\tilde{x}, \tilde{y}) = \left\{ \frac{a}{\pi_P(x)} - \frac{1-a}{1-\pi_P(x)} \right\} \{ \Lambda_{\tilde{x},\tilde{y}}(x,y) - \mathbb{E}_P \left[ \Lambda_{\tilde{x},\tilde{y}}(x,Y) \,|\, A=a, X=x \right] \}$$
$$+ \mathbb{E}_P[\Lambda_{\tilde{x},\tilde{y}}(x,Y) \,|\, A=1, X=x] - \mathbb{E}_P[\Lambda_{\tilde{x},\tilde{y}}(x,Y) \,|\, A=0, X=x]$$
$$- \mathbb{E}_P\mathbb{E}_P[\Lambda_{\tilde{x},\tilde{y}}(X,Y) \,|\, A=1, X] + \mathbb{E}_P\mathbb{E}_P[\Lambda_{\tilde{x},\tilde{y}}(X,Y) \,|\, A=0, X]$$
$$= \left\{ \frac{a}{\pi_P(x)} - \frac{1-a}{1-\pi_P(x)} \right\} \{ \Lambda_{x,y}(\tilde{x},\tilde{y}) - \mathbb{E}_P \left[ \Lambda_{x,Y}(\tilde{x},\tilde{y}) \,|\, A=a, X=x \right] \}$$
$$+ \mathbb{E}_P[\Lambda_{x,Y}(\tilde{x},\tilde{y}) \,|\, A=1, X=x] - \mathbb{E}_P[\Lambda_{x,Y}(\tilde{x},\tilde{y}) \,|\, A=0, X=x]$$
$$- \left( \mathbb{E}_P\mathbb{E}_P[\Lambda_{X,Y}(\tilde{x},\tilde{y}) \,|\, A=1, X] - \mathbb{E}_P\mathbb{E}_P[\Lambda_{X,Y}(\tilde{x},\tilde{y}) \,|\, A=0, X] \right),$$

where the second equality holds by the symmetry of kernel functions $k$ and $\ell$. We then have by the definitions of $\theta_{P,a}$ (5) and $\psi(P)$ (6) respectively that

$$\tilde{\phi}_P(z) = \left\{ \frac{a}{\pi_P(x)} - \frac{1-a}{1-\pi_P(x)} \right\} \{ \Lambda_{x,y} - \theta_{P,a}(x) \} + \theta_{P,1}(x) - \theta_{P,0}(x) - \psi(P).$$

It follows that

$$\left\| \tilde{\phi}_P \right\|_{L^2(P;\mathcal{H})}^2 = \mathbb{E}_P \left\| \left\{ \frac{A}{\pi_P(X)} - \frac{1-A}{1-\pi_P(X)} \right\} \{ \Lambda_{X,Y} - \theta_{P,A}(X) \} + \theta_{P,1}(X) - \theta_{P,0}(X) - \psi(P) \right\|_{\mathcal{H}}^2$$

$$= \mathbb{E}_P \left\| \left\{ \frac{A}{\pi_P(X)} - \frac{1-A}{1-\pi_P(X)} \right\} \{ \Lambda_{X,Y} - \theta_{P,A}(X) \} \right\|_{\mathcal{H}}^2$$
$$+ \mathbb{E}_P \left\| \theta_{P,1}(X) - \theta_{P,0}(X) - \psi(P) \right\|_{\mathcal{H}}^2$$
$$- \mathbb{E}_P \left[ \frac{A}{\pi_P(X)} - \frac{1-A}{1-\pi_P(X)} \left\langle \mathbb{E}_P \left[ \Lambda_{X,Y} \,|\, A, X \right] - \theta_{P,A}(X), \theta_{P,1}(X) - \theta_{P,0}(X) - \psi(P) \right\rangle_{\mathcal{H}} \right]$$

via the law of total expectation (conditioning on $A, X$) applied to the cross term. Further, by (5), we have $\mathbb{E}_P[\Lambda_{X,Y} \,|\, A, X] - \theta_{P,A}(X) = 0$, so the cross-term vanishes. The display simplifies to

$$= \mathbb{E}_P \left\| \left\{ \frac{A}{\pi_P(X)} - \frac{1-A}{1-\pi_P(X)} \right\} \{ \Lambda_{X,Y} - \theta_{P,A}(X) \} \right\|_{\mathcal{H}}^2 + \mathbb{E}_P \left\| \theta_{P,1}(X) - \theta_{P,0}(X) - \psi(P) \right\|_{\mathcal{H}}^2,$$

where, using the non-negativity of the second term, we can lower bound the expression by

$$\geq \mathbb{E}_P \left[ \left| \frac{A}{\pi_P(X)} - \frac{1-A}{1-\pi_P(X)} \right|^2 \| \Lambda_{X,Y} - \theta_{P,A}(X) \|_{\mathcal{H}}^2 \right].$$

Applying the law of total expectation (conditioning on $A, X$) again, and noting that $A^2 = A$, $(1-A)^2 = (1-A)$, and $A(1-A) = 0$, yields

$$= \mathbb{E}_P \left[ \left( \frac{A}{\pi_P(X)^2} + \frac{1-A}{(1-\pi_P(X))^2} \right) \mathbb{E}_P \left[ \| \Lambda_{X,Y} - \theta_{P,A}(X) \|_{\mathcal{H}}^2 \,|\, A, X \right] \right],$$

which, upon using (5) followed by the law of total expectation (conditioning on $X$), simplifies to

$$= \mathbb{E}_P \left[ \mathbb{E}_P \left[ \left( \frac{A}{\pi_P(X)^2} + \frac{1-A}{(1-\pi_P(X))^2} \right) \mathit{Var}_P \left( \Lambda_{X,Y} \,|\, A, X \right) \,\Big|\, X \right] \right]$$
$$> 0,$$

where the strict inequality holds because the term in the parentheses is strictly positive by strong positivity, and the conditional variance is strictly positive since $P_{Y|A,X}$ is non-degenerate and the kernel $\ell$ is characteristic.

Next, the boundedness of $k$ and $\ell$ as well as strong positivity together imply that $\left\|\tilde{\phi}_P\right\|_{L^2(P;\mathcal{H})} < \infty$, i.e., that $\tilde{\phi}_P$ is $P$-Bochner square integrable. Now, Proposition D.3 and the fact that $\dot{\mathcal{H}}_P$ inherits the RKHS structure from $\mathcal{H}$ in our setting, together satisfy the conditions of Theorem 1 in Luedtke & Chung (2024), which yields that $\psi$ has an EIF $\phi_P$ at $P$, and that $\phi_P = \tilde{\phi}_P$ $P$-almost surely. Finally, since $P \in \mathcal{P}$ was arbitrary, we have the desired result. $\square$

With the explicit form of the EIF established, the following lemma verifies that it respects the additive structure of the parameter.

**Lemma D.4** (Decomposition of the EIF). *For any $P \in \mathcal{P}$, let $\phi_{P,1}$ and $\phi_{P,0}$ be defined as*

$$\phi_{P,1}(Z) := \frac{A}{\pi_P(X)} \left(\Lambda_{X,Y} - \theta_{P,1}(X)\right) + \theta_{P,1}(X) - \psi_{P,1},$$
$$\phi_{P,0}(Z) := \frac{1-A}{1-\pi_P(X)} \left(\Lambda_{X,Y} - \theta_{P,0}(X)\right) + \theta_{P,0}(X) - \psi_{P,0}. \tag{25}$$

*Then, $\phi_{P,1}$ and $\phi_{P,0}$ are the EIFs of $\psi_{P,1}$ and $\psi_{P,0}$ from the decomposition in (21) and the EIF $\phi_P$ derived in Lemma 2.2 satisfies the linear decomposition $\phi_P(Z) = \phi_{P,1}(Z) - \phi_{P,0}(Z)$ $P$-a.s.*

The proof is nearly identical to that of Lemma 2.2 and so is omitted.

# E. Weak Convergence and Efficiency of $\bar{\psi}_n$

This appendix establishes the asymptotic properties of the proposed estimator $\bar{\psi}_n$, whose estimation error decomposes into a leading EIF term, a remainder term, and a drift term. The analysis proceeds in three steps. First, we prove results establishing the conditions for convergence of the remainder and drift terms. Second, we show that the remainder and drift terms vanish sufficiently fast for our estimator to converge to a tight Gaussian Hilbert-element $\mathbb{H}$. Third, we prove that $\mathbb{H}$ is the optimal limit distribution in the local asymptotic minimax sense.

We begin by introducing some notation and additional definitions required for the analysis. We define the space $L^2(P;\mathcal{H})$ as the Hilbert space of all $P$-Bochner measurable functions $f : \mathcal{Z} \to \mathcal{H}$ such that

$$\|f\|_{L^2(P;\mathcal{H})} := \left(\int \|f(z)\|_{\mathcal{H}}^2 P(dz)\right)^{1/2} < \infty.$$

We use the empirical process notation where $Qf := \mathbb{E}_Q[f(Z)] = \int f(z)Q(dz)$ and $Q_n f := \mathbb{E}_{Q_n}[f(Z)] = \frac{1}{n}\sum_{i=1}^n f(Z_i)$. For brevity, when $P_\star$ appears in a subscript, we replace it by $\star$—e.g., we write $f_\star$ rather than $f_{P_\star}$. Similarly, we write $f_n^r$ instead of $f_{\hat{P}_n^r}$ and $f_n$ instead of $f_{\hat{P}_n}$.

## E.1. Supporting Technical Results

Recall the cross-fitted one-step estimator $\bar{\psi}_n$ defined in Eq. 9. Using empirical process notation, it rewrites as

$$\bar{\psi}_n = \frac{1}{2}\sum_{r=1}^2 \left(\psi_n^r + P_n^s \phi_n^r\right). \tag{26}$$

We restate the remainder and drift terms for each split $r \in \{1,2\}$ and $s = 3 - r$:

$$\mathcal{R}_n^r := \psi_n^r + P_\star \phi_n^r - \psi_\star, \qquad \mathcal{D}_n^r := (P_n^s - P_\star)(\phi_n^r - \phi_\star). \tag{27}$$

Adding and subtracting terms shows that the one-step estimator satisfies the decomposition

$$\bar{\psi}_n - \psi_\star = \frac{1}{n}\sum_{i=1}^n \phi_\star(Z_i) + \frac{1}{2}\sum_{r=1}^2 \left(\mathcal{R}_n^r + \mathcal{D}_n^r\right).$$

Thus, to establish asymptotic linearity, it suffices to show that for each split $r$, $\|\mathcal{R}_n^r\|_{\mathcal{H}} = o_p(n^{-1/2})$ and $\|\mathcal{D}_n^r\|_{\mathcal{H}} = o_p(n^{-1/2})$. The following lemma provides a sufficient condition on the EIF estimator for the drift term to vanish at this rate.

**Lemma E.1** (Lemma 3 in Luedtke & Chung, 2024). *Suppose $\psi$ is pathwise differentiable at $P_\star$ with EIF $\phi_\star \in L^2(P_\star; \mathcal{H})$. For each data split $r \in \{1, 2\}$,*

$$\|\phi_n^r - \phi_\star\|_{L^2(P_\star; \mathcal{H})} = o_p(1) \implies \|\mathcal{D}_n^r\|_{\mathcal{H}} = o_p(n^{-1/2}).$$

Next we establish that consistency of the nuisance estimators is sufficient for consistency of the EIF estimator.

**Lemma E.2.** *Let $\widehat{P}_n \in \mathcal{P}$ be an initial estimate of the data-generating distribution $P_\star$ that is independent of the empirical measure $P_n$. If the following conditions are also satisfied:*

*(i) $\|\pi_n - \pi_\star\|_{L^2(P_\star, X)} = o_p(1)$, and*

*(ii) $\|\theta_{n,a} - \theta_{\star,a}\|_{L^2(P_\star, X; \mathcal{H})} = o_p(1)$ for each $a \in \{0, 1\}$,*

*then $\|\phi_n - \phi_\star\|_{L^2(P_\star; \mathcal{H})} = o_p(1)$.*

*Proof.* By Lemma D.4 and the triangle inequality, $\|\phi_n - \phi_\star\|_{L^2(P_\star; \mathcal{H})} \leq \|\phi_{n,1} - \phi_{\star,1}\|_{L^2(P_\star; \mathcal{H})} + \|\phi_{n,0} - \phi_{\star,0}\|_{L^2(P_\star; \mathcal{H})}$. Thus, it suffices to show that $\|\phi_{n,1} - \phi_{\star,1}\|_{L^2(P_\star; \mathcal{H})} = o_p(1)$ as $\|\phi_{n,0} - \phi_{\star,0}\|_{L^2(P_\star; \mathcal{H})} = o_p(1)$ will hold by an analogous argument. Observe that

$$\phi_{n,1}(z) - \phi_{\star,1}(z) = \frac{a}{\pi_n(x)} \{\Lambda_{x,y} - \theta_{n,1}(x)\} + \theta_{n,1}(x) - \psi_{n,1} - \frac{a}{\pi_\star(x)} \{\Lambda_{x,y} - \theta_{\star,1}(x)\} - \theta_{\star,1}(x) + \psi_{\star,1}.$$

By adding and subtracting $\{\Lambda_{x,y} - \theta_{n,1}(x)\} a/\pi_\star(x)$, this becomes

$$= \frac{a}{\pi_n(x)} \{\Lambda_{x,y} - \theta_{n,1}(x)\} - \frac{a}{\pi_\star(x)} \{\Lambda_{x,y} - \theta_{\star,1}(x)\} + \{\theta_{n,1}(x) - \theta_{\star,1}(x)\} - \{\psi_{n,1} - \psi_{\star,1}\}$$
$$+ \frac{a}{\pi_\star(x)} \{\Lambda_{x,y} - \theta_{n,1}(x)\} - \frac{a}{\pi_\star(x)} \{\Lambda_{x,y} - \theta_{n,1}(x)\}$$
$$= \left( \frac{a}{\pi_n(x)} - \frac{a}{\pi_\star(x)} \right) \{\Lambda_{x,y} - \theta_{n,1}(x)\} + \left( 1 - \frac{a}{\pi_\star(x)} \right) \{\theta_{n,1}(x) - \theta_{\star,1}(x)\} - \{\psi_{n,1} - \psi_{\star,1}\}.$$

Now, let $u_P(z) := a/\pi_P(x)$, and $w_P(z) := \Lambda_{x,y} - \theta_{P,1}(x)$. Then, applying the triangle inequality to the preceding display yields

$$\|\phi_{n,1} - \phi_{\star,1}\|_{L^2(P_\star; \mathcal{H})} \leq \underbrace{\|(u_n - u_\star) w_n\|_{L^2(P_\star; \mathcal{H})}}_{\text{I}} + \underbrace{\|(1 - u_\star)(\theta_{n,1} - \theta_{\star,1})\|_{L^2(P_\star; \mathcal{H})}}_{\text{II}} + \underbrace{\|\psi_{n,1} - \psi_{\star,1}\|_{L^2(P_\star; \mathcal{H})}}_{\text{III}}.$$

**Analysis of I:** Using the fact that $u_n(z) - u_\star(z)$ is a scalar for all $z \in \mathcal{Z}$, we have

$$\|(u_n - u_\star) w_n\|_{L^2(P_\star; \mathcal{H})} = \left( \int \|(u_n(z) - u_\star(z)) w_n(z)\|_{\mathcal{H}}^2 P(dz) \right)^{1/2} = \left[ \mathbb{E}_\star \left( |u_n(Z) - u_\star(Z)|^2 \|w_n(Z)\|_{\mathcal{H}}^2 \right) \right]^{1/2}.$$

Using Hölder's inequality with $(p, q) = (1, \infty)$ yields the following upper bound:

$$\text{I} \leq \left[ \mathbb{E}_\star \left( |u_n(Z) - u_\star(Z)|^2 \right) \right]^{1/2} \left[ \operatorname*{ess\,sup}_{Z \sim P_\star} \left( \|w_n(Z)\|_{\mathcal{H}}^2 \right) \right]^{1/2}$$
$$= \|u_n - u_\star\|_{L^2(P_\star)} \|w_n\|_{L^\infty(P_\star; \mathcal{H})}.$$

We now upper bound this product. First, we have

$$\|u_n - u_\star\|_{L^2(P_\star)}^2 = \mathbb{E}_\star \left( \left| \frac{A}{\pi_n(X)} - \frac{A}{\pi_\star(X)} \right|^2 \right) = \mathbb{E}_\star \left( \left| \frac{1}{\pi_n(X)} - \frac{1}{\pi_\star(X)} \right|^2 A^2 \right).$$

Using the fact that $A^2 = a$ for $A \in \{0, 1\}$, and by Fubini's theorem—permitted due to strong positivity— this becomes

$$= \int \frac{|\pi_\star(x) - \pi_n(x)|^2}{\pi_n^2(x) \pi_\star^2(x)} \left[ \int a g_\star(da \,|\, x) \right] P_{\star, X}(dx),$$

which, by recalling that $g_\star(1 \mid x) = \pi_\star(x)$, simplifies to

$$= \int \frac{|\pi_\star(x) - \pi_n(x)|^2}{\pi_n^2(x)\pi_\star(x)} P_{\star,X}(dx).$$

Thus, by strong positivity, we obtain the following upper bound:

$$\|u_n - u_\star\|_{L^2(P_\star)}^2 \leq \left[ \frac{1}{\{\inf_{P \in \mathcal{P}} \text{ess} \inf_x \pi_P(x)\}^3} \right] \int |\pi_\star(x) - \pi_n(x)|^2 P_{\star,X}(dx) \qquad (*)$$

$$= C_1 \|\pi_n - \pi_\star\|_{L^2(P_{\star,X})}^2,$$

where $C_1$ is some finite constant which does not depend on any $P \in \mathcal{P}$.

Next, we have

$$\|w_n\|_{L^\infty(P_\star;\mathcal{H})}^2 = \operatorname*{ess\,sup}_{X,Y \sim P_{\star,X,Y}} \left( \|\Lambda_{X,Y} - \theta_{n,1}(X)\|_{\mathcal{H}}^2 \right),$$

which, using the inequality $(b - c)^2 \leq 2(b^2 + c^2)$ and the definition of $\theta_{n,1}(X)$ according to Eq. 5, is upper bounded by

$$\leq 2 \operatorname*{ess\,sup}_{X,Y \sim P_{\star,X,Y}} \left( \|\Lambda_{X,Y}\|_{\mathcal{H}}^2 + \|\theta_{n,1}(X)\|_{\mathcal{H}}^2 \right)$$

$$= 2 \operatorname*{ess\,sup}_{X,Y \sim P_{\star,X,Y}} \left( \|\Lambda_{X,Y}\|_{\mathcal{H}}^2 + \left\| \mathbb{E}_{\widehat{P}_n} [\Lambda_{X,Y} \mid A = 1, X] \right\|_{\mathcal{H}}^2 \right).$$

Due to the convexity of the squared Hilbert norm, Jensen's inequality yields that

$$\|w_n\|_{L^\infty(P_\star;\mathcal{H})}^2 \leq 2 \operatorname*{ess\,sup}_{X,Y \sim P_{\star,X,Y}} \left[ \|\Lambda_{X,Y}\|_{\mathcal{H}}^2 + \mathbb{E}_{\widehat{P}_n} \left[ \|\Lambda_{X,Y}\|_{\mathcal{H}}^2 \mid A = 1, X \right] \right],$$

which, by the fact that $\|\Lambda_{x,y}\|_{\mathcal{H}} = \|K_x\|_{\mathcal{H}_\mathcal{X}} \|L_y\|_{\mathcal{H}_\mathcal{Y}} = \sqrt{k(x,x)}\sqrt{\ell(y,y)}$, simplifies to

$$= 2 \operatorname*{ess\,sup}_{X,Y \sim P_{\star,X,Y}} \left[ k(X,X)\ell(Y,Y) + k(X,X)\mathbb{E}_{\widehat{P}_n} [\ell(Y,Y) \mid A = 1, X] \right]$$

$$\leq 4 \left[ \sup_{(x,y) \in (\mathcal{X} \times \mathcal{Y})} |k(x,x)| \, |l(y,y)| \right] =: C_2,$$

where $C_2$ is finite since both $k$ and $\ell$ are bounded kernels.

Combining the upper bounds for $\|u_n - u_\star\|_{L^2(P_\star)}^2$ and $\|w_n\|_{L^\infty(P_\star;\mathcal{H})}^2$ with condition (i) therefore yields that

$$\mathrm{I} \leq \sqrt{C_1} \|\pi_n - \pi_\star\|_{L^2(P_{\star,X})} \sqrt{C_2} = O_p(1)o_p(1) = o_p(1).$$

**Analysis of II:** Observe that by Hölder's inequality with $(p,q) = (\infty, 1)$, we have that

$$\|(1 - u_\star)(\theta_{n,1} - \theta_{\star,1})\|_{L^2(P_\star;\mathcal{H})} \leq \|1 - u_\star\|_{L^\infty(P_\star)} \|\theta_{n,1} - \theta_{\star,1}\|_{L^2(P_{\star,X};\mathcal{H})}$$

$$= \operatorname*{ess\,sup}_{A,X \sim P_{\star,A,X}} \left| 1 - \frac{A}{\pi_\star(X)} \right| \|\theta_{n,1} - \theta_{\star,1}\|_{L^2(P_{\star,X};\mathcal{H})},$$

which, by the triangle inequality and the non-negativity of $A$ and $\pi_\star$, yields

$$\leq \left( 1 + \operatorname*{ess\,sup}_{A,X \sim P_{\star,A,X}} \frac{A}{\pi_\star(X)} \right) \|\theta_{n,1} - \theta_{\star,1}\|_{L^2(P_{\star,X};\mathcal{H})}.$$

Thus, by strong positivity, we obtain

$$\text{II} \le \left(1 + \frac{1}{\inf_{P \in \mathcal{P}} \operatorname{ess\,inf}_x \pi_P(x)}\right) \|\theta_{n,1} - \theta_{\star,1}\|_{L^2(P_{\star,X};\mathcal{H})}$$
$$= C_3 \|\theta_{n,1} - \theta_{\star,1}\|_{L^2(P_{\star,X};\mathcal{H})},$$

where $C_3$ is a constant. Combining this with condition (ii) immediately yields that

$$\text{II} = O_p(1)o_p(1) = o_p(1).$$

**Analysis of III:** Recall that $\psi_{P,1} = \mathbb{E}_P[\theta_{P,1}(X)] = P\theta_{P,1}$ and $\psi_{\star,1}$ are non-random elements of $\mathcal{H}$. We have that

$$\|\psi_{n,1} - \psi_{\star,1}\|_{L^2(P_\star;\mathcal{H})} = \left(\|\psi_{n,1} - \psi_{\star,1}\|_{\mathcal{H}}^2 \int P(dz)\right)^{1/2} = \|\psi_{n,1} - \psi_{\star,1}\|_{\mathcal{H}}.$$

Adding and subtracting $P_\star\theta_{n,1}$, this expression becomes

$$= \|P_n\theta_{n,1} - P_\star\theta_{n,1} + P_\star\theta_{n,1} - P_\star\theta_{\star,1}\|_{\mathcal{H}}$$
$$= \|(P_n - P_\star)\theta_{n,1} + P_\star(\theta_{n,1} - \theta_{\star,1})\|_{\mathcal{H}},$$

which, by the triangle inequality, is upper bounded by

$$\le \|(P_n - P_\star)\theta_{n,1}\|_{\mathcal{H}} + \left\{\|P_\star[\theta_{n,1} - \theta_{\star,1}]\|_{\mathcal{H}}^2\right\}^{1/2}$$
$$= \|(P_n - P_\star)\theta_{n,1}\|_{\mathcal{H}} + \left\{\|\mathbb{E}_\star[\theta_{n,1}(X) - \theta_{\star,1}(X)]\|_{\mathcal{H}}^2\right\}^{1/2}.$$

Due to the convexity of the squared Hilbert norm, Jensen's inequality applied to the second term yields that

$$\text{III} \le \|(P_n - P_\star)\theta_{n,1}\|_{\mathcal{H}} + \left\{\int \|\theta_{n,1}(X) - \theta_{\star,1}(X)\|_{\mathcal{H}}^2 P_{\star,X}(dx)\right\}^{1/2}$$
$$= \|(P_n - P_\star)\theta_{n,1}\|_{\mathcal{H}} + \|\theta_{n,1} - \theta_{\star,1}\|_{L^2(P_{\star,X};\mathcal{H})}.$$

Now, since $\theta_{n,1}$ is deterministic given $\widehat{P}_n$, the expectation of the square of the first term conditioned on $\widehat{P}_n$ is $\frac{1}{n}\operatorname{Var}_{P_\star}(\theta_{n,1}(X)\,|\,\widehat{P}_n) \le \frac{1}{n}\mathbb{E}_\star\left[\|\theta_{n,1}(X)\|_{\mathcal{H}}^2\,|\,\widehat{P}_n\right]$. As established in the analysis of I, $\|\theta_{n,1}(X)\|_{\mathcal{H}}^2$ is uniformly bounded by a finite constant which does not depend on $\widehat{P}_n$. Therefore, by the law of total expectation and Markov's inequality, the first term is $O_p(n^{-1/2}) = o_p(1)$. The second term is $o_p(1)$ by condition (ii). Consequently,

$$\text{III} = o_p(1) + o_p(1) = o_p(1).$$

Thus, $\|\phi_{n,1} - \phi_{\star,1}\|_{L^2(P_\star;\mathcal{H})} = o_p(1) + o_p(1) + o_p(1) = o_p(1)$, completing the proof. $\qquad\square$

We now turn to the remainder term. Using the form given in Eq. 27, we first define it more generally for any candidate distribution $\widehat{P}_n \in \mathcal{P}$ which estimates $P_\star$:

$$\mathcal{R}_n := \psi_n + P_\star\phi_n - \psi_\star.$$

In the following lemma, we establish the double robustness property: the convergence rate of the remainder term is determined by the *product* of the convergence rates of the propensity and outcome estimators.

**Lemma E.3.** *Let $\widehat{P}_n \in \mathcal{P}$ be an initial estimate of the data-generating distribution $P_\star$ that is independent of the empirical measure $P_n$. If the following conditions are satisfied:*

*(i) $\|\pi_n - \pi_\star\|_{L^2(P_{\star,X})} = O_p(n^{-\tau})$ for some scalar $\tau > 0$, and*

*(ii) $\|\theta_{n,a} - \theta_{\star,a}\|_{L^2(P_{\star,X};\mathcal{H})} = O_p(n^{-\gamma_a})$ for some scalar $\gamma_a > 0$ for each $a \in \{0,1\}$,*

*then*

$$\|\mathcal{R}_n\|_{\mathcal{H}} = O_p\left(n^{-[\tau+\min\{\gamma_1,\gamma_0\}]}\right).$$

*In particular, if $\tau + \min\{\gamma_1,\gamma_0\} > 1/2$, then $\|\mathcal{R}_n\|_{\mathcal{H}} = o_p(n^{-1/2})$.*

Note that conditions (i) and (ii) of the above lemma imply the conditions of Lemma E.2.

*Proof.* First, using the decompositions in Eq. 21 and Eq. 25, observe that $\mathcal{R}_n$ rewrites as

$$\begin{aligned}
\mathcal{R}_n &= \psi_n + P_\star\phi_n - \psi_\star \\
&= \psi_{n,1} - \psi_{n,0} + P_\star\left(\phi_{n,1} - \phi_{n,0}\right) - \psi_{\star,1} + \psi_{\star,0} \\
&= \psi_{n,1} + P_\star\phi_{n,1} - \psi_{\star,1} - \left[\psi_{n,0} + P_\star\phi_{n,0} - \psi_{\star,0}\right].
\end{aligned}$$

Let $\mathcal{R}_{n,1} := \psi_{n,1} + P_\star\phi_{n,1} - \psi_{\star,1}$ and $\mathcal{R}_{n,0} := \psi_{n,0} + P_\star\phi_{n,0} - \psi_{\star,0}$. We have by the triangle inequality that

$$\|\mathcal{R}_n\|_{\mathcal{H}} \le \|\mathcal{R}_{n,1}\|_{\mathcal{H}} + \|\mathcal{R}_{n,0}\|_{\mathcal{H}}.$$

Therefore, to establish the rate for $\|\mathcal{R}_n\|_{\mathcal{H}}$, it suffices to bound the norms of the treatment group-wise remainder terms. Here we focus on bounding $\|\mathcal{R}_{n,1}\|_{\mathcal{H}}$, and $\|\mathcal{R}_{n,0}\|_{\mathcal{H}}$ bounds by analogous arguments.

By the definition of $\mathcal{R}_{n,1}$ and the form of $\phi_{n,1}$ due to Lemma D.4, we have that

$$\begin{aligned}
\mathcal{R}_{n,1} &= \psi_{n,1} + P_\star\phi_{n,1} - \psi_{\star,1} \\
&= \psi_{n,1} + \mathbb{E}_\star\left[\frac{A}{\pi_n(X)}\left\{\Lambda_{X,Y} - \theta_{n,1}(X)\right\} + \theta_{n,1}(X) - \psi_{n,1}\right] - \mathbb{E}_\star\left[\theta_{\star,1}(X)\right] \\
&= \mathbb{E}_\star\left[\frac{A}{\pi_n(X)}\left\{\Lambda_{X,Y} - \theta_{n,1}(X)\right\}\right] + \mathbb{E}_\star\left[\theta_{n,1}(X) - \theta_{\star,1}(X)\right].
\end{aligned}$$

Using Fubini's theorem, this display rewrites as

$$\begin{aligned}
\mathcal{R}_{n,1} &= \int \frac{1}{\pi_n(x)}\left[\int a\left\{\int \Lambda_{x,y}\, P_{\star,Y|A,X}(dy \mid a, x) - \theta_{n,1}(x)\right\} g_\star(da \mid x)\right] P_{\star,X}(dx) \\
&\quad + \mathbb{E}_\star\left[\theta_{n,1}(X) - \theta_{\star,1}(X)\right] \\
&= \int \frac{1}{\pi_n(x)}\left[g_\star(1 \mid x)\left\{\int \Lambda_{x,y}\, P_{\star,Y|A,X}(dy \mid 1, x) - \theta_{n,1}(x)\right\}\right] P_{\star,X}(dx) \\
&\quad + \mathbb{E}_\star\left[\theta_{n,1}(X) - \theta_{\star,1}(X)\right],
\end{aligned}$$

which, by recognizing that $\int \Lambda_{x,y}\, P_{\star,Y|A,X}(dy \mid 1, x) = \theta_{\star,1}$ (by Eq. 5) and using $g_\star(1|x) = \pi_\star(x)$, becomes

$$\begin{aligned}
&= \mathbb{E}_\star\left[\frac{\pi_\star(X)}{\pi_n(X)}\left\{\theta_{\star,1}(X) - \theta_{n,1}(X)\right\}\right] + \mathbb{E}_\star\left[\theta_{n,1}(X) - \theta_{\star,1}(X)\right] \\
&= \mathbb{E}_\star\left[\left(1 - \frac{\pi_\star(X)}{\pi_n(X)}\right)\left\{\theta_{n,1}(X) - \theta_{\star,1}(X)\right\}\right].
\end{aligned}$$

We now bound the norm of this term. By Jensen's inequality for Bochner integrals and the fact that $\pi_\star(x)/\pi_n(x)$ is real-valued for all $x$, we have

$$\begin{aligned}
\|\mathcal{R}_{n,1}\|_{\mathcal{H}} &\le \mathbb{E}_\star\left[\left\|\left(1 - \frac{\pi_\star(X)}{\pi_n(X)}\right)\left\{\theta_{n,1}(X) - \theta_{\star,1}(X)\right\}\right\|_{\mathcal{H}}\right] \\
&= \mathbb{E}_\star\left[\left|1 - \frac{\pi_\star(X)}{\pi_n(X)}\right| \|\{\theta_{n,1}(X) - \theta_{\star,1}(X)\}\|_{\mathcal{H}}\right].
\end{aligned}$$

Using the Cauchy-Schwarz inequality for Bochner integrals, this expression is upper bounded by

$$\leq \left( \mathbb{E}_\star \left[ \left| 1 - \frac{\pi_\star(X)}{\pi_n(X)} \right|^2 \right] \right)^{1/2} \left( \mathbb{E}_\star \left[ \|\theta_{n,1}(X) - \theta_{\star,1}(X)\|_{\mathcal{H}}^2 \right] \right)^{1/2},$$

which, by recalling the definition of $L^2(P_{\star,X}; \mathcal{H})$, simplifies to

$$= \left( \mathbb{E}_\star \left[ \frac{|\pi_n(X) - \pi_\star(X)|^2}{\pi_n^2(X)} \right] \right)^{1/2} \|\theta_{n,1} - \theta_{\star,1}\|_{L^2(P_{\star,X};\mathcal{H})}.$$

Due to the strong positivity assumption from Section 2.1, we then have the following bound:

$$\|\mathcal{R}_{n,1}\|_{\mathcal{H}} \leq \frac{1}{\inf_{P \in \mathcal{P}} \operatorname{ess\,inf}_x \pi_P(x)} \left( \mathbb{E}_\star \left[ |\pi_n(X) - \pi_\star(X)|^2 \right] \right)^{1/2} \|\theta_{n,1} - \theta_{\star,1}\|_{L^2(P_{\star,X};\mathcal{H})}$$

$$= C_1 \|\pi_n - \pi_\star\|_{L^2(P_{\star,X})} \|\theta_{n,1} - \theta_{\star,1}\|_{L^2(P_{\star,X};\mathcal{H})},$$

where $C_1$ is a finite constant which does not depend on any $P \in \mathcal{P}$. Using conditions (i) and (ii) therefore yields that

$$\|\mathcal{R}_{n,1}\|_{\mathcal{H}} = O_p(1) O_p(n^{-\tau}) O_p(n^{-\gamma_1}) = O_p\left( n^{-[\tau+\gamma_1]} \right).$$

An analogous result holds for the control group due to symmetry, so that

$$\|\mathcal{R}_{n,0}\|_{\mathcal{H}} = O_p\left( n^{-[\tau+\gamma_0]} \right).$$

Combining these bounds yields:

$$\|\mathcal{R}_n\|_{\mathcal{H}} = O_p\left( n^{-[\tau+\gamma_1]} \right) + O_p\left( n^{-[\tau+\gamma_0]} \right) = O_p\left( n^{-[\tau+\min\{\gamma_1,\gamma_0\}]} \right)$$

since the slower convergence between $\|\mathcal{R}_{n,1}\|_{\mathcal{H}}$ and $\|\mathcal{R}_{n,0}\|_{\mathcal{H}}$ determines the rate of their sum. $\qquad \square$

### E.2. Proof of Theorem 3.1

We now combine all preceding results in this appendix to prove the central result of our main text, restated below.

**Theorem 3.1** (Weak convergence). *Let $\phi_\star$ be the EIF of $\psi$ at $P_\star$. For $r \in \{1, 2\}$, suppose $\widehat{P}_n^r$ is such that:*

*(i)* $\|\pi_n^r - \pi_\star\|_{L^2(P_{\star,X})} = O_p(n^{-\tau_r})$ *for scalar $\tau_r > 0$,*

*(ii)* $\|\theta_{n,a}^r - \theta_{\star,a}\|_{L^2(P_{\star,X};\mathcal{H})} = O_p(n^{-\gamma_{a,r}})$ *for scalar $\gamma_{a,r} > 0$ for each $a \in \{0, 1\}$, and*

*(iii)* $\tau_r + \min\{\gamma_{0,r}, \gamma_{1,r}\} > 1/2$.

*Then, letting '$\rightsquigarrow$' denote weak convergence in $\mathcal{H}$, we have*

*1.* $\bar{\psi}_n - \psi_\star = \frac{1}{n} \sum_{i=1}^n \phi_\star(Z_i) + o_p(n^{-1/2})$,

*2.* $\sqrt{n}\left( \bar{\psi}_n - \psi_\star \right) \rightsquigarrow \mathbb{H}$,

*where $\mathbb{H}$ is a tight $\mathcal{H}$-valued random variable such that $\langle \mathbb{H}, h \rangle_{\mathcal{H}} \sim \mathcal{N}\left( 0, \mathbb{E}_\star \left[ \langle \phi_\star(Z), h \rangle_{\mathcal{H}}^2 \right] \right)$ for every $h \in \mathcal{H}$.*

*Proof.* Proposition D.3 yields that $\psi$ is pathwise differentiable at $P_\star$ and Lemma 2.2 shows it has EIF $\phi_\star \in L^2(P_\star; \mathcal{H})$.

Since $\widehat{P}_n^r$ is the initial estimate of $P_\star$, conditions (i) and (ii) imply, via Lemma E.2 for each split $r \in \{1, 2\}$, that $\|\phi_n^r - \phi_\star\|_{L^2(P_\star;\mathcal{H})} = o_p(1)$ for each $r$. Consequently, Lemma E.1 implies that $\|\mathcal{D}_n^r\|_{\mathcal{H}} = o_p(n^{-1/2})$ for each $r$.

Conditions (i), (ii), (iii) also imply, by way of Lemma E.3 for each split $r \in \{1, 2\}$, that $\|\mathcal{R}_n^r\|_{\mathcal{H}} = o_p(n^{-1/2})$ for each $r$.

These results satisfy the conditions of Theorem 2 in Luedtke & Chung (2024), which we invoke to conclude the proof. $\qquad \square$

### E.3. Proof of Theorem 3.2

To establish the optimality of our estimator, we use the general theory of efficiency for 'statistical experiments' developed in Chapter 3.12 of van der Vaart & Wellner (2023). We map our setting (Sec. 2.1) to their framework as follows: a statistical experiment corresponds to i.i.d. sampling of $n$ observations from $P_{n,s} \in \mathcal{P}$, a perturbation of $P_\star$ that is indexed by score functions $s \in L_0^2(P_\star)$. The resulting sequence of statistical experiments is the collection $(\mathcal{Z}^n, \mathcal{B}_{\mathcal{Z}}^n, P_{n,s}^n : s \in L_0^2(P_\star))$, where the superscript denotes the usual $n$-fold product space/measure.

Note that an estimator (sequence implied by) $\widetilde{\psi}_n$ is said to be regular at $P_\star$ if and only if, for all $s \in L_0^2(P_\star)$, every QMD submodel $\{P_\epsilon\} \subset \mathcal{P}$ at $P_\star$ with score $s$, and all $\epsilon_n = O(n^{-1/2})$, the sequence $\sqrt{n}[\widetilde{\psi}_n - \psi(P_{\epsilon_n})]$ converges weakly to a fixed, tight $\mathcal{H}$-valued random variable $\widetilde{\mathbb{H}}$ under i.i.d. sampling of $n$ observations from $P_{\epsilon_n}$.

We now prove that our estimator achieves the minimax lower bound.

**Theorem 3.2** (Local asymptotic minimax optimality). *For any score $s \in L_0^2(P_\star)$, let $\{P_{s,\epsilon}\} \subset \mathcal{P}$ be a QMD submodel such that $P_{s,0} = P_\star$. Define the local asymptotic minimax risk for an estimator sequence $(\check{\psi}_n)_{n=1}^\infty$ as*

$$\mathrm{LAMRisk}_\rho(\check{\psi}_n; P_\star) \coloneqq$$
$$\sup_I \liminf_{n \to \infty} \sup_{s \in I} \mathbb{E}_{s, \frac{1}{\sqrt{n}}} \left[ \rho \left( \sqrt{n} \left[ \check{\psi}_n - \psi_{s, \frac{1}{\sqrt{n}}} \right] \right) \right],$$

*where $\rho : \mathcal{H} \to \mathbb{R}$ is a nonnegative map, the first supremum is over all finite subsets of $L_0^2(P_\star)$, and the expectation is under the product measure $P_{s,1/\sqrt{n}}^n$. Suppose the conditions of Thm. 3.1 hold. Further, let $(\check{\psi}_n)_{n=1}^\infty$ be any Borel-measurable estimator sequence and $\rho$ be any subconvex function that is continuous a.s. under the law of $\mathbb{H}$. Provided that the sequence $\rho(\sqrt{n}(\check{\psi}_n - \psi_{s,1/\sqrt{n}}))$ is asymptotically uniformly integrable under $P_{s,1/\sqrt{n}}$, we have:*

$$\mathrm{LAMRisk}_\rho(\widetilde{\psi}_n; P_\star) \geq \mathbb{E}_\star [\rho(\mathbb{H})] = \mathrm{LAMRisk}_\rho(\bar{\psi}_n; P_\star).$$

*Proof.* Note that $\mathcal{P}$ is assumed to be locally nonparametric, and define $H \coloneqq L_0^2(P_\star)$, the tangent space at $P_\star$. For any score $s \in H$, let $\{P_{s,t}\} \subset \mathcal{P}$ be a QMD submodel at $P_\star$ with score $s$. We define our sequence of statistical experiments via i.i.d. sampling from $P_{n,s} \coloneqq P_{s,1/\sqrt{n}}$. As noted in Example 3.12.1 of van der Vaart & Wellner (2023), this sequence of experiments is locally asymptotically normal (LAN).

We now consider the sequence of parameters $\psi(P_{n,s})$ and the norming operators defined by $r_n(h) \coloneqq \sqrt{n}h$. By Proposition D.3, $\psi$ is pathwise differentiable at $P_\star$. By definition, this implies the existence of a continuous linear map $\dot{\psi}_{P_\star} : H \to \mathcal{H}$ (specifically, the local parameter from the statement of Proposition D.3) such that for the sequence $P_{n,s}$, which corresponds to the path $t_n = 1/\sqrt{n}$, we have

$$\sqrt{n}(\psi(P_{n,s}) - \psi(P_\star)) = \left[ \frac{\psi(P_{s,1/\sqrt{n}}) - \psi(P_\star)}{1/\sqrt{n}} \right] \to \dot{\psi}_{P_\star}(s) \quad \text{in } \mathcal{H}. \tag{28}$$

As this convergence holds for all $s \in H$, the above sequence of parameters is regular at $P_\star$ with respect to the norming operators $h \mapsto \sqrt{n}h$.

Moreover, since our pathwise differentiability result holds for every score in $H = L_0^2(P_\star)$ and all QMD submodels generated by those scores, both the LAN and parameter regularity conditions are satisfied regardless of the specific submodel chosen to construct the statistical experiments.

Now, we establish the lower bound of the desired result. We have by supposition that the conditions of Thm. 3.1 hold. These imply, via Theorem 2 in Luedtke & Chung (2024), that $\bar{\psi}_n$ is a regular estimator. We invoke Theorem 3.12.2 from van der Vaart & Wellner (2023) with the linear subspace $H$ and the regular parameter sequence $\psi(P_{s,1/\sqrt{n}})$ as defined above, and with $\mathbf{B} \coloneqq \mathcal{H}$. Since $\mathcal{H}$ is a Hilbert space, we identify the dual space $\mathbf{B}^*$ with $\mathcal{H}$ via the Riesz representation theorem. Moreover, $\bar{H} = H = L_0^2(P_\star)$ by the completeness of Hilbert spaces. Subsequently, the duality condition (24) identifies the Hermitian adjoint of the local parameter $\dot{\psi}_\star$ as the efficient influence operator $\dot{\psi}_\star^* : \mathcal{H} \to \bar{H}$.

Consequently, Theorem 3.12.2 implies that the sequence $\sqrt{n}(\bar{\psi}_n - \psi_\star)$ converges weakly to a tight limit $G + W$ in $\mathcal{H}$, where the law of $G$ concentrates on the local parameter space $\mathcal{H}_\star \coloneqq \dot{\psi}_\star(L_0^2(P_\star))$ and is such that

$$\langle G, h \rangle_{\mathcal{H}} \sim \mathcal{N} \left( 0, \left\| \dot{\psi}_\star^*(h) \right\|_{L^2(P_\star)}^2 \right) \quad \text{for all } h \in \mathcal{H}.$$

Recall that, by definition, the EIF $\phi_\star(z)$ is $P_\star$-a.s. equal to the Riesz representation of $\dot{\psi}_\star^*(\cdot)(z)$. Thus, for all $h \in \mathcal{H}$,

$$\mathrm{Var}_\star(\langle G, h \rangle_\mathcal{H}) = \mathbb{E}_\star \left[ \left( \dot{\psi}_\star^*(h)(Z) \right)^2 \right] = \mathbb{E}_\star \left[ \langle h, \phi_\star(Z) \rangle_\mathcal{H}^2 \right].$$

Now, Theorem 3.1 itself implies that the sequence $\sqrt{n}(\bar{\psi}_n - \psi_\star)$ converges weakly to a tight Gaussian element $\mathbb{H}$ in $\mathcal{H}$, which is such that, for all $h \in \mathcal{H}$,

$$\mathrm{Var}(\langle \mathbb{H}, h \rangle_\mathcal{H}) = \mathbb{E}_\star \left[ \langle h, \phi_\star(Z) \rangle_\mathcal{H}^2 \right].$$

It is clear from Eq. 24 that $\dot{\psi}_\star^*$ only depends on its argument through its projection onto the local parameter space. Thus, the law of $\mathbb{H}$ also concentrates on $\dot{\mathcal{H}}_\star$.

Comparing the preceding two displays, we observe that for every $h \in \mathcal{H}$, the marginal distributions of $\langle G, h \rangle_\mathcal{H}$ and $\langle \mathbb{H}, h \rangle_\mathcal{H}$ are identical zero-mean normals. Since $\mathcal{H}$ is a separable RKHS and so is $\dot{\mathcal{H}}_\star$ and $\dot{\mathcal{H}}_\star$, the distribution of a tight Gaussian random element of this space is uniquely determined by these marginals. Therefore, $\mathbb{H} = G$ in law $P_\star$-a.s., which further implies that the noise term $W = 0$ $P_\star$-a.s. Thus, $\bar{\psi}_n$ is efficient.

We now invoke Theorem 3.12.5 from van der Vaart & Wellner (2023). The RKHS $\mathcal{H}$ is a separable Banach space. As noted in Example 3.12.6 of van der Vaart & Wellner (2023), in separable Banach spaces, '$\tau(\mathbf{B}')$-subconvexity' coincides with standard subconvexity. Furthermore, Borel-measurability under the norm topology of $\mathcal{H}$ implies asymptotic measurability (and hence, '$\mathbf{B}'$-measurability'). In fact, in this setting, inner and outer expectations collapse to the standard notion of expectation. Consequently, for any subconvex loss function $\rho$, a direct application of Theorem 3.12.5 yields the following lower bound:

$$\mathrm{LAMRisk}_\rho(\check{\psi}_n; P_\star) \geq \mathbb{E}_\star \left[ \rho(G) \right].$$

Since we established that $\mathbb{H} = G$ in law $P_\star$-a.s., it remains only to show that the local asymptotic minimax risk of our estimator $\bar{\psi}_n$ converges to $\mathbb{E}[\rho(\mathbb{H})]$. Recall that we have established the regularity of $\bar{\psi}_n$. By definition, this implies that for any $s \in L_0^2(P_\star)$, the weak convergence $\sqrt{n}(\bar{\psi}_n - \psi(P_{s,1/\sqrt{n}})) \rightsquigarrow \mathbb{H}$ holds under the sequence of probability measures $P_{s,1/\sqrt{n}}$.

We invoke Theorem 1.11.3 from van der Vaart & Wellner (2023) with $\mathbb{D} := \mathcal{H}$ and $\mathbb{D}_0 := \dot{\mathcal{H}}_\star \subset \mathcal{H}$, but applied to the sequence of expectations under $P_{s,1/\sqrt{n}}$. Thus, under the assumptions that $\rho$ is continuous at every point in $\dot{\mathcal{H}}_\star$ and the sequence $\rho(\sqrt{n}(\bar{\psi}_n - \psi(P_{1/\sqrt{n}})))$ is asymptotically uniformly integrable under $P_{s,1/\sqrt{n}}$, it follows from Theorem 1.11.3(i) that

$$\mathbb{E}_{s,1/\sqrt{n}} \left[ \rho(\sqrt{n}(\bar{\psi}_n - \psi(P_{s,1/\sqrt{n}}))) \right] \longrightarrow \mathbb{E}_\star \left[ \rho(\mathbb{H}) \right].$$

As this holds for any $s \in L_0^2(P_\star)$, it holds that, for any finite $I \subset L_0^2(P_\star)$,

$$\liminf_{n \to \infty} \sup_{s \in I} \mathbb{E}_{s,1/\sqrt{n}} \left[ \rho(\sqrt{n}(\bar{\psi}_n - \psi(P_{s,1/\sqrt{n}}))) \right] = \mathbb{E}_\star \left[ \rho(\mathbb{H}) \right].$$

As this holds for all $I$ and the right-hand side does not depend on $I$,

$$\sup_I \liminf_{n \to \infty} \sup_{s \in I} \mathbb{E}_{s,1/\sqrt{n}} \left[ \rho(\sqrt{n}(\bar{\psi}_n - \psi(P_{s,1/\sqrt{n}}))) \right] = \mathbb{E}_\star \left[ \rho(\mathbb{H}) \right].$$

By definition, the left-hand side is $\mathrm{LAMRisk}_\rho(\bar{\psi}_n; P_\star)$. $\qquad\square$

# F. Guarantees for Inference

### F.1. SKCD Test

This appendix establishes guarantees for the SKCD testing procedure. We first show that, asymptotically, our proposed test statistic controls type 1 error and has power under a fixed alternative. Subsequently, we show that inverting the testing procedure yields asymptotically valid uniform confidence bands.

We denote the $(1 - \alpha)$-quantile of the limit distribution by $q_\alpha$. Recall that we estimate this quantile via $\widehat{q}_{n,\alpha}$ using the multiplier bootstrap (Alg. 1). We define the $(1 - \alpha)$-level confidence set for $\psi_\star$ as

$$\mathcal{C}_n(\widehat{q}_{n,\alpha}) := \{h \in \mathcal{H} : \langle \Omega_n(\bar{\psi}_n - h), \bar{\psi}_n - h \rangle_{\mathcal{H}} \leq \widehat{q}_{n,\alpha}/n\} \tag{29}$$

For brevity in the upcoming proofs, we also define the norm $\|\cdot\|_\Omega$ for any $\Omega \in \mathscr{W}$. Observe that since $\Omega$ is a self-adjoint, strictly positive-definite continuous operator on a Hilbert space $\mathcal{H}$, it induces a valid inner product $\langle h_1, h_2 \rangle_\Omega := \langle \Omega h_1, h_2 \rangle_{\mathcal{H}}$, which in turn induces a valid norm $\|h\|_\Omega := \sqrt{\langle h, h \rangle_\Omega} = \sqrt{\langle \Omega h, h \rangle_{\mathcal{H}}}$.

### F.1.1. PROOF OF THEOREM 3.3

**Theorem 3.3** (Validity of the SKCD test in Alg. 1). *If the conditions of Thm. 3.1 hold, $\Omega_\star \in \mathscr{W}$, and $\Omega_n \in \mathscr{W}$ satisfies $\|\Omega_n - \Omega_\star\|_{\mathrm{op}} = o_p(1)$, then*

1. *(type 1 error control)* $\lim_{n\to\infty} P_\star^n \{T_n > \widehat{q}_{n,\alpha}\} = \alpha$ *for all $P_\star \in \mathcal{P}_0$, and*
2. *(test consistency)* $\lim_{n\to\infty} P_\star^n \{T_n > \widehat{q}_{n,\alpha}\} = 1$ *for any fixed $P_\star \in \mathcal{P} \setminus \mathcal{P}_0$.*

*Proof.* We assume that the conditions of Theorem 3.1 hold. We also have by supposition that $\Omega_n, \Omega_\star \in \mathscr{W}$ with $\|\Omega_n - \Omega_\star\|_{\mathrm{op}} = o_p(1)$. From Proposition D.3, $\psi$ is pathwise differentiable at $P_\star$, and from Lemma 2.2, it has an EIF $\phi_\star \in L^2(P_\star; \mathcal{H})$ such that $\|\phi_\star\|_{L^2(P_\star; \mathcal{H})} > 0$.

Since $\widehat{P}_n^r$ serves as the initial estimate of $P_\star$ for each $r \in \{1, 2\}$, conditions (i) and (ii) of Theorem 3.1 regarding the convergence rates of the nuisance parameters imply via Lemma E.2 that $\|\phi_n^r - \phi_\star\|_{L^2(P_\star; \mathcal{H})} = o_p(1)$ for each $r \in \{1, 2\}$.

Consequently, the conditions for Theorem 4 in Luedtke & Chung (2024) are satisfied, yielding

$$\widehat{q}_{n,\alpha} \xrightarrow{p} q_\alpha. \tag{30}$$

*Statement 1*:

Consider any $P_\star \in \mathcal{P}_0$, which implies that the sharp null hypothesis $H_0 : \psi_\star = \psi_0$ holds. Recall that our test rejects $H_0$ if $T_n > \widehat{q}_{n,\alpha}$. By the definition of the confidence set $\mathcal{C}_n(\widehat{q}_{n,\alpha})$ in (29), the rejection event is equivalent to $\psi_0$ falling outside the confidence set:

$$\{T_n > \widehat{q}_{n,\alpha}\} \iff \left\{n \|\bar{\psi}_n - \psi_0\|_{\Omega_n}^2 > \widehat{q}_{n,\alpha}\right\} \iff \{\psi_0 \notin \mathcal{C}_n(\widehat{q}_{n,\alpha})\}.$$

Since $\widehat{q}_{n,\alpha} \xrightarrow{p} q_\alpha$, the conditions of Theorem 3 (i) from Luedtke & Chung (2024) are satisfied, yielding

$$\lim_{n\to\infty} P_\star^n (T_n > \widehat{q}_{n,\alpha}) = \lim_{n\to\infty} P_\star^n (\psi_0 \notin \mathcal{C}_n(\widehat{q}_{n,\alpha})) = \lim_{n\to\infty} P_\star^n (\psi_\star \notin \mathcal{C}_n(\widehat{q}_{n,\alpha})) = 1 - (1 - \alpha) = \alpha.$$

*Statement 2*:

Consider a fixed alternative $P_\star \in \mathcal{P} \setminus \mathcal{P}_0$. This implies $\psi_\star \neq \psi_0$. Since $\Omega_\star$ is positive definite, we have that

$$\delta := \|\psi_\star - \psi_0\|_{\Omega_\star} > 0. \tag{31}$$

Observe that, by definition of $T_n$, and the reverse triangle inequality, the event $E := \{T_n \leq \widehat{q}_{n,\alpha}\}$ satisfies the following ordering of events:

$$\begin{aligned}
E &= \left\{\|\bar{\psi}_n - \psi_0\|_{\Omega_n} \leq \sqrt{\frac{\widehat{q}_{n,\alpha}}{n}}\right\} \\
&\subseteq \left\{\|\psi_\star - \psi_0\|_{\Omega_n} - \|\bar{\psi}_n - \psi_\star\|_{\Omega_n} \leq \sqrt{\frac{\widehat{q}_{n,\alpha}}{n}}\right\} \\
&= \{S_n \leq V_n\}
\end{aligned}$$

with $S_n := \left\| \psi_\star - \psi_0 \right\|_{\Omega_n}$, and $V_n := \left\| \bar{\psi}_n - \psi_\star \right\|_{\Omega_n} + \sqrt{\frac{\widehat{q}_{n,\alpha}}{n}}$. Taking $\delta$ as in (31), this yields

$$
\begin{aligned}
P_\star^n(E) \leq P_\star^n(S_n \leq V_n) &= P_\star^n(S_n \leq V_n, S_n > \delta/2) + P_\star^n(S_n \leq V_n, S_n \leq \delta/2) \\
&\leq \underbrace{P_\star^n(V_n \geq \delta/2)}_{\text{I}} + \underbrace{P_\star^n(S_n \leq \delta/2)}_{\text{II}}.
\end{aligned}
\tag{32}
$$

We now analyze the asymptotic behavior of these terms.

**Analysis of I:** Using the definitions of $\left\| \cdot \right\|_{\Omega_n}$ and the operator norm,

$$
V_n = \left\| \bar{\psi}_n - \psi_\star \right\|_{\Omega_n} + \sqrt{\frac{\widehat{q}_{n,\alpha}}{n}} = \sqrt{\left\langle \Omega_n \left( \bar{\psi}_n - \psi_\star \right), \bar{\psi}_n - \psi_\star \right\rangle_{\mathcal{H}}} + \sqrt{\frac{\widehat{q}_{n,\alpha}}{n}} \leq \sqrt{\left\| \Omega_n \right\|_{\text{op}}} \left\| \bar{\psi}_n - \psi_\star \right\|_{\mathcal{H}} + \sqrt{\frac{\widehat{q}_{n,\alpha}}{n}}.
$$

Now, since $\Omega_n$ is a continuous operator, we have $\left\| \Omega_n \right\|_{\text{op}} = O_p(1)$. By Theorem 3.1 and Prokhorov's theorem (via tightness of $\mathbb{H}$ in $\mathcal{H}$), we also have that $n^{1/2}(\bar{\psi}_n - \psi_\star) = O_p(1) \implies \bar{\psi}_n - \psi_\star = O_p(n^{-1/2})$, which implies that $\left\| \bar{\psi}_n - \psi_\star \right\|_{\mathcal{H}} = o_p(1)$. Hence,

$$
\sqrt{\left\| \Omega_n \right\|_{\text{op}}} \left\| \bar{\psi}_n - \psi_\star \right\|_{\mathcal{H}} = O_P(1) o_p(1) = o_p(1).
$$

Moreover, we established that $\widehat{q}_{n,\alpha} \xrightarrow{p} q_\alpha$ in (30), with $q_\alpha$ a constant. Thus, $\widehat{q}_{n,\alpha} = O_p(1)$, which implies that $\sqrt{\widehat{q}_{n,\alpha}/n} = o_p(1)$. Combining this with the preceding two displays yields that $V_n = o_p(1)$. Thus, since $\delta$ from (31) is strictly positive,

$$
\lim_{n \to \infty} P_\star^n(V_n \geq \delta/2) = 0.
$$

**Analysis of II:** Since $\delta > 0$, observe that

$$
S_n \leq \delta/2 \implies \delta - S_n \geq \delta/2 \implies |S_n - \delta| \geq \delta/2.
$$

Therefore,

$$
P_\star^n(S_n \leq \delta/2) \leq P_\star^n(|S_n - \delta| \geq \delta/2).
\tag{33}
$$

Now, the inequality $\left| \sqrt{b} - \sqrt{c} \right| \leq \sqrt{|b - c|}$ for $b, c \in \mathbb{R}_{\geq 0}$ yields that

$$
\begin{aligned}
|S_n - \delta| &= \left| \left\| \psi_\star - \psi_0 \right\|_{\Omega_n} - \left\| \psi_\star - \psi_0 \right\|_{\Omega_\star} \right| \\
&\leq \sqrt{\left| \left\| \psi_\star - \psi_0 \right\|_{\Omega_n}^2 - \left\| \psi_\star - \psi_0 \right\|_{\Omega_\star}^2 \right|} \\
&= \sqrt{\left| \left\langle \Omega_n \left( \psi_\star - \psi_0 \right), \psi_\star - \psi_0 \right\rangle_{\mathcal{H}} - \left\langle \Omega_\star \left( \psi_\star - \psi_0 \right), \left( \psi_\star - \psi_0 \right) \right\rangle_{\mathcal{H}} \right|},
\end{aligned}
$$

which, by linearity of the inner product and the definition of the operator norm $\left\| \cdot \right\|_{\text{op}}$, simplifies to

$$
\begin{aligned}
&= \sqrt{\left| \left\langle \left( \Omega_n - \Omega_\star \right) \left( \psi_\star - \psi_0 \right), \psi_\star - \psi_0 \right\rangle_{\mathcal{H}} \right|} \\
&\leq \sqrt{\left\| \Omega_n - \Omega_\star \right\|_{\text{op}}} \left\| \psi_\star - \psi_0 \right\|_{\mathcal{H}} = o_p(1) O_p(1) = o_p(1).
\end{aligned}
$$

Thus, $S_n \xrightarrow{p} \delta$. It follows from Eq. 33 and the definition of convergence in probability that

$$
\lim_{n \to \infty} P_\star^n(S_n \leq \delta/2) \leq \lim_{n \to \infty} P_\star^n(|S_n - \delta| \geq \delta/2) = 0.
$$

Finally, due to the upper bound (32) on the probability of failure to reject, combining the results for I and II yields $\lim_{n \to \infty} P_\star^n(E) = 0$. Taking the complement event of $E$ and rearranging terms completes the proof of asymptotic power 1 against fixed alternatives. $\qquad \square$

F.1.2. PROOF OF THEOREM 3.4

We now validate our construction of uniform confidence bands for the SCoDiTE, formed by inverting the testing procedure, and establish their asymptotic validity.

**Theorem 3.4** (Uniform confidence band for the SCoDiTE). *Suppose the conditions of Thm. 3.1 hold, $\Omega_\star \in \mathscr{W}_{\text{inv}}$, and the bootstrap quantile $\widehat{q}_{n,\alpha}$ is constructed (Alg. 1) using $\Omega_n \in \mathscr{W}_{\text{inv}}$ such that $\|\Omega_n - \Omega_\star\|_{\text{op}} = o_p(1)$. Define $w_n : \mathcal{X} \times \mathcal{Y} \to \mathbb{R}$ that satisfies $w_n^2(x,y) := \langle \Lambda_{x,y}, \Omega_n^{-1} \Lambda_{x,y} \rangle_{\mathcal{H}} \, \widehat{q}_{n,\alpha}/n$, and let $B_n(x,y) := [\bar{\psi}_n(x,y) - w_n(x,y), \ \bar{\psi}_n(x,y) + w_n(x,y)]$. Then,*

$$\lim_{n \to \infty} P_\star^n \left( \psi_\star(x,y) \in B_n(x,y) \quad \text{for all } x,y \right) \geq 1 - \alpha.$$

*Proof.* Recall that $\Omega_n \in \mathscr{W}_{\text{inv}}$ is a continuous self-adjoint positive-definite operator that is boundedly invertible. Thus, the operators $\Omega_n^{1/2}$ and $\Omega_n^{-1/2}$ exist and are self-adjoint. Recall also that by the reproducing property of the feature map $\Lambda$, $f(x,y) = \langle f, \Lambda_{x,y} \rangle_{\mathcal{H}}$ for all $f \in \mathcal{H}$. We then have for any $f \in \mathcal{H}$ that

$$|f(x,y)| = \left| \left\langle \Omega_n^{-1/2} \Omega_n^{1/2} f, \Lambda_{x,y} \right\rangle_{\mathcal{H}} \right|,$$

which, using the self-adjointness of $\Omega_n^{-1/2}$, simplifies to

$$= \left| \left\langle \Omega_n^{1/2} f, \Omega_n^{-1/2} \Lambda_{x,y} \right\rangle_{\mathcal{H}} \right|.$$

Using Cauchy-Schwarz's inequality therefore yields:

$$|f(x,y)| \leq \left\| \Omega_n^{1/2} f \right\|_{\mathcal{H}} \left\| \Omega_n^{-1/2} \Lambda_{x,y} \right\|_{\mathcal{H}} = \sqrt{\langle \Omega_n f, f \rangle_{\mathcal{H}}} \sqrt{\langle \Omega_n^{-1} \Lambda_{x,y}, \Lambda_{x,y} \rangle_{\mathcal{H}}} = \|f\|_{\Omega_n} \|\Lambda_{x,y}\|_{\Omega_n^{-1}},$$

where the penultimate equality uses the definition of the adjoint, and the final equality holds by definition.

Let $f := \bar{\psi}_n - \psi_\star$. Recall the definition of $B_n(x,y)$ and observe that

$$P_\star^n \left( \forall_{x,y}, \psi_\star(x,y) \in B_n(x,y) \right) = 1 - P_\star^n \left( \exists_{x,y} \text{ s.t. } \psi_\star(x,y) \notin B_n(x,y) \right)$$
$$= 1 - P_\star^n \left( \exists_{x,y} \text{ s.t. } |f(x,y)| > w_n(x,y) \right),$$

which, by the inequality derived above and the definition of $w_n(x,y)$, is lower bounded by

$$\geq 1 - P_\star^n \left( \exists_{x,y} \text{ s.t. } \|f\|_{\Omega_n} \|\Lambda_{x,y}\|_{\Omega_n^{-1}} > w_n(x,y) \right)$$
$$= 1 - P_\star^n \left( \exists_{x,y} \|f\|_{\Omega_n} \|\Lambda_{x,y}\|_{\Omega_n^{-1}} > \|\Lambda_{x,y}\|_{\Omega_n^{-1}} \sqrt{\frac{\widehat{q}_{n,\alpha}}{n}} \right),$$

where the final equality plugs in the definition of $w_n(x,y)$. Since $\|\Lambda_{x,y}\|_{\Omega_n^{-1}}$ is positive and bounded, it cancels on both sides, and subsequently squaring both sides yields

$$= P_\star^n \left( \|f\|_{\Omega_n}^2 \leq \frac{\widehat{q}_{n,\alpha}}{n} \right) = P_\star^n \left( n \langle \Omega_n(\bar{\psi}_n - \psi_\star), \bar{\psi}_n - \psi_\star \rangle_{\mathcal{H}} \leq \widehat{q}_{n,\alpha} \right)$$
$$= P_\star^n \left( \psi_\star \in \mathcal{C}_n (\widehat{q}_{n,\alpha}) \right),$$

where the final equality follows directly from the definition of the confidence set $\mathcal{C}_n(\widehat{q}_{n,\alpha})$ (29).

Now, we have by supposition that the conditions of Theorem 3.1 hold, and $\Omega_n, \Omega_\star \in \mathscr{W}$ with $\|\Omega_n - \Omega_\star\|_{\text{op}} = o_p(1)$. From Proposition D.3, $\psi$ is pathwise differentiable at $P_\star$, and from Lemma 2.2, it has an EIF $\phi_\star \in L^2(P_\star; \mathcal{H})$ such that $\|\phi_\star\|_{L^2(P_\star;\mathcal{H})} > 0$.

Since $\widehat{P}_n^r$ serves as the initial estimate of $P_\star$ for each $r \in \{1,2\}$, conditions (i) and (ii) of Theorem 3.1 regarding the convergence rates of the nuisance parameters imply via Lemma E.2 that $\|\phi_n^r - \phi_\star\|_{L^2(P_\star;\mathcal{H})} = o_p(1)$ for each $r \in \{1,2\}$.

Thus, the conditions for Theorem 4 in Luedtke & Chung (2024) are satisfied, yielding $\widehat{q}_{n,\alpha} \xrightarrow{p} q_\alpha$. Consequently, taking the limit on both sides of the preceding display and applying Theorem 3 (i) from Luedtke & Chung (2024) yields

$$\lim_{n \to \infty} P_\star^n \left( \forall_{x,y}, \psi_\star(x,y) \in B_n(x,y) \right) \geq \lim_{n \to \infty} P_\star^n \left\{ \psi_\star \in \mathcal{C}_n (\widehat{q}_{n,\alpha}) \right\} = 1 - \alpha,$$

establishing the desired result. $\square$

## F.2. SKCDAgg Test

This appendix establishes the asymptotic validity of the SKCDAgg test introduced in Sec. 3.1. Throughout, the bandwidth grid is fixed and finite. For each $j \in [J]$, the kernel $\lambda^j = k^j \otimes \ell^j$ is bounded and characteristic, and has corresponding RKHS $\mathcal{H}^j$. We write $\psi^j : \mathcal{P} \to \mathcal{H}^j$ for the SCoDiTE parameter constructed using $\lambda^j$, $\phi_P^j$ for its EIF at $P$, and $\bar{\psi}_n^j$ for the corresponding cross-fitted one-step estimator. The nuisance estimators are allowed to depend on $j$.

We first collect the bandwidth-specific objects into the direct-sum Hilbert space $\mathcal{H}^\oplus := \bigoplus_{j=1}^{J} \mathcal{H}^j$, equipped with the inner product

$$\langle f, g \rangle_{\mathcal{H}^\oplus} := \sum_{j=1}^{J} \langle f^j, g^j \rangle_{\mathcal{H}^j}, \qquad f = (f^1, \ldots, f^J), \quad g = (g^1, \ldots, g^J).$$

Since each $\mathcal{H}^j$ is a real separable Hilbert space and $J < \infty$ is fixed, $\mathcal{H}^\oplus$ is also a real separable Hilbert space. Define a stacking of the parameter, estimator, and EIF by

$$\psi^\oplus(P) := \left( \psi^1(P), \ldots, \psi^J(P) \right), \qquad \bar{\psi}_n^\oplus := \left( \bar{\psi}_n^1, \ldots, \bar{\psi}_n^J \right), \qquad \phi_P^\oplus := \left( \phi_P^1, \ldots, \phi_P^J \right).$$

We use the shorthand $\psi_\star^\oplus := \psi^\oplus(P_\star)$ and $\phi_\star^\oplus := \phi_{P_\star}^\oplus$. Under the global null $H_0$ (2), $\psi_\star^j = 0$ for every $j \in [J]$, and therefore $\psi_\star^\oplus = 0$. We begin by proving some intermediate results.

### F.2.1. SUPPORTING LEMMAS

**Lemma F.1** (Joint weak convergence over the finite bandwidth grid)**.** *Suppose that, for each $j \in [J]$, the conditions of Theorem 3.1 hold for the kernel $\lambda^j$. Then*

1. *$\bar{\psi}_n^\oplus - \psi_\star^\oplus = \frac{1}{n} \sum_{i=1}^{n} \phi_\star^\oplus(Z_i) + o_p(n^{-1/2})$, and*
2. *$\sqrt{n} \left( \bar{\psi}_n^\oplus - \psi_\star^\oplus \right) \rightsquigarrow \mathbb{H}^\oplus$*

*in $\mathcal{H}^\oplus$, where $\mathbb{H}^\oplus = (\mathbb{H}^1, \ldots, \mathbb{H}^J)$ is a tight centered Gaussian random element satisfying $\langle \mathbb{H}^\oplus, h \rangle_{\mathcal{H}^\oplus} \sim \mathcal{N} \left( 0, \mathbb{E}_\star \left[ \langle \phi_\star^\oplus(Z), h \rangle_{\mathcal{H}^\oplus}^2 \right] \right)$ for every $h \in \mathcal{H}^\oplus$.*

*Proof.* For each $j \in [J]$, Theorem 3.1 applied with the kernel $\lambda^j$ yields

$$\bar{\psi}_n^j - \psi_\star^j = \frac{1}{n} \sum_{i=1}^{n} \phi_\star^j(Z_i) + r_n^j, \qquad \left\| r_n^j \right\|_{\mathcal{H}^j} = o_p(n^{-1/2}),$$

where $r_n^j$ denotes the sum of the corresponding remainder and drift terms. Stacking these expansions yields

$$\bar{\psi}_n^\oplus - \psi_\star^\oplus = \frac{1}{n} \sum_{i=1}^{n} \phi_\star^\oplus(Z_i) + r_n^\oplus, \qquad r_n^\oplus := (r_n^1, \ldots, r_n^J).$$

Now, since $J$ is fixed, $\|r_n^\oplus\|_{\mathcal{H}^\oplus}^2 = \sum_{j=1}^{J} \left\| r_n^j \right\|_{\mathcal{H}^j}^2 = o_p(n^{-1})$, and therefore $\|r_n^\oplus\|_{\mathcal{H}^\oplus} = o_p(n^{-1/2})$. Moreover, by Lemma 2.2 applied to each coordinate $j$, $\mathbb{E}_\star \|\phi_\star^\oplus(Z)\|_{\mathcal{H}^\oplus}^2 = \sum_{j=1}^{J} \mathbb{E}_\star \left\| \phi_\star^j(Z) \right\|_{\mathcal{H}^j}^2 < \infty$, so that $\phi_\star^\oplus \in L^2(P_\star; \mathcal{H}^\oplus)$. Then, since $\mathcal{H}^\oplus$ is a real separable Hilbert space, the Hilbert-space central limit theorem invoked in the proof of Theorem 3.1 yields

$$\frac{1}{\sqrt{n}} \sum_{i=1}^{n} \phi_\star^\oplus(Z_i) \rightsquigarrow \mathbb{H}^\oplus$$

in $\mathcal{H}^\oplus$. Applying Slutsky's lemma in $\mathcal{H}^\oplus$ concludes the proof. $\square$

We next formalize the bootstrap quantities used by the SKCDAgg test. Let $\xi_i^{(b)}$ denote the generic centered multinomial weight for the $b$th bootstrap draw, generated independently within each split as in Alg. 1. For an observation index $i$,

let $s(i) \in \{1, 2\}$ denote its split, $n_{s(i)} := |\mathcal{I}_{s(i)}|$, and $r(i) := 3 - s(i)$ the complementary training split. To match the split-weighted estimator in (9), let

$$\varphi_i^j := \frac{n}{2n_{s(i)}} \left( \psi_n^{j,r(i)} + \phi_n^{j,r(i)}(Z_i) \right)$$

denote the $j$th bandwidth-specific one-step summand, so that $\bar{\psi}_n^j = n^{-1} \sum_{i=1}^n \varphi_i^j$. Define

$$\Delta_n^{j,(b)} := \frac{1}{n} \sum_{i=1}^n \xi_i^{(b)} \varphi_i^j, \qquad T_n^{j,(b)} := n \left\langle \Omega_n^j \Delta_n^{j,(b)}, \Delta_n^{j,(b)} \right\rangle_{\mathcal{H}^j}. \tag{34}$$

The same splitwise resampling draws are used for every $j \in [J]$, so that the bootstrap preserves the dependence among the bandwidth-specific statistics. Because the centered multinomial weights sum to zero within each split, the plug-in component $\psi_n^{j,r(i)}$ cancels in $\Delta_n^{j,(b)}$, i.e.,

$$\sum_{i=1}^n \xi_i^{(b)} \frac{n}{2n_{s(i)}} \psi_n^{j,r(i)} = \sum_{s=1}^2 \frac{n}{2n_s} \psi_n^{j,3-s} \sum_{i \in \mathcal{I}_s} \xi_i^{(b)} = 0.$$

Thus the scaled perturbation satisfies

$$\sqrt{n} \Delta_n^{j,(b)} = \frac{1}{\sqrt{n}} \sum_{i=1}^n \xi_i^{(b)} \frac{n}{2n_{s(i)}} \phi_n^{j,r(i)}(Z_i).$$

This is the EIF-centered bootstrap quantity used in the proof below. When the two folds have equal size, the factor $n/(2n_{s(i)}) = 1$, so this coincides with the simpler analogous display in Alg. 1. For fixed $\tau \in (0, 1)$, let $\widehat{q}_{n,\tau}^j$ denote the exact conditional $(1 - \tau)$-quantile of $T_n^{j,(b)}$ given the original data. The empirical bootstrap with finite $B$ used in practice is the usual Monte Carlo approximation to this exact conditional quantile.

**Lemma F.2** (Coordinatewise bootstrap quantile consistency). *Suppose the conditions of Lemma F.1 hold. Further suppose that, for each $j \in [J]$, $\Omega_\star^j, \Omega_n^j \in \mathscr{W}$ and $\left\| \Omega_n^j - \Omega_\star^j \right\|_{\mathrm{op}} = o_p(1)$. Let $Q^j := \left\langle \Omega_\star^j \mathbb{H}^j, \mathbb{H}^j \right\rangle_{\mathcal{H}^j}$, and let $q_\tau^j$ denote the $(1 - \tau)$-quantile of $Q^j$. Then,*

$$\max_{j \in [J]} \left| \widehat{q}_{n,\tau}^j - q_\tau^j \right| \xrightarrow{p} 0.$$

*If, in addition, $q_\tau^j > 0$ for every $j \in [J]$, then $\min_{j \in [J]} \widehat{q}_{n,\tau}^j \xrightarrow{p} \min_{j \in [J]} q_\tau^j > 0$.*

*Proof.* Fix $j \in [J]$. Applying the same fixed-bandwidth bootstrap argument used in the proof of Theorem 3.3, now with the kernel $\lambda^j$ and operator limit $\Omega_\star^j$, gives conditional weak convergence, in probability, of

$$T_n^{j,(b)} = n \left\langle \Omega_n^j \Delta_n^{j,(b)}, \Delta_n^{j,(b)} \right\rangle_{\mathcal{H}^j}$$

to $Q^j$. The required consistency of the estimated EIF follows from Lemma E.2 applied to the $j$th bandwidth-specific EIF, using conditions (i) and (ii) of Theorem 3.1 for the kernel $\lambda^j$.

It remains only to justify that the limiting distribution is continuous at its quantile. By Lemma 2.2, $\phi_\star^j \in L^2(P_\star; \mathcal{H}^j)$ and $\left\| \phi_\star^j \right\|_{L^2(P_\star; \mathcal{H}^j)} > 0$, so the Gaussian element $\mathbb{H}^j$ is nondegenerate. Since $\Omega_\star^j \in \mathscr{W}$ is strictly positive definite, $Q^j$ is a nondegenerate Gaussian quadratic form. Hence, by a standard weighted chi-square representation of Gaussian quadratic forms,

$$Q^j \stackrel{d}{=} \sum_{\ell \geq 1} \lambda_{j\ell} Z_{j\ell}^2, \qquad Z_{j\ell} \stackrel{\mathrm{iid}}{\sim} \mathcal{N}(0, 1).$$

where $\lambda_{j\ell} \geq 0$, $\sum_{\ell \geq 1} \lambda_{j\ell} < \infty$, and at least one weight is positive. Conditioning on all terms except one with positive weight shows that $Q^j$ has no atoms. Hence, the distribution function of $Q^j$ is continuous at $q_\tau^j$. Consequently, applying Theorem 4 of Luedtke & Chung (2024) to the $j$th fixed-bandwidth quadratic statistics

$$T_n^j = n \left\langle \Omega_n^j \bar{\psi}_n^j, \bar{\psi}_n^j \right\rangle_{\mathcal{H}^j}, \qquad T_n^{j,(b)} = n \left\langle \Omega_n^j \Delta_n^{j,(b)}, \Delta_n^{j,(b)} \right\rangle_{\mathcal{H}^j},$$

with operator limit $\Omega_\star^j$ yields $\widehat{q}_{n,\tau}^j \xrightarrow{p} q_\tau^j$. This is precisely the fixed-bandwidth bootstrap quantile argument used in the proof of Theorem 3.3. Since $J < \infty$ is finite and fixed, the stated uniform convergence follows by a union bound. The final display follows from the continuity of the minimum map and the assumption $\min_{j \in [J]} q_\tau^j > 0$. $\qquad \square$

**Lemma F.3** (Bootstrap convergence of the aggregated statistic)**.** *Suppose the conditions of Lemma F.2 hold. Define*

$$
S_n^{\mathrm{Agg},(b)} := \max_{j \in [J]} \frac{T_n^{j,(b)}}{\widehat{q}_{n,\tau}^j}, \qquad S^{\mathrm{Agg}} := \max_{j \in [J]} \frac{\left\langle \Omega_\star^j \mathbb{H}^j, \mathbb{H}^j \right\rangle_{\mathcal{H}^j}}{q_\tau^j}.
$$

*If $q_\tau^j > 0$ for every $j \in [J]$, then $S_n^{\mathrm{Agg},(b)}$ converges weakly conditionally on the original data, in probability, to $S^{\mathrm{Agg}}$. Consequently, if $c_\alpha^{\mathrm{Agg}}$ denotes the exact $(1-\alpha)$-quantile of $S^{\mathrm{Agg}}$, and $\widehat{c}_{n,\alpha}^{\mathrm{Agg}}$ denotes the exact conditional $(1-\alpha)$-quantile of $S_n^{\mathrm{Agg},(b)}$, then*

$$
\widehat{c}_{n,\alpha}^{\mathrm{Agg}} \xrightarrow{p} c_\alpha^{\mathrm{Agg}}.
$$

*Proof.* Let $\phi_n^{\oplus,r} := \left( \phi_n^{1,r}, \ldots, \phi_n^{J,r} \right)$ and $\phi_\star^\oplus := \left( \phi_\star^1, \ldots, \phi_\star^J \right)$. By Lemma E.2 applied coordinatewise, and since $J < \infty$,

$$
\max_{r \in \{1,2\}} \left\| \phi_n^{\oplus,r} - \phi_\star^\oplus \right\|_{L^2(P_\star;\mathcal{H}^\oplus)}^2 = \max_{r \in \{1,2\}} \sum_{j=1}^J \left\| \phi_n^{j,r} - \phi_\star^j \right\|_{L^2(P_\star;\mathcal{H}^j)}^2
$$
$$
= o_p(1).
$$

Since $\mathcal{H}^\oplus$ is a real separable Hilbert space and $\phi_\star^\oplus \in L^2(P_\star;\mathcal{H}^\oplus)$, the same conditional bootstrap central limit argument used in the proof of Theorem 3.3, now carried out in $\mathcal{H}^\oplus$, yields conditional weak convergence, in probability, of

$$
\sqrt{n} \Delta_n^{\oplus,(b)} := \frac{1}{\sqrt{n}} \sum_{i=1}^n \xi_i^{(b)} \frac{n}{2 n_{s(i)}} \phi_n^{\oplus,r(i)}(Z_i)
$$

to $\mathbb{H}^\oplus$ in $\mathcal{H}^\oplus$. Since $\left\| \Omega_n^j - \Omega_\star^j \right\|_{\mathrm{op}} = o_p(1)$ as well for every $j$, we also have $\max_{j \in [J]} \left\| \Omega_n^j - \Omega_\star^j \right\|_{\mathrm{op}} = o_p(1)$. Therefore, by Slutsky's lemma and the continuous mapping theorem applied to the finite-dimensional map $(h^1, \ldots, h^J, \Omega^1, \ldots, \Omega^J) \mapsto \max_{j \in [J]} \left\langle \Omega^j h^j, h^j \right\rangle_{\mathcal{H}^j} / q_\tau^j$, we have:

$$
\max_{j \in [J]} \frac{n \left\langle \Omega_n^j \Delta_n^{j,(b)}, \Delta_n^{j,(b)} \right\rangle_{\mathcal{H}^j}}{q_\tau^j} \rightsquigarrow S^{\mathrm{Agg}}
$$

conditionally, in probability. Thus, Lemma F.2, together with another application of Slutsky's lemma, yields:

$$
S_n^{\mathrm{Agg},(b)} = \max_{j \in [J]} \frac{n \left\langle \Omega_n^j \Delta_n^{j,(b)}, \Delta_n^{j,(b)} \right\rangle_{\mathcal{H}^j}}{\widehat{q}_{n,\tau}^j} \rightsquigarrow S^{\mathrm{Agg}}
$$

conditionally, in probability.

Finally, the limiting distribution of $S^{\mathrm{Agg}}$ is continuous. Indeed, the proof of Lemma F.2 shows that each coordinate quadratic form $Q^j$ is atomless. Hence, for any $t \in \mathbb{R}$, $P_\star \left( S^{\mathrm{Agg}} = t \right) \leq \sum_{j=1}^J P_\star \left( Q^j = t q_\tau^j \right) = 0$, where $q_\tau^j > 0$ and $J < \infty$. The same bootstrap quantile consistency argument as in Lemma F.2 therefore yields $\widehat{c}_{n,\alpha}^{\mathrm{Agg}} \xrightarrow{p} c_\alpha^{\mathrm{Agg}}$ and concludes the proof. $\quad \square$

### F.2.2. PROOF OF PROPOSITION 3.5

**Proposition 3.5** (Validity of the SKCDAgg test)**.** *Suppose that the conditions of Thm. 3.1 hold for each $j \in [J]$, and $\Omega_n^j, \Omega_\star^j \in \mathcal{W}$ with $\left\| \Omega_n^j - \Omega_\star^j \right\|_{\mathrm{op}} = o_p(1)$ for each $j \in [J]$. Assume $q_\tau^j > 0$ for every $j \in [J]$. Then,*

1. $\lim_{n \to \infty} P_\star^n \left( S_n^{\mathrm{Agg}} > \widehat{c}_{n,\alpha}^{\mathrm{Agg}} \right) = \alpha$ *under $H_0$, and*
2. $\lim_{n \to \infty} P_\star^n \left( S_n^{\mathrm{Agg}} > \widehat{c}_{n,\alpha}^{\mathrm{Agg}} \right) = 1$ *if $\psi_\star^j \neq 0$ for some $j \in [J]$.*

*Proof.* We prove the two statements in turn.

*Statement 1*: Suppose $P_\star$ is such that the global null holds: $\psi_\star^j = 0$ for every $j \in [J]$. By Lemma F.1,

$$\sqrt{n}\bar{\psi}_n^\oplus \rightsquigarrow \mathbb{H}^\oplus$$

in $\mathcal{H}^\oplus$. Since $\max_{j \in [J]} \left\| \Omega_n^j - \Omega_\star^j \right\|_{\mathrm{op}} = o_p(1)$, Slutsky's lemma and the continuous mapping theorem yield

$$\left(T_n^1, \ldots, T_n^J\right) \rightsquigarrow \left(Q^1, \ldots, Q^J\right),$$

where

$$T_n^j = n \left\langle \Omega_n^j \bar{\psi}_n^j, \bar{\psi}_n^j \right\rangle_{\mathcal{H}^j}, \qquad Q^j = \left\langle \Omega_\star^j \mathbb{H}^j, \mathbb{H}^j \right\rangle_{\mathcal{H}^j}.$$

By Lemma F.2, $\max_{j \in [J]} \left| \widehat{q}_{n,\tau}^j - q_\tau^j \right| = o_p(1)$ and $\min_{j \in [J]} \widehat{q}_{n,\tau}^j$ is bounded away from zero in probability. Another application of Slutsky's lemma and the continuous mapping theorem therefore gives

$$S_n^{\mathrm{Agg}} = \max_{j \in [J]} \frac{T_n^j}{\widehat{q}_{n,\tau}^j} \rightsquigarrow S^{\mathrm{Agg}} := \max_{j \in [J]} \frac{Q^j}{q_\tau^j}.$$

By Lemma F.3,

$$\widehat{c}_{n,\alpha}^{\mathrm{Agg}} \xrightarrow{p} c_\alpha^{\mathrm{Agg}},$$

where $c_\alpha^{\mathrm{Agg}}$ is the exact $(1-\alpha)$-quantile of $S^{\mathrm{Agg}}$. Since the distribution function of $S^{\mathrm{Agg}}$ is continuous by the atomlessness argument in the proof of Lemma F.3, we obtain

$$\lim_{n \to \infty} P_\star^n \left(S_n^{\mathrm{Agg}} > \widehat{c}_{n,\alpha}^{\mathrm{Agg}}\right) = P_\star \left(S^{\mathrm{Agg}} > c_\alpha^{\mathrm{Agg}}\right) = \alpha,$$

establishing type 1 error control.

*Statement 2*: Now suppose $P_\star$ is a fixed alternative such that $\psi_\star^{j_0} \neq 0$ for some $j_0 \in [J]$. By Lemma F.1 applied coordinatewise and Prokhorov's theorem,

$$\bar{\psi}_n^{j_0} = \psi_\star^{j_0} + O_p(n^{-1/2}),$$

and hence, $\bar{\psi}_n^{j_0} \xrightarrow{p} \psi_\star^{j_0}$ in $\mathcal{H}^{j_0}$. The same fixed-alternative argument used in the proof of Theorem 3.3, applied to the $j_0$th coordinate with $\Omega_n = \Omega_n^{j_0}$ and $\Omega_\star = \Omega_\star^{j_0}$, yields

$$T_n^{j_0} = n \left\langle \Omega_n^{j_0} \bar{\psi}_n^{j_0}, \bar{\psi}_n^{j_0} \right\rangle_{\mathcal{H}^{j_0}} \xrightarrow{p} \infty.$$

Now, since $\widehat{q}_{n,\tau}^{j_0} \xrightarrow{p} q_\tau^{m_0} > 0$ by Lemma F.2, we have that

$$S_n^{\mathrm{Agg}} \geq \frac{T_n^{j_0}}{\widehat{q}_{n,\tau}^{j_0}} \xrightarrow{p} \infty.$$

On the other hand, the bootstrap convergence in Lemma F.3 implies $\widehat{c}_{n,\alpha}^{\mathrm{Agg}} = O_p(1)$. Thus, it follows that

$$P_\star^n \left(S_n^{\mathrm{Agg}} > \widehat{c}_{n,\alpha}^{\mathrm{Agg}}\right) \to 1,$$

establishing consistency against fixed alternatives. $\qquad\square$

## G. Test Statistics in Closed-Form

This appendix derives the explicit algebraic expressions for our test statistics that can be used in the SKCD test (Alg. 1). We establish this first for $T_n^{\mathrm{MMD}}$ and then for $T_n^{\mathrm{Wald}}$.

Recall that $\mathbf{K}, \mathbf{L} \in \mathbb{R}^{n \times n}$ are Gram matrices corresponding to kernels $k$ and $\ell$. Since we assume $\bar{\psi}_n$ to be a linear combination of feature maps, it lies in the following finite-dimensional subspace of $\mathcal{H}$:

$$\mathcal{F}_n := \mathrm{span}\left\{\Lambda_{x_i, y_j} : i, j \in [n]\right\}. \tag{35}$$

## G.1. MMD Formulation

We begin with the MMD statistic ($T_n^{\text{MMD}}$), which corresponds to the choice $\Omega_n = I$. The derivation relies on expressing the cross-fitted estimator coefficients in matrix form.

### G.1.1. SUPPORTING LEMMA

Recall that $\boldsymbol{\beta}_a^r(x) \in \mathbb{R}^n$ denotes the vector of coefficients for the outcome model $\theta_{n,a}^r(x) = \sum_j [\boldsymbol{\beta}_a^r(x)]_j \Lambda_{x,y_j}$ such that $[\boldsymbol{\beta}_a^r(x)]_j = 0$ for any observation where $j \notin \mathcal{I}^r$ or $a_j \neq a$.

**Lemma G.1.** *For any index $i$, let $s(i) \in \{1, 2\}$ be the split containing $i$, and $r(i) = 3 - s(i)$ be the complement. Construct $\mathbf{E} \in \mathbb{R}^{n \times n}$ using:*

$$[\mathbf{E}]_{i,j} := \begin{cases} \frac{1}{2n_{s(i)}} \left( [\boldsymbol{\beta}_1^{r(i)}(x_i)]_j - [\boldsymbol{\beta}_0^{r(i)}(x_i)]_j \right) & \text{if } j \neq i \\ 0 & \text{otherwise.} \end{cases} \tag{36}$$

*Define $e_{ij} := [\mathbf{E}]_{i,j}$ and $c_{ij} := [\mathbf{C}]_{i,j}$, where $\mathbf{C} \in \mathbb{R}^{n \times n}$ is constructed using* (15). *Then, the cross-fitted plugin estimator $\psi_n = \sum_{i,j \in [n]} e_{ij} \Lambda_{x_i, y_j}$ and the cross-fitted one-step estimator $\bar{\psi}_n = \sum_{i,j \in [n]} c_{ij} \Lambda_{x_i, y_j}$.*

*Proof.* For $r \in \{1, 2\}$, set $s = 3 - r$. Observe that the cross-fitted plug-in estimator is given by

$$\psi_n = \frac{1}{2} \sum_{r=1}^{2} \psi(\widehat{P}_n^r) = \frac{1}{2} \sum_{r=1}^{2} \mathbb{E}_{P_n^s} \left[ \theta_{n,1}^r(X) - \theta_{n,0}^r(X) \right]$$

$$= \frac{1}{2} \sum_{r=1}^{2} \frac{1}{n_s} \sum_{i \in \mathcal{I}^s} \left[ \sum_{j \in \mathcal{I}^r, a_j = 1} [\boldsymbol{\beta}_1^r(x_i)]_j \Lambda_{x_i, y_j} - \sum_{j \in \mathcal{I}^r, a_j = 0} [\boldsymbol{\beta}_0^r(x_i)]_j \Lambda_{x_i, y_j} \right]$$

$$= \sum_{r=1}^{2} \sum_{i \in \mathcal{I}^s} \left[ \frac{1}{2n_s} \left( \sum_{j \in \mathcal{I}^r, a_j = 1} [\boldsymbol{\beta}_1^r(x_i)]_j \Lambda_{x_i, y_j} - \sum_{j \in \mathcal{I}^r, a_j = 0} [\boldsymbol{\beta}_0^r(x_i)]_j \Lambda_{x_i, y_j} \right) \right]$$

$$= \sum_{i,j \in [n]} \left[ \sum_{r=1}^{2} \mathbb{1}_{\{i \in \mathcal{I}^s, j \in \mathcal{I}^r\}} \left( \mathbb{1}_{\{a_j = 1\}} \frac{[\boldsymbol{\beta}_1^r(x_i)]_j}{2n_s} - \mathbb{1}_{\{a_j = 0\}} \frac{[\boldsymbol{\beta}_0^r(x_i)]_j}{2n_s} \right) \right] \Lambda_{x_i, y_j}.$$

Let $e_{i,j} := [\mathbf{E}]_{i,j}$. Comparing the terms in the preceding display with (36) yields $\psi_n = \sum_{i,j \in [n]} e_{ij} \Lambda_{x_i, y_j}$. We now use the same steps to derive the form of the $c_{ij} := [\mathbf{C}]_{i,j}$ for the cross-fitted one-step estimator. Observe that

$$\bar{\psi}_n = \frac{1}{2} \sum_{r=1}^{2} \mathbb{E}_{P_n^s} \left[ \left( \frac{A}{\pi_n^r(X)} - \frac{1-A}{1-\pi_n^r(X)} \right) (\Lambda_{X,Y} - \theta_{n,A}^r(X)) + \theta_{n,1}^r(X) - \theta_{n,0}^r(X) \right]$$

$$= \sum_{r=1}^{2} \sum_{i \in \mathcal{I}^s} \left[ \frac{1}{2n_s} \left\{ \left( \frac{a_i}{\pi_n^r(x_i)} - \frac{1-a_i}{1-\pi_n^r(x_i)} \right) \Lambda_{x_i, y_i} + \left( 1 - \frac{a_i}{\pi_n^r(x_i)} \right) \theta_{n,1}^r(x_i) - \left( 1 - \frac{1-a_i}{1-\pi_n^r(x_i)} \right) \theta_{n,0}^r(x_i) \right\} \right]$$

$$= \sum_{i,j \in [n]} \left[ \sum_{r=1}^{2} \mathbb{1}_{\{i \in \mathcal{I}^s, j = i\}} \frac{1}{2n_s} \left( \frac{a_i}{\pi_n^r(x_i)} - \frac{1-a_i}{1-\pi_n^r(x_i)} \right) \right.$$

$$\left. + \sum_{r=1}^{2} \mathbb{1}_{\{i \in \mathcal{I}^s, j \in \mathcal{I}^r\}} \left\{ \left( 1 - \frac{a_i}{\pi_n^r(x_i)} \right) \mathbb{1}_{\{a_j = 1\}} \frac{[\boldsymbol{\beta}_1^r(x_i)]_j}{2n_s} - \left( 1 - \frac{1-a_i}{1-\pi_n^r(x_i)} \right) \mathbb{1}_{\{a_j = 0\}} \frac{[\boldsymbol{\beta}_0^r(x_i)]_j}{2n_s} \right\} \right] \Lambda_{x_i, y_j}.$$

Thus, comparing the terms in the above display with (15) in the same vein as the derivation of (36) concludes the proof. $\square$

### G.1.2. PROOF OF PROPOSITION 3.6

**Proposition 3.6** (Closed-form MMD statistic from Alg. 1). *If $\Omega_n = I$ and $\mathbf{C}$ is as constructed using* (15), *then the squared MMD test statistic from Alg. 1 takes the form $T_n^{\text{MMD}} := n \left\| \bar{\psi}_n \right\|_{\mathcal{H}}^2 = n \langle \mathbf{C}, \mathbf{KCL} \rangle_{\text{F}}$.*

*Proof.* In Lemma G.1, we show that $\bar{\psi}_n = \sum_{i,j} c_{ij} \Lambda_{x_i,y_j} \in \mathcal{F}_n$ with $c_{ij} := [\mathbf{C}]_{i,j}$ constructed using (15). It follows, by the linearity of the inner product and the reproducing property of $\Lambda$, that

$$\|\bar{\psi}_n\|_{\mathcal{H}}^2 = \left\langle \sum_{i,j} c_{ij} \Lambda_{x_i,y_j}, \sum_{i',j'} c_{i',j'} \Lambda_{x_{i'},y_{j'}} \right\rangle_{\mathcal{H}} = \sum_{i,j} \sum_{i',j'} c_{ij} c_{i',j'} \Lambda_{x_i,y_j}(x_{i'}, y_{j'}) = \sum_{i,i',j,j'} c_{ij} k(x_i, x_{i'}) \ell(y_j, y_{j'}) c_{i'j'}$$

$$= \text{vec}\left(\mathbf{C}^\top\right)^\top (\mathbf{K} \otimes \mathbf{L}) \text{vec}\left(\mathbf{C}^\top\right) = \text{vec}\left(\mathbf{C}^\top\right)^\top \text{vec}\left((\mathbf{KCL})^\top\right) = \langle \mathbf{C}, \mathbf{KCL} \rangle_{\text{F}},$$

where $\text{vec}\left(\mathbf{C}^\top\right)$ is the row-wise vectorization of $\mathbf{C}$, $\mathbf{K}_{i,i'} = k(x_i, x_{i'})$ and $\mathbf{L}_{j,j'} = \ell(y_j, y_{j'})$ by definition, and the penultimate equality uses the vec trick (Roth, 1934).

$\square$

### G.2. Wald-type Formulation

The Wald statistic incorporates the inverse covariance operator $\Omega_n = [(1-\varepsilon)\Sigma_n + \varepsilon I]^{-1}$ defined in Eq. 16.

#### G.2.1. SUPPORTING LEMMAS

The following two lemmas show that $\Omega_n$ satisfies the consistency properties required for the SKCD test to retain asymptotic validity.

**Lemma G.2** (Luedtke & Chung, 2024 Lemma S12). *Fix $\varepsilon > 0$. Suppose that $\|\phi_\star\|_{L^2(P_\star;\mathcal{H})} < \infty$ and $\|\phi_n^r - \phi_\star\|_{L^2(P_\star;\mathcal{H})} = o_p(1)$ for each $r \in \{1, 2\}$. Let $\Sigma_\star : h \mapsto \mathbb{E}_\star\left[\langle\phi_\star(Z), h\rangle_{\mathcal{H}} \phi_\star(Z)\right]$. If $\Sigma_n : h \mapsto \frac{1}{2} \sum_{r=1}^2 \mathbb{E}_{P_n^s}[\langle\phi_n^r(Z), h\rangle_{\mathcal{H}} \phi_n^r(Z)]$ (where $s = 3 - r$), then $\|\Sigma_n - \Sigma_\star\|_{\text{op}} = o_p(1)$.*

**Lemma G.3.** *Suppose that the conditions of Lemma G.2 are satisfied, and $\Sigma_\star$ and $\Sigma_n$ are as defined therein. Let $\Omega_\star := [(1-\varepsilon)\Sigma_\star + \varepsilon I]^{-1}$ and $\Omega_n := [(1-\varepsilon)\Sigma_n + \varepsilon I]^{-1}$. We have that $\|\Omega_n - \Omega_\star\|_{\text{op}} = o_p(1)$.*

*Proof.* See Appendix D of Luedtke & Chung (2024), specifically the discussion preceding the statement of Lemma S12 therein. Their argument relies on the Lipschitz-ness of the map $\Sigma \mapsto [(1-\varepsilon)\Sigma + \varepsilon I]^{-1}$ and the continuous mapping theorem. $\square$

The following lemma shows that $\Omega_n$ maps elements of $\mathcal{F}_n$ (as defined in Eq. 35) into $\mathcal{F}_n$, which will allow us to compute our test statistics using the finite-dimensional Gram matrix $\mathbf{G} := \mathbf{K} \otimes \mathbf{L}$.

**Lemma G.4.** *Let $\Sigma_n : \mathcal{H} \to \mathcal{H}$ be as defined in Lemma G.2, and let $\mathcal{F}_n \subseteq \mathcal{H}$ be the finite-dimensional subspace defined in Eq. 35. Then, for all $\varepsilon \in (0, 1]$, it holds that $[(1-\varepsilon)\Sigma_n + \varepsilon I]^{-1} \in \mathscr{W}_{\text{inv}}$ and is such that $[(1-\varepsilon)\Sigma_n + \varepsilon I]^{-1}(f) \in \mathcal{F}_n$ for all $f \in \mathcal{F}_n$.*

*Proof.* First, we show that $\Sigma_n(\mathcal{F}_n) \subseteq \mathcal{F}_n$. For any $i, j \in [n]$, let $r \in \{1, 2\}$ be the fold containing $j$ and set $s = 3 - r$. Recalling the matrices $\mathbf{C}$ and $\mathbf{E}$ from Lemma G.1, we have:

$$\phi_n^r(z_i) = \left(\frac{a_i}{\pi_n^r(x_i)} - \frac{1-a_i}{1-\pi_n^r(x_i)}\right)\left\{\Lambda_{x_i,y_i} - \theta_{n,a_i}^r(x_i)\right\} + \theta_{n,1}^r(x_i) - \theta_{n,0}^r(x_i) - \mathbb{E}_{P_n^s}\left[\theta_{n,1}^r(X) - \theta_{n,0}^r(X)\right]$$

$$= \left(\frac{a_i}{\pi_n^r(x_i)} - \frac{1-a_i}{1-\pi_n^r(x_i)}\right)\left\{\Lambda_{x_i,y_i} - \theta_{n,a_i}^r(x_i)\right\} + \theta_{n,1}^r(x_i) - \theta_{n,0}^r(x_i) - \frac{1}{n_s} \sum_{i'' \in \mathcal{I}^s}\left[\theta_{n,1}^r(x_{i''}) - \theta_{n,0}^r(x_{i''})\right]$$

$$= \sum_{j \in [n]} \mathbb{1}_{\{i \in \mathcal{I}^s\}} 2n_s c_{ij} \Lambda_{x_i,y_j} - \sum_{i'' \in \mathcal{I}^s} \sum_{j'' \in \mathcal{I}^r} 2e_{i''j''} \Lambda_{x_{i''},y_{j''}}. \tag{37}$$

Since the indices $i, j, i'', j''$ all belong to $[n]$, it follows that $\phi_n^r(Z_k) \in \mathcal{F}_n$ for all $r \in \{1, 2\}$ and $k \in \mathcal{I}_s$. Observe that for any $h \in \mathcal{H}$, $\Sigma_n(h) = \frac{1}{2} \sum_{r=1}^2 \frac{1}{n_s} \sum_{k \in \mathcal{I}_s} \langle\phi_n^r(Z_k), h\rangle_{\mathcal{H}} \phi_n^r(Z_k)$. As this is just a linear combination of the cross-fitted EIF evaluations, which lie in $\mathcal{F}_n$, it follows that $\Sigma_n(h) \in \mathcal{F}_n$ for all $h \in \mathcal{H}$. Restricting the input $h$ to the subspace $\mathcal{F}_n$, trivially yields that $\Sigma_n(\mathcal{F}_n) \subseteq \mathcal{F}_n$.

Let $\Upsilon_n := (1 - \varepsilon)\Sigma_n + \varepsilon I$. Since $\Sigma_n$ and the identity operator $I$ are continuous and self-adjoint on $\mathcal{H}$, $\Upsilon_n$ is also a continuous, self-adjoint operator acting on the entire space $\mathcal{H}$. For any $h \in \mathcal{H}$, we have:

$$\langle \Upsilon_n h, h \rangle_{\mathcal{H}} = (1 - \varepsilon) \langle \Sigma_n h, h \rangle_{\mathcal{H}} + \varepsilon \|h\|_{\mathcal{H}}^2 \geq \varepsilon \|h\|_{\mathcal{H}}^2$$

where the inequality follows from $\varepsilon > 0$ and the positive semi-definiteness of $\Sigma_n$. Hence, for all $h \in \mathcal{H}$ such that $h \neq 0$, it holds that $\langle \Upsilon_n h, h \rangle_{\mathcal{H}} > 0$, i.e., $\Upsilon_n$ is strictly positive and bounded below. Consequently, $\Upsilon_n$ is boundedly invertible on $\mathcal{H}$. Since $\Upsilon_n$ and its inverse are bounded, self-adjoint, and strictly positive, we have that $\Upsilon_n^{-1} \in \mathcal{W}_{\mathrm{inv}}$.

Next, we establish that $\mathcal{F}_n$ is invariant under $\Upsilon_n^{-1}$. $\Sigma_n(\mathcal{F}_n) \subseteq \mathcal{F}_n$ immediately implies $\Upsilon_n(\mathcal{F}_n) \subseteq \mathcal{F}_n$. Let $\Upsilon_n|_{\mathcal{F}_n} : \mathcal{F}_n \to \mathcal{F}_n$ denote the restriction of $\Upsilon_n$ to the finite-dimensional subspace $\mathcal{F}_n$. Since $\Upsilon_n$ is strictly positive on all of $\mathcal{H}$, its restriction $\Upsilon_n|_{\mathcal{F}_n}$ is injective. By the invertible matrix theorem, any injective linear operator mapping a finite-dimensional space to itself is invertible, and therefore surjective. Thus, $\Upsilon_n(\mathcal{F}_n) = \mathcal{F}_n$. Consequently, for any $f \in \mathcal{F}_n$, its unique pre-image under $\Upsilon_n$ must also lie in $\mathcal{F}_n$. $\qquad \square$

The restriction of $(1 - \varepsilon)\Sigma_n + \varepsilon I$ to the finite-dimensional space $\mathcal{F}_n$ can be represented by an $n^2 \times n^2$ matrix; however, inverting this matrix using standard software would require $\mathcal{O}(n^6)$ operations, infeasible even for moderate $n$. In the following lemma, we reduce this complexity to $\mathcal{O}(n^3)$ by exploiting the low-rank structure of $\Sigma_n$ and applying the Woodbury matrix identity.

**Lemma G.5.** *Let $\Omega_n$ be as defined in Eq. 16, and let $\mathbf{T}$ and $\mathbf{U}$ be constructed using Eqs. 17 and 18. Define the $(2n + 4) \times (2n + 4)$ matrix*

$$\widetilde{\Omega}_n := \frac{1}{\varepsilon}\mathbf{I} - \frac{1 - \varepsilon}{\varepsilon}\mathbf{T}\left(\varepsilon\mathbf{I} + (1 - \varepsilon)\mathbf{U}^\top\mathbf{T}\right)^{-1}\mathbf{U}^\top.$$

*Then, $\Omega_n(\bar{\psi}_n) = \sum_{i,j} b_{ij}\Lambda_{x_i,y_j}$, where the coefficients $b_{ij}$ form a matrix $\mathbf{B} \in \mathbb{R}^{n \times n}$ satisfying $\mathbf{b}^\top := [\mathrm{vec}(\mathbf{B}^\top)]^\top = \mathbf{c}^\top\widetilde{\Omega}_n$.*

*Proof.* For any $i, j \in [n]$, let $r \in \{1, 2\}$ be the fold containing $j$ and set $s = 3 - r$. Recalling the matrices $\mathbf{C}$ and $\mathbf{E}$ from the proof of Lemma G.1 together with Eq. 37, we have:

$$\phi_n^r(z_i) = \sum_{j \in [n]} \mathbb{1}_{\{i \in \mathcal{I}^s\}} 2n_s c_{ij}\Lambda_{x_i,y_j} - \sum_{i'' \in \mathcal{I}^s} \sum_{j'' \in \mathcal{I}^r} 2e_{i''j''}\Lambda_{x_{i''},y_{j''}}$$

$$= \sum_{j \in [n]} \mathbb{1}_{\{i \in \mathcal{I}^s\}} 2n_s c_{ij}\Lambda_{x_i,y_j} - \sum_{i'' \in \mathcal{I}^s} \sum_{j'' \in [n]} 2e_{i''j''}\Lambda_{x_{i''},y_{j''}},$$

where we use that $e_{ij} = 0$ whenever $i, j \in \mathcal{I}^s$. Therefore,

$$\frac{\phi_n^r(z_i)}{\sqrt{2n_s}} = \sum_{j \in [n]} \mathbb{1}_{\{i \in \mathcal{I}^s\}}\sqrt{2n_s}c_{ij}\Lambda_{x_i,y_j} - \sum_{i'' \in \mathcal{I}^s} \sum_{j'' \in [n]} \mathbb{1}_{\{i'' \in \mathcal{I}^s\}}\sqrt{\frac{2}{n_s}}e_{i''j''}\Lambda_{x_{i''},y_{j''}}.$$

Now, recall the definitions of $d_{ij}^s := [\mathbf{D}^s]_{i,j} = \mathbb{1}_{\{i \in \mathcal{I}^s\}}\sqrt{2n_s}[\mathbf{C}]_{ij}$ and $v_{ij}^s := [\mathbf{V}^s]_{i,j} = \mathbb{1}_{\{i \in \mathcal{I}^s\}}\sqrt{2/n_s}[\mathbf{E}]_{ij}$ from (17). Thus, we have:

$$\frac{\phi_n^r(z_i)}{\sqrt{2n_s}} = \sum_{j \in [n]} d_{ij}^s\Lambda_{x_i,y_j} - \sum_{i'' \in \mathcal{I}^s} \sum_{j'' \in [n]} v_{i''j''}^s\Lambda_{x_{i''},y_{j''}}. \tag{38}$$

Note that the expression above is fold-specific—i.e., if $i \in \mathcal{I}^s$, the plug-in mean (second term in the above display) must correspond to data-fold $\mathcal{D}^s$. Also, since the indicator is preserved under squaring, $\mathbb{1}_{\{i \in \mathcal{I}^s\}}d_{ij}^s = d_{ij}^s$ and $\mathbb{1}_{\{i \in \mathcal{I}^s\}}v_{ij}^s = v_{ij}^s$.

The definition of $\Omega_n$ in Eq. 16 matches that in Lemma G.3. Consequently, Lemma G.4 yields $\Omega_n(\bar{\psi}_n) \in \mathcal{F}_n$, implying that there exists $\mathbf{B} \in \mathbb{R}^{n \times n}$, with $[\mathbf{B}]_{i,j} =: b_{ij}$, such that $\Omega_n(\bar{\psi}_n) = \sum_{i,j} b_{ij}\Lambda_{x_i,y_j}$. Now, let $f := \bar{\psi}_n$ and $g := \Omega_n(\bar{\psi}_n)$. Thus, it follows that

$$[(1 - \varepsilon)\Sigma_n + \varepsilon I]^{-1}(f) = g \implies f = (1 - \varepsilon)\Sigma_n(g) + \varepsilon g = (1 - \varepsilon)\left(\frac{1}{2}\sum_{r=1}^2 \mathbb{E}_{P_n^s}\left[\langle g, \phi_n^r(Z)\rangle_{\mathcal{H}}\phi_n^r(Z)\right]\right) + \varepsilon g. \tag{39}$$

Now, observe that

$$
\begin{aligned}
\frac{1}{2}\sum_{r=1}^{2}\mathbb{E}_{P_n^s}\left[\langle g,\phi_n^r(Z)\rangle_{\mathcal{H}}\,\phi_n^r(Z)\right] &= \frac{1}{2}\sum_{r=1}^{2}\mathbb{E}_{P_n^s}\left[\left\langle\sum_{i,j\in[n]}b_{ij}\Lambda_{(x_i,y_j)},\phi_n^r(Z)\right\rangle_{\mathcal{H}}\phi_n^r(Z)\right] \\
&= \frac{1}{2}\sum_{r=1}^{2}\mathbb{E}_{P_n^s}\left[\sum_{i,j\in[n]}b_{ij}\phi_n^r(Z)(x_i,y_j)\,\phi_n^r(Z)\right] \\
&= \frac{1}{2}\sum_{r=1}^{2}\frac{1}{n_s}\sum_{i\in\mathcal{I}^s}\left[\sum_{i',j'\in[n]}b_{i'j'}\phi_n^r(z_i)(x_{i'},y_{j'})\,\phi_n^r(z_i)\right] \\
&= \sum_{r=1}^{2}\sum_{i\in\mathcal{I}^s}\left[\sum_{i',j'\in[n]}b_{i'j'}\frac{\phi_n^r(z_i)}{\sqrt{2n_s}}(x_{i'},y_{j'})\,\frac{\phi_n^r(z_i)}{\sqrt{2n_s}}\right] \qquad (\dagger)
\end{aligned}
$$

Using (38), we have:

$$
\begin{aligned}
(\dagger) = \sum_{r=1}^{2}\sum_{i\in\mathcal{I}^s}\Bigg(\sum_{i',j'\in[n]}b_{i'j'}&\left[\sum_{j''\in[n]}d_{ij''}^s\Lambda_{(x_i,y_{j''})}(x_{i'},y_{j'})-\sum_{i''\in\mathcal{I}^s}\sum_{j''\in[n]}v_{i''j''}^s\Lambda_{(x_{i''},y_{j''})}(x_{i'},y_{j'})\right] \\
&\times\left[\sum_{j\in[n]}d_{ij}^s\Lambda_{(x_i,y_j)}-\sum_{i\in\mathcal{I}^s}\sum_{j\in[n]}v_{ij}^s\Lambda_{(x_i,y_j)}\right]\Bigg).
\end{aligned}
$$

Let $q_i^s-\tilde{q}^s := \sum_{i',j'\in[n]}b_{i'j'}\left[\sum_{j''\in[n]}d_{ij''}^s\Lambda_{(x_i,y_{j''})}(x_{i'},y_{j'})-\sum_{i''\in\mathcal{I}^s}\sum_{j''\in[n]}v_{i''j''}^s\Lambda_{(x_{i''},y_{j''})}(x_{i'},y_{j'})\right]$. We have,

$$
\begin{aligned}
(\dagger) &= \sum_{r=1}^{2}\sum_{i\in\mathcal{I}^s}\left((q_i^s-\tilde{q}^s)\left[\sum_{j\in[n]}d_{ij}^s\Lambda_{(x_i,y_j)}-\sum_{i,j\in[n]}v_{ij}^s\Lambda_{(x_i,y_j)}\right]\right) \\
&= \sum_{r=1}^{2}\left(\left[\sum_{i\in\mathcal{I}^s}\sum_{j\in[n]}(q_i^s-\tilde{q}^s)d_{ij}^s\Lambda_{(x_i,y_j)}-\sum_{i''\in\mathcal{I}^s}\sum_{j''\in[n]}\left\{\sum_{i\in\mathcal{I}^s}(q_i^s-\tilde{q}^s)\right\}v_{i''j''}^s\Lambda_{(x_{i''},y_{j''})}\right]\right) \\
&= \sum_{r=1}^{2}\left(\left[\sum_{i\in\mathcal{I}^s}\sum_{j\in[n]}(q_i^s-\tilde{q}^s)d_{ij}^s\Lambda_{(x_i,y_j)}-\sum_{i\in\mathcal{I}^s}\sum_{j\in[n]}\left\{\sum_{i''\in\mathcal{I}^s}q_{i''}^s-n_s\tilde{q}^s\right\}v_{ij}^s\Lambda_{(x_i,y_j)}\right]\right) \\
&= \sum_{r=1}^{2}\left(\left[\sum_{i\in\mathcal{I}^s}\sum_{j\in[n]}\mathbb{1}_{\{i\in\mathcal{I}^s\}}(q_i^s-\tilde{q}^s)d_{ij}^s\Lambda_{(x_i,y_j)}-\sum_{i\in\mathcal{I}^s}\sum_{j\in[n]}\left\{\sum_{i''\in\mathcal{I}^s}q_{i''}^s-n_s\tilde{q}^s\right\}\mathbb{1}_{\{i\in\mathcal{I}^s\}}v_{ij}^s\Lambda_{(x_i,y_j)}\right]\right) \\
&= \sum_{r=1}^{2}\left(\left[\sum_{i,j\in[n]}(q_i^s-\tilde{q}^s)d_{ij}^s\Lambda_{(x_i,y_j)}-\sum_{i,j\in[n]}\left\{\sum_{i''\in\mathcal{I}^s}q_{i''}^s-n_s\tilde{q}^s\right\}v_{ij}^s\Lambda_{(x_i,y_j)}\right]\right) \\
&= \sum_{i,j\in[n]}\left(\sum_{r=1}^{2}\left[(q_i^s-\tilde{q}^s)d_{ij}^s-\left\{\sum_{i''\in\mathcal{I}^s}q_{i''}^s-n_s\tilde{q}^s\right\}v_{ij}^s\right]\right)\Lambda_{(x_i,y_j)}
\end{aligned}
$$

Recall from (17) that $w_{ij}^s := [\mathbf{W}^s]_{i,j} = [\mathbf{D}^s - n_s\mathbf{V}^s]_{i,j} = d_{ij}^s - n_s v_{ij}^s$. It follows that $\sum_{i''\in\mathcal{I}^s}(q_i^s-\tilde{q}^s) =$

$\sum_{i',j'\in[n]} b_{i'j'} \left[ \sum_{i''\in\mathcal{I}^s} \sum_{j''\in[n]} w^s_{i''j''} \Lambda_{(x_{i''},y_{j''})}(x_{i'},y_{j'}) \right]$. The preceding display thus rewrites as

$$
(\dagger) = \sum_{i,j\in[n]} \left( \sum_{r=1}^{2} \left[ \left\{ \sum_{i',j'\in[n]} b_{i'j'} \left[ \sum_{j''\in[n]} d^s_{ij''} \Lambda_{(x_i,y_{j''})}(x_{i'},y_{j'}) - \sum_{i''\in\mathcal{I}^s} \sum_{j''\in[n]} v^s_{i''j''} \Lambda_{(x_{i''},y_{j''})}(x_{i'},y_{j'}) \right] \right\} d^s_{ij} \right. \right.
$$
$$
\left. \left. - \left\{ \sum_{i',j'\in[n]} b_{i'j'} \left[ \sum_{i''\in\mathcal{I}^s} \sum_{j''\in[n]} w^s_{i''j''} \Lambda_{(x_{i''},y_{j''})}(x_{i'},y_{j'}) \right] \right\} v^s_{ij} \right] \right) \Lambda_{(x_i,y_j)}
$$
$$
= \sum_{i,j\in[n]} \left[ \sum_{r=1}^{2} \sum_{i',j'\in[n]} b_{i'j'} \left\{ \mathbb{1}_{\{i\in\mathcal{I}^s\}} \left( \sum_{j''\in[n]} d^s_{ij''} \Lambda_{(x_i,y_{j''})}(x_{i'},y_{j'}) - \sum_{i''\in\mathcal{I}^s} \sum_{j''\in[n]} v^s_{i''j''} \Lambda_{(x_{i''},y_{j''})}(x_{i'},y_{j'}) \right) d^s_{ij} \right. \right.
$$
$$
\left. \left. - \mathbb{1}_{\{i\in\mathcal{I}^s\}} \left( \sum_{i''\in\mathcal{I}^s} \sum_{j''\in[n]} w^s_{i''j''} \Lambda_{(x_{i''},y_{j''})}(x_{i'},y_{j'}) \right) v^s_{ij} \right\} \right] \Lambda_{(x_i,y_j)}
$$
$$
= \sum_{i,j\in[n]} \left[ \sum_{r=1}^{2} \sum_{i',j'\in[n]} b_{i'j'} \left\{ \left( \sum_{j''\in[n]} d^s_{ij''} \Lambda_{(x_i,y_{j''})}(x_{i'},y_{j'}) - \sum_{i'',j''\in[n]} v^s_{i''j''} \Lambda_{(x_{i''},y_{j''})}(x_{i'},y_{j'}) \right) d^s_{ij} \right. \right.
$$
$$
\left. \left. - \left( \sum_{i'',j''\in[n]} w^s_{i''j''} \Lambda_{(x_{i''},y_{j''})}(x_{i'},y_{j'}) \right) v^s_{ij} \right\} \right] \Lambda_{(x_i,y_j)}
$$
$$
= \sum_{i,j\in[n]} \left[ \sum_{i',j'\in[n]} b_{i'j'} \left\{ \sum_{r=1}^{2} \left( \sum_{j''\in[n]} d^s_{ij''} \Lambda_{(x_i,y_{j''})}(x_{i'},y_{j'}) - \sum_{i'',j''\in[n]} v^s_{i''j''} \Lambda_{(x_{i''},y_{j''})}(x_{i'},y_{j'}) \right) d^s_{ij} \right. \right.
$$
$$
\left. \left. - \sum_{r=1}^{2} \left( \sum_{i'',j''\in[n]} w^s_{i''j''} \Lambda_{(x_{i''},y_{j''})}(x_{i'},y_{j'}) \right) v^s_{ij} \right\} \right] \Lambda_{(x_i,y_j)}.
$$

Consequently, we have from (39) that

$$
\sum_{i,j\in[n]} c_{ij} \Lambda_{x_i,y_j}
$$
$$
= \sum_{i,j\in[n]} \left[ (1-\varepsilon) \sum_{i',j'\in[n]} b_{i'j'} \left\{ \sum_{r=1}^{2} \left( \sum_{j''\in[n]} d^s_{ij''} \Lambda_{(x_i,y_{j''})}(x_{i'},y_{j'}) - \sum_{i'',j''\in[n]} v^s_{i''j''} \Lambda_{(x_{i''},y_{j''})}(x_{i'},y_{j'}) \right) d^s_{ij} \right. \right.
$$
$$
\left. \left. - \sum_{r=1}^{2} \left( \sum_{i'',j''\in[n]} w^s_{i''j''} \Lambda_{(x_{i''},y_{j''})}(x_{i'},y_{j'}) \right) v^s_{ij} \right\} + \varepsilon b_{ij} \right] \Lambda_{(x_i,y_j)}.
$$

This implies, under the condition that the points $x_i$ and $y_i$ in the dataset $\mathscr{D}$ are unique, that for all $i,j\in[n]$,

$$
c_{ij} = (1-\varepsilon) \sum_{i',j'\in[n]} b_{i'j'} \left\{ \sum_{r=1}^{2} \left( \underbrace{\sum_{j''\in[n]} d^s_{ij''} \Lambda_{(x_i,y_{j''})}(x_{i'},y_{j'})}_{\text{Term I}} - \underbrace{\sum_{i'',j''\in[n]} v^s_{i''j''} \Lambda_{(x_{i''},y_{j''})}(x_{i'},y_{j'})}_{\text{Term II}} \right) d^s_{ij} \right.
$$
$$
\left. - \sum_{r=1}^{2} \left( \underbrace{\sum_{i'',j''\in[n]} w^s_{i''j''} \Lambda_{(x_{i''},y_{j''})}(x_{i'},y_{j'})}_{\text{Term III}} \right) v^s_{ij} \right\} + \varepsilon b_{ij}. \tag{$\ddagger$}
$$

Recall that $\mathbf{v}^s := \mathrm{vec}\left(\mathbf{V}^{s\,\top}\right)$ and $\mathbf{w}^s := \mathrm{vec}\left(\mathbf{W}^{s\,\top}\right)$ are the row-wise vectorizations of $\mathbf{V}^s$ and $\mathbf{W}^s$. It is then easy to see that Terms II and III can be expressed as the $(i'j')^{\text{th}}$ elements of $\mathbf{G}\mathbf{v}^s$ and $\mathbf{G}\mathbf{w}^s$ respectively.

Term I is more complicated because it involves a summation over $j''$ for a *fixed* $i$. In the vectorized space $\mathbb{R}^{n^2}$, the vector corresponding to fixing $x_i$ and summing over weighted $y_{j''}$ can be written using the canonical basis vector $\tilde{e}_i$ as $\tilde{e}_i \otimes (\mathbf{D}^{s\,\top}\tilde{e}_i)$. Thus, Term I can be expressed as the $(i'j')^{\text{th}}$ element of $\mathbf{G}\left(\tilde{e}_i \otimes \left[\mathbf{D}^{s\,\top}\tilde{e}_i\right]\right)$.

Now, observe that in the curly braces of Eq. (‡), Term I (for a fixed index $i$) is multiplied by $d_{ij}^s$ on the right for the same $i$. With some abuse of notation, let $\tilde{e}_.$ denote that $\tilde{e}_i$ is adaptively chosen to be consistent with index $i$ of the right-multiplying $d_{ij}^s$. Then, using the face-splitting (row-wise Kronecker) product (denoted by $\bullet$), it holds that

$$\mathbf{G}\left(\tilde{e}_. \otimes \left[\mathbf{D}^{s\,\top}\tilde{e}_.\right]\right)\mathbf{d}^{s\,\top} = \mathbf{G}\mathrm{diag}\left(\mathbf{d}^s\right)\left(\mathbf{I}_n \otimes \mathbf{1}_n\mathbf{1}_n^\top\right)\mathrm{diag}\left(\mathbf{d}^s\right)$$
$$= \mathbf{G}\mathrm{diag}\left(\mathbf{d}^s\right)\left(\mathbf{I}_n \otimes \mathbf{1}_n\right)\left(\mathbf{I}_n \otimes \mathbf{1}_n\right)^\top\mathrm{diag}\left(\mathbf{d}^s\right)^\top = \mathbf{G}(\mathbf{I}_n \bullet \mathbf{D}^s)^\top(\mathbf{I}_n \bullet \mathbf{D}^s).$$

Recall that $\mathbf{S}^s := (\mathbf{I}_n \bullet \mathbf{D}^s)^\top$. Subsequently, (‡) rewrites as

$$\mathbf{c}^\top = \mathbf{b}^\top\left[\varepsilon\mathbf{I} + (1-\varepsilon)\mathbf{G}\left(\sum_{r=1}^2\left(\mathbf{S}^s\mathbf{S}^{s\,\top} - \mathbf{v}^s\mathbf{d}^{s\,\top} - \mathbf{w}^s\mathbf{v}^{s\,\top}\right)\right)\right]$$
$$\implies \mathbf{b}^\top = \mathbf{c}^\top\left[\varepsilon\mathbf{I} + (1-\varepsilon)\mathbf{G}\left(\sum_{r=1}^2\left(\mathbf{S}^s\mathbf{S}^{s\,\top} - \mathbf{v}^s\mathbf{d}^{s\,\top} - \mathbf{w}^s\mathbf{v}^{s\,\top}\right)\right)\right]^{-1}.$$

Recalling the definitions of $\mathbf{T}$ and $\mathbf{U}$ from (18), we can simplify this to

$$\mathbf{b}^\top = \mathbf{c}^\top\left[\varepsilon\mathbf{I} + (1-\varepsilon)\,\mathbf{T}\mathbf{U}^\top\right]^{-1}. \tag{40}$$

Then we have, by the Kailath variant of Woodbury's identity (Petersen et al., 2008, 3.2.3), that

$$\mathbf{b}^\top = \mathbf{c}^\top\left[\frac{1}{\varepsilon}\mathbf{I} - \frac{1-\varepsilon}{\varepsilon}\mathbf{T}\left(\varepsilon\mathbf{I} + (1-\varepsilon)\mathbf{U}^\top\mathbf{T}\right)^{-1}\mathbf{U}^\top\right], \tag{41}$$

and therefore, by the definition of $\widetilde{\Omega}_n$, that $\mathbf{b}^\top = \mathbf{c}^\top\widetilde{\Omega}_n$. $\square$

### G.2.2. PROOF OF PROPOSITION 3.7

**Proposition 3.7** (Closed-form Wald-type statistic from Alg. 1). *If $\Omega_n$ is as in (16) and $\mathbf{c} := \mathrm{vec}(\mathbf{C}^\top)$ is constructed from (15), then the Wald-type statistic from Alg. 1 can be computed in $\mathcal{O}(n^3)$ operations as*

$$T_n^{\mathrm{Wald}} := n\left\langle\Omega_n(\bar{\psi}_n), \bar{\psi}_n\right\rangle_{\mathcal{H}} = \frac{n}{\varepsilon}\left\langle\mathbf{C}, \mathbf{KCL}\right\rangle_F$$
$$- \frac{n(1-\varepsilon)}{\varepsilon}\mathbf{c}^\top\mathbf{T}\left(\varepsilon\mathbf{I} + (1-\varepsilon)\mathbf{U}^\top\mathbf{T}\right)^{-1}\mathbf{U}^\top\mathbf{G}\mathbf{c}.$$

*Proof.* Let $\mathcal{F}_n$ be the finite subspace defined in (35). We have from Proposition 3.6 that $\bar{\psi}_n \in \mathcal{F}_n$, and from Lemma G.1, that $\bar{\psi}_n = \sum_{i,j\in[n]} c_{ij}\Lambda_{x_i,y_j}$ with $c_{ij} := [\mathbf{C}]_{i,j}$ as defined in Eq. 15.

Consequently, for $\Omega_n$ the regularized inverse of the covariance operator, Lemma G.4 yields $\Omega_n(\bar{\psi}_n) \in \mathcal{F}_n$, which implies that there exists $\mathbf{B} \in \mathbb{R}^{n\times n}$, with $[\mathbf{B}]_{i,j} =: b_{ij}$, such that $\Omega_n(\bar{\psi}_n) = \sum_{i,j} b_{ij}\Lambda_{x_i,y_j}$. Let $\mathbf{b} := \mathrm{vec}\left(\mathbf{B}^\top\right)$. We show in Lemma G.5 that $\mathbf{b}^\top = \mathbf{c}^\top\widetilde{\Omega}_n$, where $\widetilde{\Omega}_n = \frac{1}{\varepsilon}\mathbf{I} - \frac{1-\varepsilon}{\varepsilon}\mathbf{T}\left(\varepsilon\mathbf{I} + (1-\varepsilon)\mathbf{U}^\top\mathbf{T}\right)^{-1}\mathbf{U}^\top$ with $\mathbf{T}$ and $\mathbf{U}$ constructed using Eqs. 17 and 18.

It then follows using the same arguments as in Proposition 3.6, and by Lemma G.5, that

$$\left\langle\Omega_n(\bar{\psi}_n), \bar{\psi}_n\right\rangle_{\mathcal{H}} = \mathbf{b}^\top\mathbf{G}\mathbf{c} = \mathbf{c}^\top\widetilde{\Omega}_n\mathbf{G}\mathbf{c} = \frac{1}{\varepsilon}\mathbf{c}^\top\mathbf{G}\mathbf{c} - \frac{1-\varepsilon}{\varepsilon}\mathbf{c}^\top\mathbf{T}\left(\varepsilon\mathbf{I} + (1-\varepsilon)\mathbf{U}^\top\mathbf{T}\right)^{-1}\mathbf{U}^\top\mathbf{G}\mathbf{c}$$
$$= \frac{1}{\varepsilon}\left\langle\mathbf{C}, \mathbf{KCL}\right\rangle_F - \frac{1-\varepsilon}{\varepsilon}\mathbf{c}^\top\mathbf{T}\left(\varepsilon\mathbf{I} + (1-\varepsilon)\mathbf{U}^\top\mathbf{T}\right)^{-1}\mathbf{U}^\top\mathbf{G}\mathbf{c}.$$

Now, let '$\circ$' denote the Hadamard product and '$*$' the Khatri-Rao (column-wise Kronecker) product. It is evident from (18) that computing the terms $\mathbf{c}^\top \mathbf{T}$, $\mathbf{U}^\top \mathbf{T}$, and $\mathbf{U}^\top \mathbf{G}\mathbf{c}$ involves terms of the following three types (letting $s, \bar{s}$ take values in $\{1, 2\}$ independently, and letting $\mathbf{x}, \mathbf{y} \in \mathbb{R}^{n^2}$ be arbitrary vectors):

$$
\begin{aligned}
\mathbf{x}^\top \mathbf{G} \mathbf{y} &= \langle \mathbf{X}, \mathbf{K}\mathbf{Y}\mathbf{L} \rangle_{\mathrm{F}} \in \mathbb{R}, \\
\mathbf{y}^\top \mathbf{G} \mathbf{S}^{\bar{s}} &= \left( \mathbf{S}^{\bar{s}\,\top} \mathbf{G}\mathbf{y} \right)^\top = \left[ (\mathbf{I}_n \bullet \mathbf{D}^{\bar{s}})\mathrm{vec}\left( \mathbf{L}\mathbf{Y}^\top \mathbf{K} \right) \right]^\top = \left[ \left( \mathbf{D}^{\bar{s}} \circ \mathbf{K}\mathbf{Y}\mathbf{L} \right) \mathbf{1}_n \right]^\top \in \mathbb{R}^{1\times n}, \text{ and} \\
\mathbf{S}^{s\,\top} \mathbf{G} \mathbf{S}^{\bar{s}} &= (\mathbf{I}_n \bullet \mathbf{D}^s)(\mathbf{K} \otimes \mathbf{L})\left( \mathbf{I}_n * \mathbf{D}^{\bar{s}\,\top} \right) = (\mathbf{I}_n \bullet \mathbf{D}^s)\left( \mathbf{K} * \mathbf{L}\mathbf{D}^{\bar{s}\,\top} \right) = \mathbf{K} \circ \mathbf{D}^s \mathbf{L}\mathbf{D}^{\bar{s}\,\top} \in \mathbb{R}^{n \times n},
\end{aligned}
\tag{42}
$$

where the first equation holds by the same steps as in the proof of Proposition 3.6, the second follows directly from the definitions of the face-splitting and hadamard products, and the third holds by Slyusar (1999) Eq. 3, Rao (1970) Lemma A1, and Slyusar (1999) Theorem 1.

Using these expressions allows us to avoid ever having to store or manipulate $n^2 \times n^2$ matrices or $n^2$-dimensional vectors directly. Thus, since the matrix inversion $\left( \varepsilon \mathbf{I} + (1 - \varepsilon) \mathbf{U}^\top \mathbf{T} \right)^{-1}$ in $\mathbb{R}^{(2n+4) \times (2n+4)}$ becomes the dominating operation, we can compute the Wald-type statistic with a worst-case computational complexity of $\mathcal{O}(n^3)$. Note that there is no need to save $\mathbf{G}$ in memory.

Subsequently, we have from Lemma 2.2 that $\phi_\star \in L^2(P_\star; \mathcal{H})$, and by supposition, that the conditions of Theorem 3.1 hold. Further, $\Omega_\star \in \mathscr{W}$ and $\Omega_n \in \mathscr{W}$ by way of Lemma G.4 due to the respective definitions of $\Sigma_\star$ and $\Sigma_n$.

Since $\widehat{P}_n^r$ serves as the initial estimate of $P_\star$ for each $r \in \{1, 2\}$, conditions (i) and (ii) of Theorem 3.1 regarding the convergence rates of the nuisance parameters imply via Lemma E.2 that $\|\phi_n^r - \phi_\star\|_{L^2(P_\star; \mathcal{H})} = o_p(1)$ for each $r \in \{1, 2\}$. Hence, Lemma G.2, and consequently, Lemma G.3 yield that $\|\Omega_n - \Omega_\star\|_{\mathrm{op}} = o_p(1)$.

Therefore, the conditions of Theorem 3.3 are satisfied, and we have the desired guarantees for the test of $\psi_\star = 0$, using Algorithm 1 with $T_n \equiv T_n^{\mathrm{Wald}}$. $\qquad\square$

### G.2.3. HEURISTIC FOR CHOOSING $\varepsilon$

The regularization parameter $\varepsilon \in (0, 1)$ controls the trade-off between the empirical covariance $\Sigma_n$ and the identity matrix $I$ and stabilizes the inversion of the covariance operator $\varepsilon I + (1 - \varepsilon)\Sigma_n$. Specifically, the eigenvalues $\lambda_i$ of the empirical covariance $\Sigma_n$ are transformed in the inverse operator as $\lambda_i^{\mathrm{inv}} = 1/((1 - \varepsilon)\lambda_i + \varepsilon)$. Thus, for the regularization to be effective, $\varepsilon$ must be comparable in magnitude to the to the spectral scale of $(1 - \varepsilon)\Sigma_n$.

However, fixing $\varepsilon$ to a universal constant is a poor choice, since the scaling of the Gram matrices depends arbitrarily on the kernel choice and the actual data points: if the kernel values are large, $\Sigma_n$ dominates, and we lose the well-conditioning due to regularization; if they are small, $I$ dominates, and we do not account for the signal.

To determine a stable choice for $\varepsilon$, we can therefore consider the total "magnitude" of the signal captured by $\Sigma_n$. Observe that, as shown in the proof of Lemma G.5, the restriction of $(1 - \varepsilon)\Sigma_n + \varepsilon I$ to the finite-dimensional space $\mathcal{F}_n$ can be represented by an $n^2 \times n^2$ matrix $\varepsilon \mathbf{I} + (1 - \varepsilon)\mathbf{T}\mathbf{U}^\top$. Moreover, while $\Sigma_n$ acts on a subspace of dimension $n^2 \times n^2$, its rank is bounded by $2n + 4$, and it converges in operator norm to a fixed limit $\Sigma_\star$ by Lemma G.3.

Consequently, the trace of the empirical covariance operator can be computed via its matrix representation. By the cyclic property of the trace, $\mathrm{tr}(\Sigma_n) = \mathrm{tr}(\mathbf{T}\mathbf{U}^\top) = \mathrm{tr}(\mathbf{U}^\top \mathbf{T})$. This sums only the non-zero eigenvalues $\sum_{i=1}^{2n+4} \lambda_i$, which converges to the total variance of the EIF, $\mathbb{E}_\star[\|\phi_\star(Z)\|_{\mathcal{H}}^2]$, meaning it is $O_p(1)$. This stability arises because the matrices $\mathbf{C}$ and $\mathbf{E}$ used in the construction $\mathbf{T}$ and $\mathbf{U}$ are already correctly scaled.

This motivates a heuristic: set $\varepsilon$ so that $\varepsilon/(1 - \varepsilon) \propto \mathrm{tr}(\mathbf{T}\mathbf{U}^\top)$. We can introduce a hyperparameter $\gamma > 0$ to define the desired balance between these two terms. Setting the identity weight to be $\gamma$ times the covariance weight yields the condition $\varepsilon/(1 - \varepsilon) = \gamma\mathrm{tr}(\mathbf{T}\mathbf{U}^\top)$, and, by the cyclic property of the trace, solving this for $\varepsilon$ yields

$$
\varepsilon = \frac{\gamma\mathrm{tr}(\mathbf{T}\mathbf{U}^\top)}{1 + \gamma\mathrm{tr}(\mathbf{T}\mathbf{U}^\top)} = \frac{\gamma\mathrm{tr}(\mathbf{U}^\top \mathbf{T})}{1 + \gamma\mathrm{tr}(\mathbf{U}^\top \mathbf{T})}.
\tag{43}
$$

The hyperparameter $\gamma$ can be interpreted as our "trust" in the covariance estimate. Equal weighting ($\gamma = 1$) assigns a 50% balance to the regularization and empirical covariance terms. Larger values ($\gamma > 1$) pull the estimate towards the identity (and therefore, towards the MMD statistic). This may be useful for smaller sample sizes where the estimate $\Sigma_n$ may be

ill-conditioned or noisy. Smaller values ($\gamma < 1$) rely more heavily on the covariance estimate, which may be appropriate when $n$ is large and/or $\Sigma_n$ is well-estimated. We empirically analyze the impact of varying $\gamma$ in further detail in App. J.1.

## H. Fast SKCD Test Implementation

### H.1. Exact Closed-Form

Naively implementing the bootstrap in Alg. 1 would result in a computational complexity of $\mathcal{O}(Bn^3)$. In this appendix, we show that when using the closed-form test statsitics given in Section 3.2, we can amortize expensive operations to achieve a complexity of $\mathcal{O}(n^3 + Bn^2)$. We provide this optimized implementation in Alg. 2, and describe it in more detail below.

We can construct the coefficient matrix $\mathbf{C}$ defined in Eq. 15 using a block structure induced by the sample splits. Assume the data are ordered such that indices $1, \ldots, n_1$ correspond to fold $\mathcal{I}_1$ and $n_1 + 1, \ldots, n$ correspond to fold $\mathcal{I}_2$. We write $\mathbf{C}$ as a $2 \times 2$ block matrix:

$$\mathbf{C} = \begin{pmatrix} \mathbf{C}_{11} & \mathbf{C}_{12} \\ \mathbf{C}_{21} & \mathbf{C}_{22} \end{pmatrix}. \tag{44}$$

The diagonal blocks $\mathbf{C}_{ss} \in \mathbb{R}^{n_s \times n_s}$ for $s \in \{1, 2\}$ are diagonal matrices containing the inverse propensity weights:

$$\mathbf{C}_{ss} = \frac{1}{2n_s} \mathrm{diag} \left( \frac{a_i}{\pi_n^r(x_i)} - \frac{1 - a_i}{1 - \pi_n^r(x_i)} \right)_{i \in \mathcal{I}_s}, \tag{45}$$

where $r = 3 - s$ denotes the complementary fold (i.e., nuisances are fit on fold $r$ and evaluated on fold $s$).

The off-diagonal blocks $\mathbf{C}_{sr} \in \mathbb{R}^{n_s \times n_r}$ (where $r \neq s$) encode the augmentation term. These are constructed as the row-scaled product:

$$\mathbf{C}_{sr} = \frac{1}{2n_s} \mathbf{\Gamma}_s \mathbf{H}_{sr}, \tag{46}$$

where $\mathbf{\Gamma}_s \in \mathbb{R}^{n_s \times n_s}$ is the diagonal matrix of augmentation coefficients with entries

$$[\mathbf{\Gamma}_s]_{ii} = \begin{cases} 1 - \dfrac{1}{\pi_n^r(x_i)} & \text{if } a_i = 1, \\ \dfrac{1}{1 - \pi_n^r(x_i)} - 1 & \text{if } a_i = 0, \end{cases} \tag{47}$$

and $\mathbf{H}_{sr} \in \mathbb{R}^{n_s \times n_r}$ is a matrix whose entry $[\mathbf{H}_{sr}]_{ij}$ weights the training observation $j \in \mathcal{I}_r$ on the prediction for test point $i \in \mathcal{I}_s$. For kernel ridge regression with regularization $\lambda > 0$, this matrix takes the form $\mathbf{H}_{sr} = \mathbf{K}_{\mathcal{I}_s, \mathcal{I}_r} (\mathbf{K}_{\mathcal{I}_r, \mathcal{I}_r} + \lambda \mathbf{I})^{-1}$, though our approach accommodates any regression method that produces such weights.

Similarly, the auxiliary matrix $\mathbf{E}$ used in the Wald-type statistic has block structure is:

$$\mathbf{E} = \begin{pmatrix} \mathbf{0} & \mathbf{E}_{12} \\ \mathbf{E}_{21} & \mathbf{0} \end{pmatrix}, \tag{48}$$

where $\mathbf{E}_{12} = \frac{1}{2n_1} \mathbf{H}_{12}$ and $\mathbf{E}_{21} = -\frac{1}{2n_2} \mathbf{H}_{21}$.

Recall from Alg. 1 that our one-step estimator is an empirical mean such that $\bar{\psi}_n = \frac{1}{n} \sum_{k=1}^n \varphi_k$, and the bootstrap replicate is the weighted sum $\Delta_n^{(b)} = \frac{1}{n} \sum_{k=1}^n \xi_k \varphi_k$. Moreover, Lemma G.1 establishes that $\bar{\psi}_n = \sum_{i,j} [\mathbf{C}]_{ij} \Lambda_{x_i, y_j}$. Inspecting the construction of $\mathbf{C}$ derived in Eqs. 44 to 47, it is evident that the $i$-th row of $\mathbf{C}$ collects the terms specific to the observation $Z_i$. Thus, by the linearity of the map $w \mapsto \sum_k w_k \varphi_k$, the coefficient matrix $\mathbf{C}^{(b)}$ corresponding to the weighted sum $\Delta_n^{(b)}$ is given by row-scaling $\mathbf{C}$ by the multipliers $\xi = [\xi_1, \ldots, \xi_n]^\top$, i.e.

$$\mathbf{C}^{(b)} := \mathrm{diag}\,(\xi)\, \mathbf{C}. \tag{49}$$

Now, using Proposition 3.6, the MMD bootstrap statistic is $T_n^{(b), \mathrm{MMD}} = n \left\langle \mathbf{C}^{(b)}, \mathbf{K}\mathbf{C}^{(b)}\mathbf{L} \right\rangle_F$. Substituting (49) and applying the cyclic property of the trace, we have

$$T_n^{(b), \mathrm{MMD}} = n\,\mathrm{tr}\left( (\mathrm{diag}\,(\xi)\,\mathbf{C})^\top \mathbf{K} (\mathrm{diag}\,(\xi)\,\mathbf{C})\mathbf{L} \right) = n\,\mathrm{tr}\left( \mathbf{C}^\top \mathrm{diag}\,(\xi)\, \mathbf{K} \mathrm{diag}\,(\xi)\, \mathbf{C}\mathbf{L} \right)$$
$$= n\,\mathrm{tr}\left( \mathrm{diag}\,(\xi)\, \mathbf{K} \mathrm{diag}\,(\xi)\, (\mathbf{C}\mathbf{L}\mathbf{C}^\top) \right).$$

Using the identity $\mathrm{diag}\,(\xi)\,\mathbf{A}\mathrm{diag}\,(\xi) = \mathbf{A}\circ(\xi\xi^\top)$ for any matrix $\mathbf{A}$ yields

$$T_n^{(b),\mathrm{MMD}} = n\mathrm{tr}\left((\mathbf{K}\circ(\xi\xi^\top))(\mathbf{CLC}^\top)\right) = n\sum_{i,j}[\mathbf{K}]_{ij}\xi_i\xi_j[\mathbf{CLC}^\top]_{ji} = n\xi^\top\left(\mathbf{K}\circ(\mathbf{CLC}^\top)\right)\xi. \tag{50}$$

Let $\mathbf{M} := \mathbf{K}\circ(\mathbf{CLC}^\top)$. Since $\mathbf{M}$ is independent of $b$, it can be computed before the bootstrap loop.

Now, from Proposition 3.7, the Wald-type statistic involves a correction term based on the covariance. The statistic takes the form:

$$T_n^{\mathrm{Wald}} = n\left(\frac{1}{\varepsilon}\langle\mathbf{C},\mathbf{KCL}\rangle_F - \frac{1-\varepsilon}{\varepsilon}\mathbf{c}^\top\mathbf{TZ}^{-1}\mathbf{U}^\top\mathbf{Gc}\right),$$

where $\mathbf{c} = \mathrm{vec}(\mathbf{C}^\top)$ and $\mathbf{G} = \mathbf{L}\otimes\mathbf{K}$, and $\mathbf{Z} = \varepsilon\mathbf{I} + (1-\varepsilon)\mathbf{U}^\top\mathbf{T}$ is the regularized covariance matrix, fixed for the observed data.

For the bootstrap replicate with $\mathbf{C}^{(b)} = \mathrm{diag}(\xi)\mathbf{C}$, we have

$$\mathbf{c}^{(b)} = \mathrm{vec}((\mathbf{C}^{(b)})^\top) = \mathrm{vec}(\mathbf{C}^\top\mathrm{diag}(\xi)) = (\mathrm{diag}(\xi)\otimes\mathbf{I}_n)\mathbf{c}. \tag{51}$$

Now, observe that $\mathbf{U}^\top\mathbf{Gc}^{(b)}$ and $\mathbf{T}^\top\mathbf{c}^{(b)}$ are linear functions of $\xi$. We can derive these by analyzing the block structure of $\mathbf{U}$ and $\mathbf{T}$.

Recall from (18) that $\mathbf{U}$ contains block matrices $\mathbf{S}^s = (\mathbf{I}_n\bullet\mathbf{D}^s)^\top$ for $s\in\{1,2\}$, where $\bullet$ denotes the face-splitting product and $\mathbf{D}^s$ are the scaled coefficient matrices.

The $i$-th column of $\mathbf{S}^s$ corresponds to $\mathrm{vec}(d_i^s\tilde{e}_i^\top)$, where $d_i^s$ is the $i$-th row of $\mathbf{D}^s$, and $\tilde{e}_i$ is the $i$-th canonical basis vector. Using (51), the $i$-th component of $\mathbf{S}^{s\top}\mathbf{Gc}^{(b)}$ is

$$[(\mathbf{S}^s)^\top\mathbf{Gc}^{(b)}]_i = \mathrm{vec}(d_i^s\tilde{e}_i^\top)^\top\mathbf{G}(\mathrm{diag}(\xi)\otimes\mathbf{I}_n)\mathbf{c} = \mathrm{tr}\left(\tilde{e}_i(d_i^s)^\top\mathbf{K}\mathrm{diag}(\xi)\mathbf{CL}\right) = \tilde{e}_i^\top\mathbf{K}\mathrm{diag}(\xi)\mathbf{CL}d_i^s$$

$$= \sum_{j=1}^n\xi_j[\mathbf{K}]_{ij}[(\mathbf{CL})(\mathbf{D}^s)^\top]_{ji},$$

inspecting which allows us to define a matrix $\mathbf{H}_{\mathbf{S}^s}\in\mathbb{R}^{n\times n} : \mathbf{H}_{\mathbf{S}^s} := \mathbf{K}\circ\left(\mathbf{D}^s(\mathbf{CL})^\top\right)$ such that

$$(\mathbf{S}^s)^\top\mathbf{Gc}^{(b)} = \mathbf{H}_{\mathbf{S}^s}\xi. \tag{52}$$

Now, we consider the last 4 columns of $\mathbf{T}$ corresponding to $\mathbf{v}^s$ and $\mathbf{w}^s$. For a generic matrix $\mathbf{F}\in\{\mathbf{V}^s,\mathbf{W}^s\}$, we have

$$(\mathbf{G}\mathrm{vec}(\mathbf{F}^\top))^\top\mathbf{c}^{(b)} = \mathrm{vec}(\mathbf{F}^\top)^\top\mathbf{G}(\mathrm{diag}(\xi)\otimes\mathbf{I}_n)\mathbf{c} = \mathrm{tr}\left(\mathbf{F}^\top\mathbf{K}\mathrm{diag}(\xi)\mathbf{CL}\right) = \mathrm{tr}\left(\mathrm{diag}(\xi)\mathbf{CLF}^\top\mathbf{K}\right)$$

$$= \xi^\top\mathbf{h_F}, \tag{53}$$

where $\mathbf{h_F} := \mathrm{diag}(\mathbf{CLF}^\top\mathbf{K})\in\mathbb{R}^n$.

Similarly, $\mathbf{U}^\top\mathbf{T}$ can be constructed using (42) and the projection matrices $\mathbf{H}_{\mathbf{S}^s}$ and $\mathbf{h_F}$. Note that all three of these operators are independent of $b$, and therefore can be computed outside the bootstrap loop. In fact, to avoid having to invert $\mathbf{Z} = \varepsilon\mathbf{I} + (1-\varepsilon)\mathbf{U}^\top\mathbf{T}\in\mathbb{R}^{2n+4\times2n+4}$ in the loop, we can precompute its LU factorization.

Thus, within the bootstrap loop, using (18), (52) and (53), the expression for $\mathbf{U}^\top\mathbf{Gc}^{(b)}\in\mathbb{R}^{2n+4}$ is given by

$$\mathbf{U}^\top\mathbf{Gc}^{(b)} = \left[(\mathbf{H}_{\mathbf{S}^1}\xi)^\top \quad (\mathbf{H}_{\mathbf{S}^2}\xi)^\top \quad -\mathbf{1}^\top\mathbf{H}_{\mathbf{S}^1}\xi \quad -\mathbf{1}^\top\mathbf{H}_{\mathbf{S}^2}\xi \quad -\mathbf{h}_{\mathbf{V}^1}^\top\xi \quad -\mathbf{h}_{\mathbf{V}^2}^\top\xi\right]^\top, \tag{54}$$

and similarly,

$$\mathbf{T}^\top\mathbf{c}^{(b)} = \left[(\mathbf{H}_{\mathbf{S}^1}\xi)^\top \quad (\mathbf{H}_{\mathbf{S}^2}\xi)^\top \quad \mathbf{h}_{\mathbf{V}^1}^\top\xi \quad \mathbf{h}_{\mathbf{V}^2}^\top\xi \quad \mathbf{h}_{\mathbf{W}^1}^\top\xi \quad \mathbf{h}_{\mathbf{W}^2}^\top\xi\right]^\top. \tag{55}$$

Thus, each bootstrap Wald-type statistic can be computed using (50) and the preceding two displays as

$$T_n^{(b),\mathrm{Wald}} = n\left(\frac{1}{\varepsilon}(\xi^\top\mathbf{M}\xi) - \frac{1-\varepsilon}{\varepsilon}(\mathbf{T}^\top\mathbf{c}^{(b)})^\top\mathbf{z}^{(b)}\right), \tag{56}$$

where $\mathbf{z}^{(b)}$ solves the linear system $\mathbf{Zz}^{(b)} = \mathbf{U}^\top\mathbf{Gc}^{(b)}$. Since we precomputed the LU factorization of $\mathbf{Z}\in\mathbb{R}^{(2n+4)\times(2n+4)}$, each bootstrap iteration requires only an $\mathcal{O}(n^2)$ forward/back substitution operation to obtain $\mathbf{z}^{(b)}$.

---

**Algorithm 2** Fast SKCD test using closed-form test statistics

---

**Input:** Data $\mathscr{D} = \{Z_i\}_{i=1}^n$, kernels $k, \ell$, level $\alpha$, bootstrap replicates $B$, regularization $\varepsilon$, test type $\in \{\text{MMD}, \text{WALD}\}$.
**Output:** Rejection decision.

1: Fit cross-fitted propensity models $\pi_n^1, \pi_n^2$ on folds $\mathcal{I}_1, \mathcal{I}_2$.
2: Construct $\mathbf{C}, \mathbf{E}$ via (44)–(48).
3: $\mathbf{K} \leftarrow [k(x_i, x_j)]_{i,j}; \quad \mathbf{L} \leftarrow [\ell(y_i, y_j)]_{i,j}; \quad \mathbf{M} \leftarrow \mathbf{K} \circ (\mathbf{C}\mathbf{L}\mathbf{C}^\top)$
4: **if** test type $=$ WALD **then**
5: $\quad$ Compute $\mathbf{H}_{\mathbf{S}^s}, \mathbf{h}_\mathbf{F}$ via (52) and (53); $\quad$ Construct $\mathbf{U}^\top \mathbf{T}$ using (42), and $\mathbf{H}_{\mathbf{S}^s}$ and $\mathbf{h}_\mathbf{F}$
6: $\quad$ LU-factorize $\mathbf{Z} = \varepsilon \mathbf{I}_{2n+4} + (1 - \varepsilon) \mathbf{U}^\top \mathbf{T}$ .
7: **end if**
8: $T_n \leftarrow n \cdot \mathbf{1}^\top \mathbf{M} \mathbf{1}$ {MMD}
9: **if** test type $=$ WALD **then**
10: $\quad$ Construct $\mathbf{U}^\top \mathbf{G}\mathbf{c}$ and $\mathbf{T}^\top \mathbf{c}$ via (54)–(55) with $\xi \leftarrow \mathbf{1}$
11: $\quad$ Solve $\mathbf{Z}\mathbf{z} = \mathbf{U}^\top \mathbf{G}\mathbf{c}$ using LU factors; $\quad T_n \leftarrow \frac{n}{\varepsilon} \mathbf{1}^\top \mathbf{M} \mathbf{1} - \frac{n(1-\varepsilon)}{\varepsilon} (\mathbf{T}^\top \mathbf{c})^\top \mathbf{z}$
12: **end if**
13: **for** $b = 1$ to $B$ **do**
14: $\quad$ Draw multipliers $\xi$ via split-independent multinomial resampling.
15: $\quad$ **if** test type $=$ MMD **then**
16: $\quad\quad T_n^{(b)} \leftarrow n \cdot \xi^\top \mathbf{M} \xi$
17: $\quad$ **else**
18: $\quad\quad$ Compute $\mathbf{U}^\top \mathbf{G}\mathbf{c}^{(b)}, \mathbf{T}^\top \mathbf{c}^{(b)}$ via (54)–(55) {$\mathcal{O}(n^2)$ operation}
19: $\quad\quad$ Solve $\mathbf{Z}\mathbf{z}^{(b)} = \mathbf{U}^\top \mathbf{G}\mathbf{c}^{(b)}; \quad T_n^{(b)} \leftarrow \frac{n}{\varepsilon} \xi^\top \mathbf{M} \xi - \frac{n(1-\varepsilon)}{\varepsilon} (\mathbf{T}^\top \mathbf{c}^{(b)})^\top \mathbf{z}^{(b)}$ via (56) {$\mathcal{O}(n^2)$ operations}
20: $\quad$ **end if**
21: **end for**
22: **return** $\mathbb{I}(T_n > \widehat{q}_{n,\alpha})$, where $\widehat{q}_{n,\alpha} \leftarrow (1 - \alpha)$-quantile of $\{T_n^{(b)}\}_{b=1}^B$.

---

## H.2. Nyström Approximation

While we do not prove the guarantees from Section 3.1 under Nyström approximation here, the constructions below are supported by existing results in related contexts. We replace the covariate- and outcome-space kernel gram matrices by Nyström factorizations $\widetilde{\mathbf{K}} \approx \mathbf{Z}_x \mathbf{Z}_x^\top$ and $\widetilde{\mathbf{L}} \approx \mathbf{Z}_y \mathbf{Z}_y^\top$, where $\mathbf{Z}_x \in \mathbb{R}^{n \times r_x}$ and $\mathbf{Z}_y \in \mathbb{R}^{n \times r_y}$. We use these same approximate kernels throughout: in the cross-fitted kernel ridge regressions defining the coefficient matrices $\mathbf{C}$ and $\mathbf{E}$, and in the final test statistics. This is inspired by prior work showing that Nyström approximations can preserve estimation performance for kernel ridge regression under appropriate low rank-sclaing (Alaoui & Mahoney, 2015), and can preserve kernel mean embedding and MMD estimation performance under suitable spectral assumptions (Chatalic et al., 2022). Thus, the resulting procedure is the SKCD test associated with the Nyström-approximated kernels, with only the propensity score estimation (using gradient-boosted trees) unchanged.

For the cross-fitted nuisance fits, the dense KRR matrices are replaced by their feature-space forms. If fold $r$ is used to predict on fold $s$, then

$$\mathbf{H}_{sr}^{\text{Ny}} = \mathbf{Z}_{x,s}(\mathbf{Z}_{x,r}^\top \mathbf{Z}_{x,r} + \lambda_{\text{reg}}\mathbf{I})^{-1}\mathbf{Z}_{x,r}^\top,$$

with the treated and control fits as in the original construction. Using these matrices in the same block formulas as the original construction yields Nyström versions $\widetilde{\mathbf{C}}$ and $\widetilde{\mathbf{E}}$.

**Nyström SKCD-MMD.** This construction is inspired by related results for Nyström approximation of kernel mean embeddings and MMD (Chatalic et al., 2022), and a recent Nyström-based two-sample test that established exact level and power guarantees in that setting (Chatalic et al., 2025). For the approximate coefficient matrix $\widetilde{\mathbf{C}}$, we define its representation in the Nyström feature space as $\widetilde{\mathbf{W}} := \mathbf{Z}_x^\top \widetilde{\mathbf{C}} \mathbf{Z}_y \in \mathbb{R}^{r_x \times r_y}$. The Nyström SKCD-MMD statistic then has the closed form:

$$T_{n,\text{Ny}}^{\text{MMD}} = n\langle \widetilde{\mathbf{C}}, \widetilde{\mathbf{K}} \widetilde{\mathbf{C}} \widetilde{\mathbf{L}} \rangle_F = n\|\widetilde{\mathbf{W}}\|_F^2.$$

Under the bootstrap, $\widetilde{\mathbf{C}}^{(b)} = \text{diag}(\boldsymbol{\xi})\widetilde{\mathbf{C}}$, so $T_{n,\text{Ny}}^{(b),\text{MMD}} = n\|\mathbf{Z}_x^\top \text{diag}(\boldsymbol{\xi})\widetilde{\mathbf{C}}\mathbf{Z}_y\|_F^2$. Thus the fast-bootstrap structure is preserved exactly after replacing the dense kernels by their Nyström feature representations.

**Nyström SKCD-Wald.** This construction is in the spirit of Zhang et al. (2018) who, in the related context of independence testing, similarly replace their kernel statistic by a Nyström feature-space version and calibrate the resulting test through an approximated finite-dimensional covariance matrix. Let $d = r_x r_y$, and define $\widetilde{\mathbf{w}} := \text{vec}(\mathbf{Z}_x^\top \widetilde{\mathbf{C}} \mathbf{Z}_y) \in \mathbb{R}^d$. Using the same foldwise constructions as in the exact Wald-type derivation, let $\widetilde{\mathbf{D}}^1, \widetilde{\mathbf{D}}^2, \widehat{\mathbf{V}}^1, \widehat{\mathbf{V}}^2$ be formed from $\widetilde{\mathbf{C}}, \widetilde{\mathbf{E}}$. For each fold $s \in \{1, 2\}$, define

$$\mathbf{P}_s = \widetilde{\mathbf{D}}^s \mathbf{Z}_y \in \mathbb{R}^{n \times r_y}, \qquad \widetilde{\mathbf{v}}_s = \text{vec}(\mathbf{Z}_x^\top \widehat{\mathbf{V}}^s \mathbf{Z}_y) \in \mathbb{R}^d.$$

If $i \in \mathcal{I}_s$ (belongs to fold $s$), and $\mathbf{z}_{x,i}^\top$ and $\mathbf{p}_{s,i}^\top$ denote the $i$th rows of $\mathbf{Z}_x$ and $\mathbf{P}_s$, then the representation of the EIF at $i$ (normalized by $\sqrt{2n_s}$) in the Nyström feature space is

$$\mathbf{f}_i = \text{vec}(\mathbf{z}_{x,i} \mathbf{p}_{s,i}^\top) - \widetilde{\mathbf{v}}_s \in \mathbb{R}^d.$$

Stacking these columns gives $\widetilde{\mathbf{F}} = [\mathbf{f}_1, \ldots, \mathbf{f}_n] \in \mathbb{R}^{d \times n}$. The Nyström covariance proxy is therefore $\widehat{\Sigma}_{\text{Ny}} = \widetilde{\mathbf{F}} \widetilde{\mathbf{F}}^\top$. The corresponding Nyström SKCD-Wald statistic is then

$$T_{n,\text{Ny}}^{\text{Wald}} = n\, \widetilde{\mathbf{w}}^\top \left( \varepsilon \mathbf{I}_d + (1-\varepsilon) \widehat{\Sigma}_{\text{Ny}} \right)^{-1} \widetilde{\mathbf{w}}.$$

Therefore in the fast SKCD algorithm, we pre-compute the factorization of the $d \times d$ SPD matrix $\varepsilon \mathbf{I}_d + (1-\varepsilon) \widetilde{\mathbf{F}} \widetilde{\mathbf{F}}^\top$. Note that under the bootstrap,

$$\widetilde{\mathbf{w}}^{(b)} = \text{vec}\left( \mathbf{Z}_x^\top \text{diag}(\boldsymbol{\xi}) \widetilde{\mathbf{C}} \mathbf{Z}_y \right),$$

and the bootstrapped Wald-type statistic is obtained by replacing $\widetilde{\mathbf{w}}$ with $\widetilde{\mathbf{w}}^{(b)}$ in the same precomputed linear system.

**Computational complexity.** We use Algorithm 2, but with the above constructions instead of the exact closed-forms. For MMD, after forming the Nyström-KRR coefficients $\widetilde{\mathbf{C}}, \widetilde{\mathbf{E}}$, the dominant costs are $\widetilde{\mathbf{C}} \mathbf{Z}_y \in \mathbb{R}^{n \times r_y}$ and $\mathbf{Z}_x^\top (\widetilde{\mathbf{C}} \mathbf{Z}_y) \in \mathbb{R}^{r_x \times r_y}$, so the total MMD cost is

$$O\left( n^2(r_x + r_y) + Bn r_x r_y \right).$$

For the Wald-type statistic implementation, the additional setup consists of building $\widetilde{\mathbf{F}} \in \mathbb{R}^{d \times n}$, forming $\widetilde{\mathbf{F}} \widetilde{\mathbf{F}}^\top \in \mathbb{R}^{d \times d}$, and factorizing the $d \times d$ matrix $\varepsilon \mathbf{I}_d + (1-\varepsilon) \widetilde{\mathbf{F}} \widetilde{\mathbf{F}}^\top$. Hence $T_{n,\text{Ny}}^{\text{Wald}}$ has total cost

$$O\left( n^2(r_x + r_y) + nd^2 + d^3 + B(nd + d^2) \right),$$

where $d = r_x r_y$. We set $r_x = r_y = r$, so that $d = r^2$. In this case, the MMD test complexity is

$$O\left( n^2 r + Bn r^2 \right),$$

and the Wald-type test complexity becomes

$$O\left( n^2 r + n r^4 + r^6 + B(n r^2 + r^4) \right).$$

In the regimes we study, $d = r^2 < n$ and the lower-order terms satisfy $nr^4 < n^2 r$, $r^6 < nr^4$, and $r^4 < nr^2$. Therefore, both Nyström MMD and Wald-type test complexities have effectively the same order: $O\left( n^2 r + Bn r^2 \right)$. We empirically study the statistical performance and runtime gains from using this implementation under $d = r^2 < n$ in App. J.3.

# I. Experimental Details

This appendix provides complete specifications for our experiments section, including the data generation process, model architectures, inference procedures, and implementation details. The code is provided in https://github.com/saksham-jain01/CDTEs.

## I.1. Distribution Shift in Images

SETUP

Our simulation design uses the MNIST handwritten digit dataset (Deng, 2012) to create scenarios where treatment effects manifest as multivariate distribution shifts that are challenging to detect. As mentioned in Sec. 4.1, we let both covariates $X$ and outcomes $Y$ be learned representations of images.

The feature extraction pipeline learns embeddings on a subset of $25k$ images from the MNIST training set ($60k$ images),[1] and is then *fixed*. It is applied to the MNIST test set ($10k$ images)[2] pooled with the remaining $35k$ images from the training set. We henceforth denote the set of raw MNIST images used in our experiments by $\{\text{Image}_i\}_{i=1}^{45k}$.

**Feature extraction.** We train a ResNet-18-based Encoder that maps input images ($1 \times 28 \times 28$) to a 5-dimensional feature space. The network consists of the standard four residual blocks (channels: 64, 128, 256, 512). The 512-dimensional output of the final residual block is flattened and projected via a fully connected layer to dimension $d = 5$, followed by Batch Normalization. A final linear layer maps the 5-dimensional embeddings to the 10 class logits. It is trained to minimize the cross-entropy classification loss, and the optimization uses Adam ($\alpha = 10^{-3}$, weight decay $10^{-5}$) with a `ReduceLROnPlateau` scheduler for 20 epochs (batch size 512). We extract embeddings for the training set and fit a PCA model ($n_{\text{components}} = 5$) to learn the rotation matrix that diagonalizes the feature covariance.

To validate the feature extraction pipeline, we confirm that a linear classifier trained on the fixed training embeddings achieves $> 98\%$ accuracy when evaluated on the embeddings of the held-out test set. We also verify that this pipeline is sensitive to rotations, evidenced by a drop in classification accuracy to $\approx 92\%$ when applied to rotated images.

**Data generating process.** We define the covariates $X_i$ for the simulation by passing the MNIST test set ($n = 10{,}000$) images through the frozen Encoder-PCA pipeline, i.e., symbolically,

$$X_i = \text{PCA}(\text{Encoder}(\text{Image}_i)) \in \mathbb{R}^5, \quad i \in [10{,}000].$$

Binary treatments $A_i \in \{0, 1\}$ are generated via a non-linear logistic model whose parameters are functions of the pre-treatment covariate embeddings. Given $X_i = (X_{i1}, \ldots, X_{i5})$, we define the log-odds as $\ell(X_i) := 2 - 1.5 X_{i1} \tanh(2X_{i1})$. We also define a raw logistic probability $p(X_i) := \{1 + \exp[-\ell(X_i)]\}^{-1}$, which, to maintain strict overlap (positivity), is rescaled to define the propensity score $\pi(X_i) := 0.2 + 0.6 \frac{p(X_i) - \min_j p(X_j)}{\max_j p(X_j) - \min_j p(X_j)}$. The treatment is drawn as $A_i \sim$ Bernoulli$(\pi(X_i))$. Note that $\pi(X_i) \in [0.2, 0.8]$ for all $i$, which for our MNIST setup, ensures nearly equal-sized treatment and control groups.

Outcomes $Y_i \in \mathbb{R}^5$ are generated by manipulating the raw image $\text{Image}_i$ and passing it through the fixed feature extraction pipeline described above. Let $\text{IntensityChange}(\cdot; u)$ denote an operator that multiplies an image's pixel values by a factor $u$ and clips the result to $[0, 255]$. Let $\text{Rotate}(\cdot; \theta)$ denote an operator that rotates an image by $\theta$ degrees using `torchvision.transforms.functional.rotate`. Since the frozen feature extraction pipeline is not rotation-invariant, rotations induce distributional changes in the embeddings, and thus in $Y$.

We draw i.i.d factors $u_i \sim \text{Unif}(0.2, 1.8)$ for each image (regardless of treatment group).

Under the *null*, outcomes ignore treatment, and we have

$$Y_i = \text{PCA}(\text{Encoder}(\text{IntensityChange}(\text{Image}_i; u_i))) \in \mathbb{R}^5.$$

Under the *alternative*, each image in the treated group receives (on top of the intensity change) a rotation whose angle is determined as $\theta(X_i) := 20 + 5 \tanh(X_{i1}) + \epsilon_i$, where $\epsilon_i \sim \mathcal{N}(0, 1.5)$,

$$Y_i = \text{PCA}(\text{Encoder}(\text{Rotate}(\text{IntensityChange}(\text{Image}_i; u_i); A_i \theta(X_i)))) \in \mathbb{R}^5.$$

Thus, under the alternative, when $A_i = 0$ there is no rotation (yielding exactly the same outputs for $\text{Image}_i$ as under the null), and when $A_i = 1$ it generates a multivariate distributional effect of the treatment that varies with $X$ and is not limited to a mean shift.

---

[1] https://www.kaggle.com/datasets/hojjatk/mnist-dataset (train-images-idx3-ubyte.gz)
[2] https://www.kaggle.com/datasets/hojjatk/mnist-dataset (t10k-images-idx3-ubyte.gz)

HYPOTHESIS TESTING

**Common methodology.** All three methods under comparison, the baseline KCD test (Park et al., 2021) and our proposed SKCD-MMD and SKCD-Wald tests, share a common computational backbone: kernel ridge regression (KRR) for estimating conditional mean embeddings (CME) and gradient-boosted trees for propensity score estimation. A key structural difference is that our proposed methods employ cross-fitting, whereas the baseline does not. All matrix inversions are computed via `linalg.solve` to avoid explicitly forming inverse matrices.

*Kernel specification and bandwidth selection:* All methods use the Gaussian RBF kernel for both the covariate space $\mathcal{X}$ and outcome space $\mathcal{Y}$:

$$k(x, x') = \exp\left(-\|x - x'\|^2/[2\sigma_k^2]\right), \qquad \ell(y, y') = \exp\left(-\|y - y'\|^2/[2\sigma_\ell^2]\right).$$

For the proposed SKCD tests, both kernels use a single bandwidth computed via the median heuristic (Fukumizu et al., 2009) on all observations: $\sigma_k$ is set to the median of $\{\|x_i - x_j\| : i < j\}$ and $\sigma_\ell$ to the median of $\{\|y_i - y_j\| : i < j\}$. For the baseline KCD test, following Park et al. (2021), the outcome kernel $\ell$ uses a common bandwidth computed on all $\{y_i\}_{i=1}^n$, while the covariate kernels are separate: $k_1$ and $k_0$ that use treatment group-specific bandwidths computed separately on $\{x_i : a_i = 1\}$ and $\{x_i : a_i = 0\}$.

*Propensity score estimation:* We estimate the propensity score $\pi_\star(x) = P_\star(A = 1 \mid X = x)$ using gradient-boosted trees via the LightGBM library (Ke et al., 2017). Hyperparameters are tuned using the Optuna framework (Akiba et al., 2019) to minimize binary log-loss on an internal 80/20 train-validation split with early stopping (patience of 10 rounds). The resulting propensity estimates are clipped to $[10^{-6}, 1 - 10^{-6}]$ for numerical stability. For the proposed SKCD tests, this estimation is performed within a 2-fold cross-fitting procedure (training on one fold, evaluating on the other). For the baseline KCD, it is performed on the full dataset. To simulate misspecification, we restrict the model input to only the last feature of the PCA-decorrelated embeddings ($X_{:,5}$).

*Outcome nuisance estimation:* The conditional mean embedding $\nu_{\star,a}(x) = \mathbb{E}_\star[L_Y \mid A = a, X = x]$ (3), is estimated for each treatment group $a \in \{0, 1\}$ using kernel ridge regression (KRR) in closed-form. Given a training set of $\tilde{n}$ observations $\{(x_j, y_j)\}_{j=1}^{\tilde{n}}$ with $A_j = a$, the estimator takes the form

$$\nu_{\tilde{n},a}(x) = \sum_{j=1}^{\tilde{n}} [\boldsymbol{\beta}_a(x)]_j L_{y_j}, \qquad \text{where} \quad \boldsymbol{\beta}_a(x) = (\mathbf{K}_a + \lambda \mathbf{I}_{\tilde{n}})^{-1} \mathbf{k}_a(x), \tag{57}$$

where $\mathbf{K}_a \in \mathbb{R}^{\tilde{n} \times \tilde{n}}$ is the Gram matrix with $[\mathbf{K}_a]_{ij} = k(x_i, x_j)$ for observations in treatment group $a$, $\mathbf{k}_a(x) = [k(x_1, x), \ldots, k(x_{\tilde{n}}, x)]^\top$ is the cross-kernel vector evaluating the covariate kernel between the training points and the query point $x$, and $\lambda > 0$ is a regularization parameter. We fix $\lambda = 10^{-3}$ throughout the experiments. For the SKCD, the above coefficients $[\boldsymbol{\beta}_a(x)]_j$ directly populate the off-diagonal blocks of the weight matrices $\mathbf{C}$ (15) and $\mathbf{E}$ (36) used in our closed-form statistics. To simulate outcome misspecification, we recompute the covariate kernel matrices (including bandwidths) using only the last feature ($X_{:,5}$).

**Baseline KCD test implementation.** We implement the KCD test from Algorithm 1 of Park et al. (2021), with the modification that propensity scores are estimated via gradient-boosted trees rather than kernel logistic regression. Note that KCD does not employ cross-fitting: the outcome models are trained on all observations with $A = a$ and evaluated on the full dataset. Let $\mathbf{K}_a \in \mathbb{R}^{n_a \times n_a}$ denote the Gram matrix restricted to the $n_a$ observations with $A = a$, and let $\mathbf{K}_{\text{all},a} \in \mathbb{R}^{n \times n_a}$ denote the cross-kernel matrix between all $n$ observations and those with $A = a$. The matrix $\mathbf{M}_a \in \mathbb{R}^{n \times n_a}$ is defined as

$$\mathbf{M}_a = \mathbf{K}_{\text{all},a}(\mathbf{K}_a + \lambda \mathbf{I}_{n_a})^{-1},$$

where the $i$-th row satisfies $[\mathbf{M}_a]_{i,:} = \boldsymbol{\beta}_a(x_i)^\top$, with $\boldsymbol{\beta}_a(x_i)$ the KRR coefficient vector from (57) for query point $x_i, i \in [n]$. Let $\mathbf{L}_{a\tilde{a}}$ denote the submatrix of the outcome gram matrix $\mathbf{L}$ corresponding to rows with $A = a$ and columns with $A = \tilde{a}$. The KCD statistic is then computed as

$$\widehat{\text{KCD}} = \frac{1}{n} \text{tr}\left(\mathbf{M}_1 \mathbf{L}_{11} \mathbf{M}_1^\top - 2\mathbf{M}_1 \mathbf{L}_{10} \mathbf{M}_0^\top + \mathbf{M}_0 \mathbf{L}_{00} \mathbf{M}_0^\top\right),$$

where $\mathbf{L}$ is the outcome kernel Gram matrix over all observations. Note that our implementation of $\widehat{\text{KCD}}$ is numerically equivalent to that in Lemma 4.4 of Park et al. (2021). To approximate the null distribution, Park et al. (2021) employ a permutation procedure with $M$ permutations, each of which involves re-solving the KRR systems.

**Proposed SKCD test implementation.** Our proposed SKCD-MMD and SKCD-Wald tests employ 2-fold cross-fitting and the test statistics are computed using the closed-form expressions from Propositions 3.6 and 3.7. For the Wald-type statistic, following the discussion in App. G.2.3, the regularization parameter $\varepsilon$ for the covariance operator inversion is chosen by setting $\gamma = 1/3$ in Eq. 43, which heuristically gives 75% weight to the covariance operator and the rest to the regularizer, the identity. Inference is performed via the fast SKCD algorithm (Alg. 2) detailed in App. H.1 with $B = 1000$ bootstrap samples.

**Complexity and runtime comparison.** Since the training of gradient-boosted decision trees is sub-quadratic in $n$ (Ke et al., 2017), the overall cost of fitting the nuisances is dominated by the matrix inversions required for KRR. This results in a worst-case computational complexity of $\mathcal{O}(Mn^3)$ for the KCD baseline. In contrast, in our proposed fast SKCD test implementation, the cubic cost of nuisance fitting, LU factorization, and pre-computation of the weight matrices is incurred only once. Since subsequent bootstrap resampling requires only matrix-vector operations, the resulting worst-case complexity is $\mathcal{O}(n^3 + Bn^2)$. Empirically, this yields substantial speedups. At sample size $n = 2000$, the average wall-clock runtime per MC replicate is approximately 0.5 seconds for SKCD_MMD and 1.7 seconds for SKCD_Wald (using $B = 1000$), compared to 2.4 seconds for the KCD baseline (using $M = 150$).

**Compute details.** All methods are implemented in Python using PyTorch for GPU-accelerated kernel and matrix operations, LightGBM for propensity score estimation, and NumPy/SciPy for general numerical operations. In our implementation, we sort the data by treatment assignment to exploit efficient block-matrix operations on the GPU, though this does not affect the statistical definitions. All experiments were conducted on compute nodes equipped with an NVIDIA T4 GPU and 32GB RAM.

### I.2. Real Data: 401k Eligibility

DATA

We utilize data from Wave 4 of the 1990 Survey of Income and Program Participation (SIPP), consisting of $n = 9,915$ households (Chernozhukov & Hansen, 2004). As established in the literature (Poterba & Venti, 1994), while participation in 401(k) plans is endogenous, *eligibility* ($A$) can be considered plausibly unconfounded conditional on income and other household characteristics.

**Variables.** The treatment $A$ is 401(k) eligibility. The multivariate outcome $Y \in \mathbb{R}^3$ comprises Net Financial Assets (TFA), Net Non-401(k) Financial Assets (NIFA), and Total Wealth (TW). The covariates $X$ consist of four continuous variables (age, income, family size, education) and five binary indicators (defined-benefit plan, marital status, two-earner household, IRA participation, home ownership).

**Preprocessing.** Continuous covariates and all outcome variables are standardized to zero mean and unit variance prior to analysis. Binary covariates are left unscaled.

MMD-BASED CONFIDENCE BANDS

Theorem 3.4 provides uniform confidence bands over the full product space $\mathcal{X} \times \mathcal{Y}$. For visualization and interpretation at a specific covariate profile $x \in \mathcal{X}$, we adapt this construction to the RKHS slice $\mathcal{H}_x := \{h(x, \cdot) : h \in \mathcal{H}\}$. This yields a confidence band that is uniform over all $y \in \mathcal{Y}$ for the fixed profile $x$.

**Confidence band construction.** Recall from Proposition 3.6 that the squared MMD statistic takes the form $T_n^{\mathrm{MMD}} = n \langle \mathbf{C}, \mathbf{KCL} \rangle_F$, and from (50) that the bootstrap statistic is $T_n^{(b),\mathrm{MMD}} = n\xi^\top \mathbf{M}\xi$ where $\mathbf{M} = \mathbf{K} \circ (\mathbf{CLC}^\top)$.

For a fixed evaluation point $x \in \mathcal{X}$, define the kernel vector $\mathbf{k}_x := [k(x_1, x), \ldots, k(x_n, x)]^\top \in \mathbb{R}^n$. Restricting to the slice $\mathcal{H}_x$ can be done by replacing the full covariate kernel $\mathbf{K}$ with the rank-one matrix $\mathbf{k}_x \mathbf{k}_x^\top$. The slice Gram matrix is thus

$$\mathbf{M}_x := (\mathbf{k}_x \mathbf{k}_x^\top) \circ (\mathbf{CLC}^\top). \tag{58}$$

The bootstrap statistic for the slice becomes

$$T_{n,x}^{(b)} = n\xi^\top \mathbf{M}_x \xi, \tag{59}$$

which has the same quadratic form as in Eq. 50 but with the slice-restricted Gram matrix $\mathbf{M}_x$. Let $\hat{q}_{n,x}(\alpha)$ denote the $(1-\alpha)$-quantile of the bootstrap distribution $\{T_{n,x}^{(b)}\}_{b=1}^B$. The uniform-in-$y$ confidence band for $\psi_\star(x, \cdot)$ is

$$\mathcal{C}_{n,x}(y) = \left[ \bar{\psi}_n(x,y) - \sqrt{\hat{q}_{n,x}(\alpha)/n}, \ \bar{\psi}_n(x,y) + \sqrt{\hat{q}_{n,x}(\alpha)/n} \right], \tag{60}$$

where $\sqrt{\hat{q}_{n,x}(\alpha)/n}$ is constant across all $y$, ensuring uniform coverage over the outcome space.

**Witness function evaluation.** The estimated witness function at $(x, y)$ is computed as

$$\bar{\psi}_n(x,y) = \mathbf{k}_x^\top \mathbf{C} \boldsymbol{\ell}_y, \tag{61}$$

where $\boldsymbol{\ell}_y := [\ell(y_1, y), \dots, \ell(y_n, y)]^\top \in \mathbb{R}^n$ is the outcome kernel vector. This can be vectorized for a grid of $y$ values.

**Cross-sectional visualization.** Since the full witness function $\psi_\star(x, \cdot) : \mathbb{R}^3 \to \mathbb{R}$ is a surface over the 3-D outcome space, direct visualization is infeasible. We instead compute one-dimensional cross-sections by varying each wealth component $Y_j$ over its support while fixing the remaining components at zero (which corresponds to the sample mean in standardized coordinates). Note that the confidence band (60) applies uniformly to all three cross-sections since the band width $\sqrt{\hat{q}_{n,x}(\alpha)}$ is computed using the full outcome kernel and thus provides valid coverage over the entire outcome space $\mathcal{Y}$.

**Implementation.** We follow the same implementation as the simulation study in App. I.1. Specifically, we use the fast SKCD test (Alg. 2) with the MMD statistic, with Gaussian RBF kernels for both $\mathcal{X}$ and $\mathcal{Y}$, bandwidths selected via the median heuristic, propensity scores estimated via LightGBM with Optuna-based hyperparameter tuning, and kernel ridge regression for the conditional mean embeddings with regularization $\lambda = 10^{-3}$. We use $B = 1000$ bootstrap replicates at level $\alpha = 0.05$. We reserve approx. 1% of the data ($n_{\text{eval}} = 99$ households) as an evaluation set from which individual profiles are drawn. The remaining 99% ($n = 9816$) is split into two equal folds for cross-fitting.

The coefficient matrix $\mathbf{C}$ is constructed via Eqs. (44)–(46), and the outcome Gram matrix $\mathbf{L}$ is computed using the median-heuristic bandwidth. For each evaluation profile $x$, we compute $\mathbf{M}_x$ via Eq. 58 and run $B = 1000$ bootstrap iterations using split-independent multinomial resampling to obtain $\hat{q}_{n,x}(0.05)$. The witness function cross-sections are evaluated on a grid of 100 points spanning $[-3, 3]$ in standardized units, then transformed back to original units ($1k) for visualization.

**Complexity and runtime.** Since the slice Gram matrix $\mathbf{M}_x$ depends on the evaluation point $x$, it must be recomputed for each individual profile. However, the outcome covariance $\mathbf{CLC}^\top$ is shared across all profiles, and the bootstrap loop requires only $\mathcal{O}(n^2)$ operations per replicate (the quadratic form in Eq. 59). For the two profiles analyzed in Fig. 3, the total computation time is approximately 1.5 minutes on a single NVIDIA T4 GPU.

**Compute details.** The implementation uses PyTorch for GPU-accelerated kernel and matrix operations, LightGBM for propensity estimation, and NumPy/SciPy for general numerical operations. All results were computed on a node equipped with an NVIDIA T4 GPU and 32GB RAM.

## J. Additional Experiments

### J.1. Sensitivity to Wald-type Regularization

Figure 6 reports the finite-sample rejection rates (type 1 error under the null and power under the alternative) of our Wald-type test for the MNIST-based simulation study described in Sec. 4.1, over the grid $\gamma \in \left\{ \frac{1}{19}, \frac{1}{3}, 1, 3 \right\}$, which we chose to reflect, heuristically, a weight of 95%, 75%, 50%, 25%, 5% respectively on $\mathbf{U}^\top \mathbf{T}$. Note that $\gamma = 1/3$ is the default used in Sec. 4.1. In these 1000 MC runs, the $\varepsilon$ under the default $\gamma = 1/3$ averages roughly 0.36–0.43, whereas the more aggressive choice $\gamma = 1/19$ yields $\varepsilon$ around 0.08–0.13. Overall, the test is fairly stable over $\gamma \in \{1/3, 1, 3, 19\}$, while the more extreme choices behave more unstably, especially in the estimated-propensity setting. The known-propensity plots provide a clearer picture, especially under the 'outcome misspecified' setting. $\gamma = 1/19$ typically yields higher power, and only shows inflated type 1 error at smaller sample sizes.

To investigate this in more detail, we empirically study the spectral behavior of the regularized covariance proxy under different choices of $\gamma$. Let $\mathbf{M}_n := \mathbf{U}^\top \mathbf{T}$ and $\tilde{\gamma}_n := \varepsilon/(1 - \varepsilon)$, so that $\tilde{\gamma}_n = \gamma \text{tr}(\mathbf{M}_n)$. We are interested in inverting

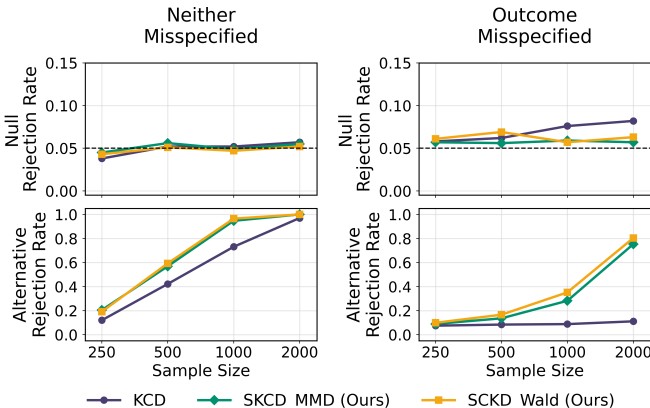

*Figure 4.* Type 1 error and power at $\alpha = 0.05$ across sample sizes when propensity scores are known. **(Left)** The outcome model is correctly specified. **(Right)** The outcome model is misspecified. Since the propensity scores are known, the product of nuisance estimation errors is $o_p(n^{-1/2})$ in both scenarios. Thus, in contrast to the baseline, type 1 error is controlled at the nominal level and power increases with sample size even under outcome misspecification.

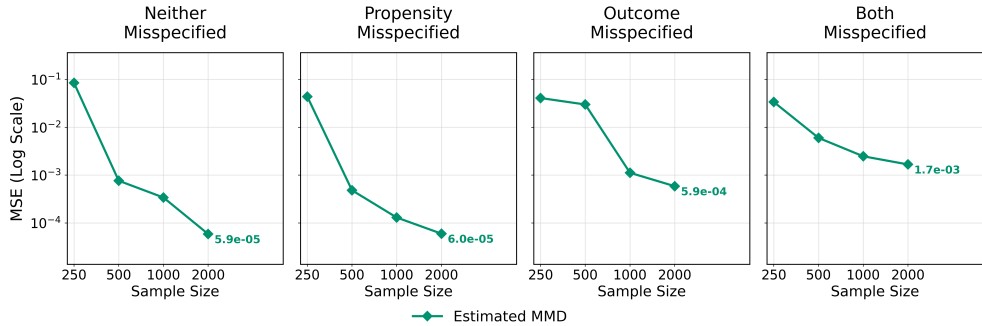

*Figure 5.* Empirical MSE of the SCoDiTE estimator $\bar{\psi}_n$ under the global null, across sample sizes and model misspecificatiom regimes. **(Left three panels)** The MSE decays sharply to zero when at least one of the nuisance models is correctly specified. **(Rightmost panel)** The MSE decays at a much slower rate when both propensity and outcome models are simultaneously misspecified.

$(1 - \varepsilon)\mathbf{M}_n + \varepsilon\mathbf{I}$. Observe that this is proportional to inverting $\mathbf{M}_n + \tilde{\gamma}_n\mathbf{I} = \mathbf{M}_n + \gamma\mathrm{tr}(\mathbf{M}_n)\mathbf{I}$. We take this to be our spectral proxy of interest. It represents a Tikonov-regularized matrix inversion in the same vein as Eric et al. (2007), with $\gamma\mathrm{tr}(\mathbf{M}_n)$ as a data-adaptive (tied to the scale of the covariance proxy) ridge-like regularization level.

This reparameterization gives a useful finite-sample interpretation. Let $\kappa(\mathbf{A})$ denote the condition number of a matrix $\mathbf{A}$. If $\mathbf{M}_n$ were positive semidefinite with eigenvalues $\mu_1 \geq \cdots \geq \mu_r \geq 0$, then

$$\kappa\left(\mathbf{M}_n + \gamma\mathrm{tr}(\mathbf{M}_n)\mathbf{I}\right) = \frac{\mu_1 + \gamma\mathrm{tr}(\mathbf{M}_n)}{\mu_r + \gamma\mathrm{tr}(\mathbf{M}_n)} \leq 1 + \frac{1}{\gamma},$$

and the associated effective dimension $d_{\mathrm{eff}}(\gamma) := \mathrm{tr}\left(\mathbf{M}_n(\mathbf{M}_n + \gamma\mathrm{tr}(\mathbf{M}_n)\mathbf{I})^{-1}\right)$ satisfies $d_{\mathrm{eff}}(\gamma) \leq \frac{1}{\gamma}$. This makes precise how $\gamma$ controls both the stability of the regularized covariance proxy and the number of covariance directions remaining after regularization. We operationalize this interpretation using a *symmetrization* of the covariance proxy $\mathbf{M}_n$. Concretely, for each sampled dataset we form $\widetilde{\mathbf{M}}_n := \frac{1}{2}(\mathbf{M}_n + \mathbf{M}_n^\top)$, whose trace agrees with that of $\mathbf{M}_n$, and compute its eigenspectrum. We report in Table 1 the condition numbers and effective dimensions for the regularized $\widetilde{\mathbf{M}}_n$ at $n = 250$ and $n = 1000$, for different $\gamma$ values under outcome misspecification.

These results support the trends observed in Figure 6. For the default $\gamma = 1/3$, the symmetrized proxy remains well-conditioned but with low effective-dimension across sample sizes; its effective dimension is close to 2 under the null and only modestly larger under the alternative. By contrast, the smaller choice $\gamma = 1/19$ substantially increases the effective dimension but also increases the conditioning number. In other words, the gain in power for the much smaller $\gamma$ comes from relying on several additional covariance directions, but this behavior is accompanied by more finite-sample instability and type 1 error inflation, especially in small sample sizes.

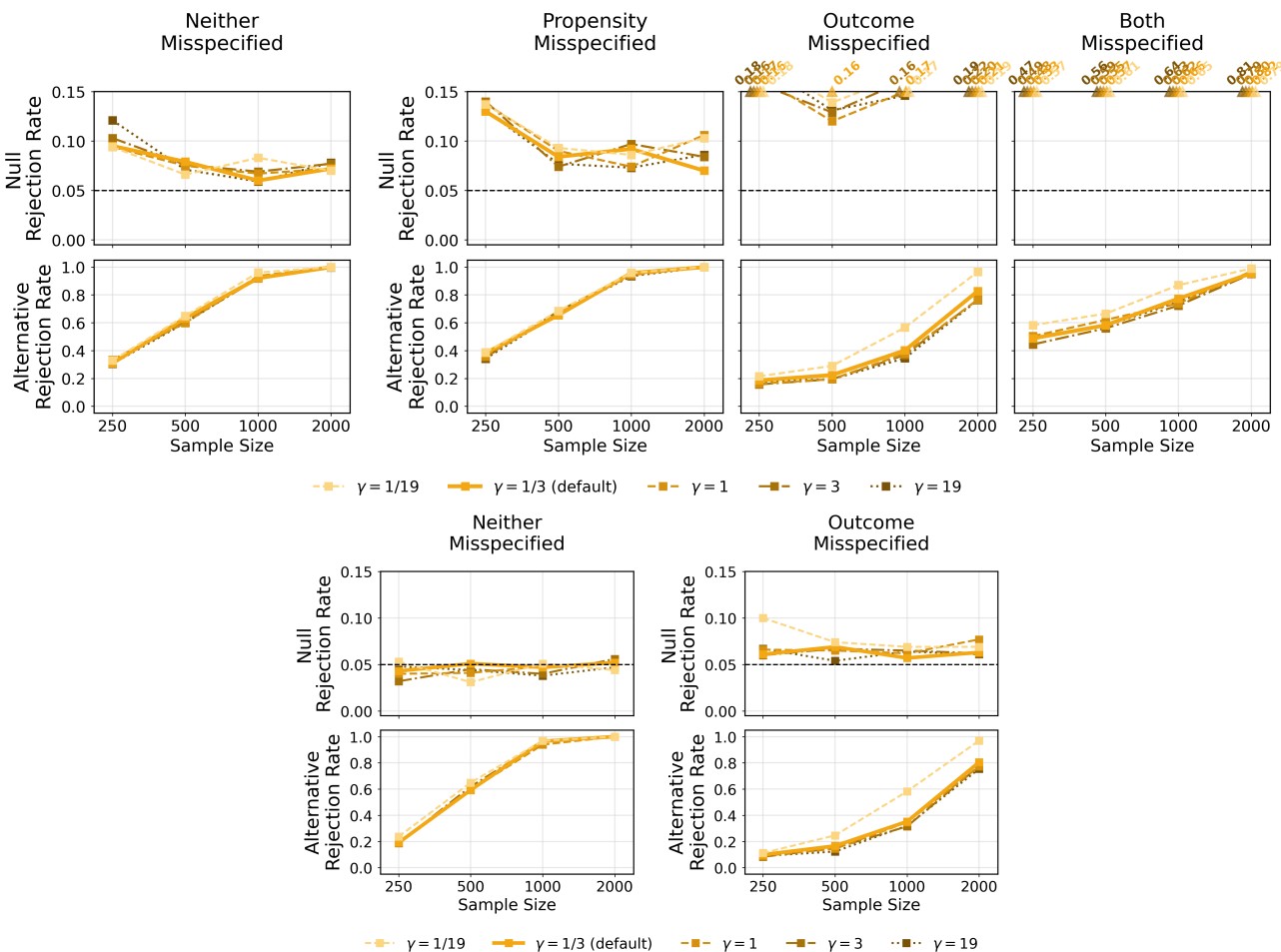

*Figure 6.* The proposed `SKCD_Wald` test rejection rates across $\gamma$ values under estimated propensities (**top**) and known propensities (**bottom**). While the default $\gamma = 1/3$ maintains nearly nominal type 1 error and retains power across all misspecification regimes, $\gamma = 1/19$ outperforms it, except at smaller sample sizes.

Taken together, these results suggest that $\gamma = 1/3$ is a reasonable default. It stabilizes the covariance estimate proxy and retains power. However, decreasing regularization indeed increases power when the covariance is estimated well.

### J.2. Sensitivity to $k$-Bandwidth and Comparison to SKCDAgg

We perform a bandwidth-sensitivity analysis for `SKCD_MMD` and `SKCD_Wald`. The covariate kernel bandwidth enters our methods in two places: it defines the smoothing granularity of the estimand SCoDiTE, and is also used in the kernel ridge regression-based outcome model. We restrict our experiment in this section to the 'neither misspecified' regime so that the outcome model remains well-estimated across the bandwidth range, targeting the question of sensitivity to the bandwidth driven primarily by the estimand definition. We also hold the outcome kernel fixed at its median heuristic for this reason.

For the implementation, we consider the log-spaced grid $h \in \{0.25, 0.5, 1, 2, 4\} \times \hat{\sigma}_x$, where $\hat{\sigma}_x$ is the median-heuristic bandwidth, which is the default for our original MNIST-based study. We evaluate the same sample sizes as in the simulations in Sec. 4.1 and report both type 1 error and power, for both the estimated-propensity and known-propensity settings, in the 'neither misspecified' regime. We also implement the SKCDAgg test for comparison. Note that while the theory developed in Section 3.1 and Appendix F.2 is stated for a *fixed* bandwidth grid, the results should extend to a grid of multiplicative perturbations of the empirical median heuristic by relying on a Slutsky-type argument and the fact that the Gaussian kernel RKHSs are nested as bandwidths shrink; we omit such a proof from this paper, however. The goal of this experiment is to assess finite-sample stability and provide practical guidance on the choice of bandwidth.

*Table 1.* Spectral diagnostics of the regularized covariance proxy $\widetilde{\mathbf{M}}_n$ under outcome misspecification. Median condition number $\kappa$ and effective dimension $d_{\text{eff}}$ over 1000 Monte Carlo replicates.

| | Estimated Propensity | | | | Known Propensity | | | |
| | $n = 250$ | | $n = 1000$ | | $n = 250$ | | $n = 1000$ | |
| $\gamma$ | $\kappa$ | $d_{\text{eff}}$ | $\kappa$ | $d_{\text{eff}}$ | $\kappa$ | $d_{\text{eff}}$ | $\kappa$ | $d_{\text{eff}}$ |
|---|---|---|---|---|---|---|---|---|
| *Null* | | | | | | | | |
| 1/19 | 15.7 | 4.72 | 18.1 | 4.73 | 14.2 | 4.76 | 16.6 | 4.72 |
| 1/3 | 3.3 | 1.84 | 3.7 | 1.83 | 3.1 | 1.80 | 3.5 | 1.77 |
| 1 | 1.8 | 0.89 | 1.9 | 0.91 | 1.7 | 0.85 | 1.8 | 0.86 |
| 3 | 1.3 | 0.37 | 1.3 | 0.39 | 1.2 | 0.34 | 1.3 | 0.36 |
| 19 | 1.0 | 0.07 | 1.0 | 0.07 | 1.0 | 0.06 | 1.0 | 0.06 |
| *Alternative* | | | | | | | | |
| 1/19 | 16.6 | 5.03 | 25.3 | 5.16 | 14.9 | 5.09 | 24.0 | 5.11 |
| 1/3 | 3.5 | 1.90 | 4.8 | 1.97 | 3.2 | 1.87 | 4.6 | 1.91 |
| 1 | 1.8 | 0.92 | 2.3 | 1.01 | 1.7 | 0.88 | 2.2 | 0.98 |
| 3 | 1.3 | 0.39 | 1.4 | 0.46 | 1.2 | 0.36 | 1.4 | 0.44 |
| 19 | 1.0 | 0.07 | 1.1 | 0.09 | 1.0 | 0.06 | 1.1 | 0.08 |

Figures 7 and 8 illustrate the `SKCD_MMD` and `SKCD_Wald` results over the bandwidth grid described above, along with the SKCDAgg test over the same grid. Across both the estimated- and known-propensity settings, the qualitative pattern is similar. In the known-propensity regime, type 1 error remains close to nominal throughout the grid. In the estimated-propensity regime, the smallest sample size exhibits over-rejection across the entire grid, but this appears to be a general small-sample nuisance-estimation effect; by $n \geq 500$, the qualitative dependence on bandwidth is stable.

Under the alternative, power increases with the bandwidth (at the cost of inflated type 1 error, however) over the range we study. In the estimated-propensity regime, for instance, at $n = 1000$ the rejection rate increases from approx. 0.7 at $0.25\hat{\sigma}_x$ to approx. 0.9 between $\hat{\sigma}_x$ and $4\hat{\sigma}_x$. The same pattern appears in the known-propensity regime: at $n = 1000$, power rises from approx. 0.7 at $0.25\hat{\sigma}_x$ to approx. 0.95 between $\hat{\sigma}_x$ and $4\hat{\sigma}_x$. By $n = 2000$, all bandwidths have power near 1. These results are consistent with the interpretation of larger bandwidths as 'borrowing strength' over broader covariate regions: for more smoothly-varying heterogeneity (as in the MNIST study), larger bandwidths can help, while smaller bandwidths trade power for a finer resolution.

**Practical guidance.** While the median heuristic is a reasonable default, it is not a universally optimal choice. In a smooth-heterogeneity regime like the MNIST-based study, bandwidths moderately larger than the median can yield noticeably higher power at moderate sample sizes, while smaller bandwidths are more conservative and less powerful. A practical recommendation is thus to use the median heuristic as a baseline and, when computation allows, use the max-aggregated version over a grid of bandwidth values.

### J.3. Varying Rank Schedules for Nyström Approximation

Recall from Appendix H.2 that if the $r$ inducing points satisfy $d = r^2 < n$, then both the Nyström MMD and Wald-type test complexities have effectively the same order: $O\left(n^2 r + Bnr^2\right)$. Now, if $r \asymp n^{1/3}$, then our Nyström-based algorithm has complexity $O\left(n^2 r + Bnr^2\right) = O\left(n^{7/3} + Bn^{5/3}\right)$. With $r \asymp n^{1/4}$, it is $O\left(n^2 r + Bnr^2\right) = O\left(n^{9/4} + Bn^{3/2}\right)$.

We can also consider a slow-growth schedule, where $r = \lceil r_0\left(\max\{1, \log(en/n_0)\}\right)^q \rceil$ with $(n_0, r_0) = (250, 5)$ (to ensure $r = 5$ at $n = 250$ for our experiments). For any fixed $q > 0$, this gives a complexity of

$$O\left(n^2 r + Bnr^2\right) = O\left(n^2 \log^q n + Bn \log^{2q} n\right).$$

In this case, our Nyström-based SKCD test implementation is near-quadratic in the precomputation phase and near-linear in the bootstrap phase (in $n$), up to polylogarithmic factors. We consider $q \in \{0.25, 0.5, 1.0\}$.

Note that our choices for the scaling of $r$ are inspired by prior work showing that approximate-kernel MMD procedures can retain guarantees under slowly growing approximation feature dimension, either (for Nyström approximation) through explicit schedules tied to spectral decay (Chatalic et al., 2022) or, in the related random fourier feature setting, under arbitrarily slow divergence (Choi & Kim, 2024).

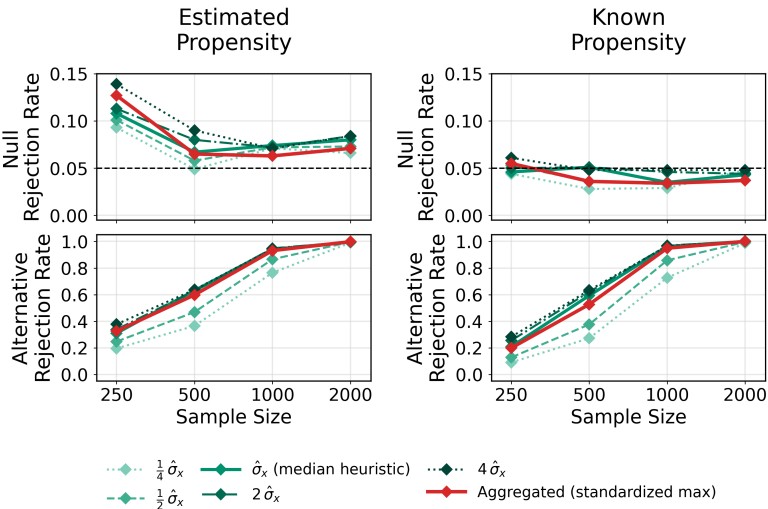

*Figure 7.* SKCD_MMD over the bandwidth grid $h \in \{0.25, 0.5, 1, 2, 4\} \times \hat{\sigma}_x$ in the 'neither misspecified' regime, with rows corresponding to type 1 error and power. The red curve corresponds to the standardized max-aggregated test over the same grid, while the remaining curves correspond to the individual bandwidth choices. The dashed horizontal line marks the nominal 0.05 level. The standardized max-aggregated test has the best performance for both type 1 error and power.

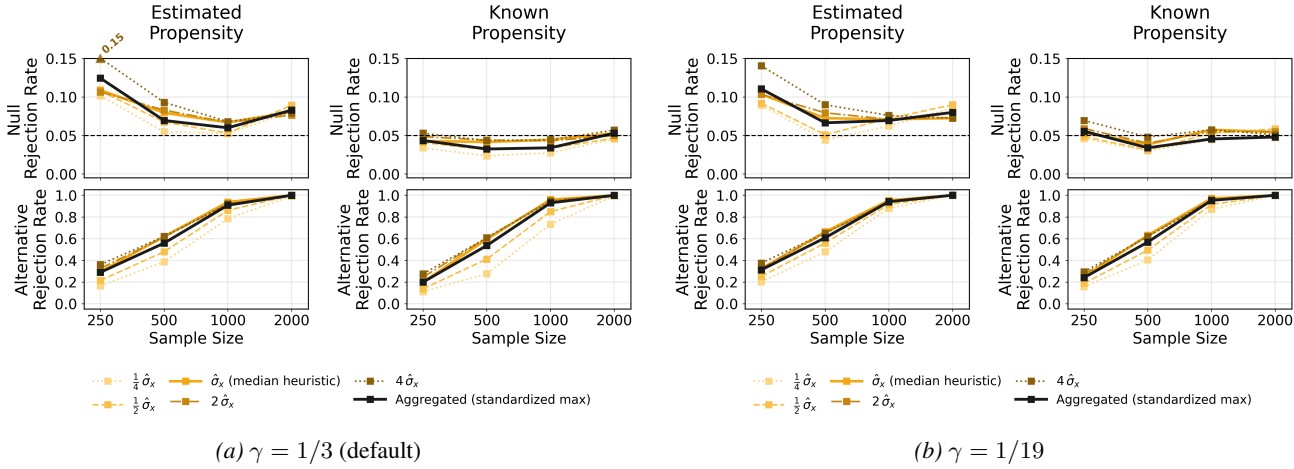

*(a)* $\gamma = 1/3$ (default)                    *(b)* $\gamma = 1/19$

*Figure 8.* SKCD_Wald over the same bandwidth grid. The only difference between (a) and (b) is the Wald regularization parameter $\gamma$.

Fig. 9 plots type 1 error and power with $n$ for different choices of growth schedules for $r$. Fig. 10 plots the average relative runtime speedup per MC run for the schedules. Note that in the known-propensity setting, overall runtime yields a fair comparison with the original implementation. The middle setting for all three schedules usually has the best performance-speedup tradeoff. The best observed average speedup for MMD is approx $1.3\times$ —a modest gain that is not surprising for current sample sizes, given that dense matrix multiplications are already highly optimized on GPUs; the observed speedup is due to the low-rank approximation used in KRR. In contrast, the Wald-type test achieves a substantially larger average speedup of approx $5.4\times$, driven primarily by the computational efficiency gains from low-rank matrix inversion.

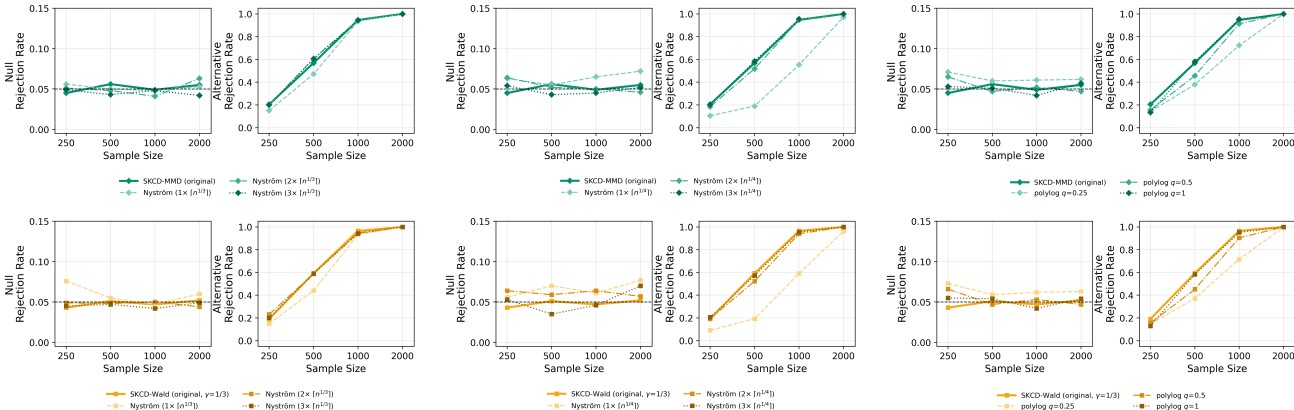

*Figure 9.* Type1 error and power of Nyström-approximated SKCD tests under three rank schedules: $\lceil n^{1/3} \rceil$ (**left**), $\lceil n^{1/4} \rceil$ (**center**), and polylogarithmic (**right**). **Top** row: SKCD-MMD. **Bottom** row: SKCD-Wald ($\gamma = 1/3$). All plots are under the known-propensities 'neither misspecified' regime.

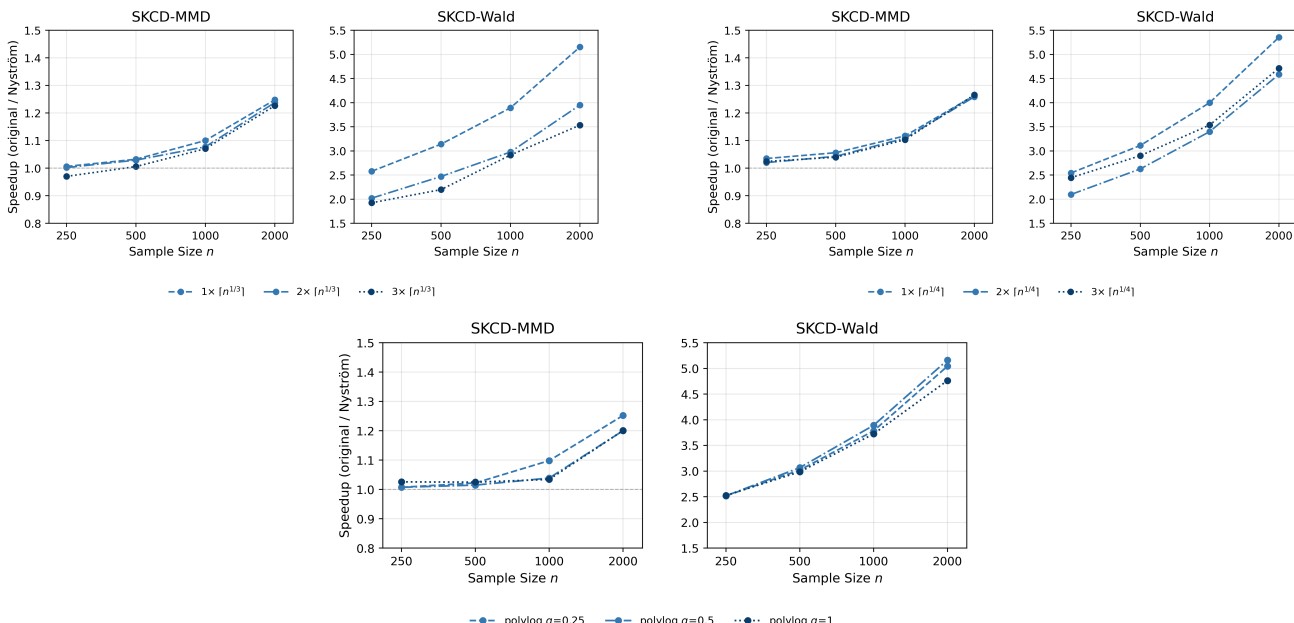

*Figure 10.* Average speedup (original / Nyström) for SKCD-MMD (left panel of each pair) and SKCD-Wald (right panel) under the three rank schedules implemented in Fig. 9: $\lceil n^{1/3} \rceil$ (**top left**), $\lceil n^{1/4} \rceil$ (**top right**), and polylogarithmic (**bottom**).

