# OpenReview forum: "Conditional Distributional Treatment Effects: Doubly Robust Estimation and Testing"
_ICML.cc/2026/Conference — ICML 2026 regular_

### Official Review · Reviewer_meTM · 2026-02-23

**Soundness:** 4
**Presentation:** 2
**Significance:** 2
**Originality:** 4
**Overall Recommendation:** 4
**Confidence:** 3

**Summary:**

The paper proposes the kernel-based conditional independence test on potential outcomes in causal setting, which uses the doubly-robust esimator of the estimand. In addtion to starndard MMD formulation, the paper proposed Wald-type test statistics which allows for constructing $1-\alpha$ quantile of null distribution without refitting the nuisance models. The paper suppots the proposed method using both theoretical analysis and empirical evalution.

**Compliance With Llm Reviewing Policy:**

Affirmed.

**Key Questions For Authors:**

My questions are:
- How the problem setting differs from the original kernel conditional independence test
- Is there any application of this test?
- Some discussion on using the Wald type formulation in other statistical independence test would be helpful

**Limitations:**

yes

**Strengths And Weaknesses:**

**Soundness**: The technical discussion seems reasonable though I have not checked the proof thoroughly.

**Presentation**: The paper could be more properly positioned by describing the difference from original kernel conditional independence test, since it seems we can have dataset of $D_1 = \\{Y^{(1)}, X\\}$ and $D_0 = \\{Y^{(0)}, X\\}$ if we split dataset by the treatment assignments. My take is that existing kernel conditional independence test testing $P_{XY|Z}$ requires the joint samples $(X, Y, Z)$, which is not fulfilled in the causal setting, but it is worth of explict discussion

**Significance**: It would be nice to have an application scenario of the independence test. Since we are considering causal setting, we would expect that distribution of potential outcomes vary, so it is slightly unclear when we should use the test.

**Originality**: I might not be a right person to judge this since I'm not an expert of statistical testing, but Wald-type formulation seems novel enough to me. I wonder whether it is used in/applicable to the original kernel conditinal independence test.

---

> ### Author Rebuttal · Authors · 2026-03-31
>
> Thank you for your review; it really helps us strengthen the discussion in our paper. We address each of your questions below.
>
> > **The paper could be more properly positioned by describing the difference from original kernel conditional independence test...How the problem setting differs from the original kernel conditional independence test**
>
> Your intuition is exactly right, and we will add an explicit discussion of this distinction to the final version of the paper (in Sec. 2.1). Splitting the dataset by treatment gives $D\_1 = \\{(Y\_i, X\_i) : A\_i = 1\\}$ and $D\_0 = \\{(Y\_i, X\_i) : A\_i = 0\\}$, but these are samples from $P\_{Y,X|A=1}$ and $P\_{Y,X|A=0}$---the distributions we *observe*---not from $P\_{Y(1), X}$ and $P\_{Y(0), X}$---the joint covariate-[potential outcome](https://www.causalconversations.com/post/po-introduction/) distributions we really *want to compare*. Whenever treatment assignment depends on covariates $X$ (confounding), which is typical in the observational causal inference setting, $P_{Y,X|A=a} \neq P_{Y(a),X}$. A naive two-sample test on the split data would therefore conflate confounding with genuine distributional treatment effects. This is precisely why our estimator requires inverse propensity score weighting and outcome modeling; the doubly robust AIPW structure (the submitted paper's Sec. 2.3, Eq. 10) corrects for this confounding. Existing kernel conditional independence tests (e.g., testing $X \perp Y | A$) assume access to fully observed joint samples. This requirement is not fulfilled in our causal setting because we only observe $Y(a)$ for units where $A = a$, meaning standard tests cannot directly account for the missing counterfactuals (i.e. $Y(1-a)$ for those units), without yielding biased results.
>
> > **"It would be nice to have an application scenario... it is slightly unclear when we should use the test."**
>
> The test is useful when the scientific question is not just "does the treatment have an effect on average?" but "does the treatment affect the distribution of outcomes in any covariate-dependent way?" This captures variance changes, tail shifts, or multimodal effects, and has real-world utility. For example, in Sec. 4.2, we study the effect of 401(k) eligibility on household wealth. The test rejects the global null, and the witness function confidence bands (Fig. 3) reveal *how* the distributional effect varies across household profiles.
>
> Another application which is safety-critical, and where our approach would have natural utility is drug safety under heterogeneous risk. Consider a drug that has the same average efficacy across all patients, but increases the variance of outcomes for specific subgroups defined by their covariates $X$---meaning, most patients respond normally, but a few experience severe adverse events. A test based on the conditional average treatment effect would fail to detect this effect. In contrast, under the conditions described in our paper, our proposed test has the power to detect that $P_{Y(1)|X} \neq P_{Y(0)|X}$ because it can capture these changes in the tails of the distribution.
>
> More broadly, the test can serve as a prerequisite before applying methods that assume homogeneous effects, or as a diagnostic for whether subgroup-specific treatment rules are warranted (e.g., in precision medicine or targeted policy). We will write these motivation in the final version (Sec. 1 and 5).
>
> > **"Some discussion on using the Wald type formulation in other statistical independence test would be helpful"**
>
> The Wald-type formulation (Section 3.2) uses a regularized inverse covariance operator $\\Omega\_P = [(1-\\varepsilon)\\Sigma\_P + \\varepsilon I]^{-1}$. This is analogous to the kernelized Hotelling $T^2$ statistic for the two-sample mean embedding test of [Harchaoui et al., (2008)](https://arxiv.org/abs/0804.1026), extended non-trivially to the cross-fitted one-step estimator in the conditional causal setting. We believe this Wald-type formulation is applicable to other kernel-based tests---including the original kernel conditional independence (KCI) test---whenever a $\\sqrt{n}$-consistent estimator of the relevant embedding is available, with an estimable covariance. We will add a discussion of this broader applicability of the Wald-type to the final version of the paper (in Sec. 5).
>
> Again, thank you for these questions. If we have addressed your concerns adequately, we hope you will consider increasing your scores. We are happy to answer additional questions during the discussion period!

---

> > ### Author Rebuttal · Reviewer_meTM · 2026-04-02
> >
> > I agree that $P_{Y^{(0)},X} \\neq P_{Y,X|A=0}$, but what I was thinking was to construct estimation of $P_{Y^{(0)}|X} = P_{Y|X,A=0}$ from splitted data $D_0$ and compare it with $P_{Y^{(1)}|X} = P_{Y|X,A=1}$ estimated from $D_1$. I do agree that this does not fit in the existing kernel testing scheme, and I would like to see the comparison to this strategy.
> >
> > For application, I'm not conviced that the test only considers the "covariate-dependent" effect. For example, if we have completely homogeneous treatment effect (i.e. $Y^{(1)} = Y^{(0)}+c$), the test would still reject null since $P(Y^{(0)}|X) \neq P(Y^{(1)}|X)$ for all $X$. Therefore, I'm not sure whether this test is good for testing heterogeneousity causal effect.

---

> > > ### Author Response · Authors · 2026-04-08
> > >
> > > Links live for 1mo: [Old Rebuttal](https://limewire.com/d/HWKWi#TpTPWUKqeC) -- [Figs](https://limewire.com/d/Pe5Q4#Vo5MH8M3uq)   |   [Updated Rebuttal](https://limewire.com/d/n6kTc#HK0Z9CETF5) -- [Figs](https://limewire.com/d/X6qwt#I4rPjDaOiX).
> > >
> > > >**...construct estimation of $P_{Y^{(0)}|X} = P_{Y|X,A=0}$ from splitted data $D_0$ and compare it with $P_{Y^{(1)}|X} = P_{Y|X,A=1}$ estimated from $D_1$...comparison to this strategy.**
> > >
> > > Thank you for clarifying the question. We agree that your suggested strategy is a sound approach for this problem. In fact, this strategy is exactly [Park et al. (2021)](https://arxiv.org/abs/2102.08208)'s approach, and we already compare to this (it is the KCD baseline) in the submitted paper. We make this explicit below.
> > >
> > > As you suggested, under standard causal assumptions, $P_{Y{(a)}|X} = P_{Y|X,A=a}$, and so $P_{Y|X,A=0}$ and $P_{Y|X,A=1}$ can be compared. But $P_{Y|X, A=a}$ is not a distribution, it is the **collection** of conditional distributions $P_{Y|X=x, A=a}$ indexed by $x\in\mathcal{X}$. [Park et al.](https://arxiv.org/abs/2102.08208) compare the treatment and control conditional distributions pointwise (at $x$) using the MMD between their conditional mean embeddings---in our notation, this is $\mathrm{CoDiTE}(x)$. Then, to test $P_{Y|X=x, A=1} = P_{Y|X=x, A=0}$ $P_X$-a.e. they form the KCD test statistic as $\mathbb E_P[\mathrm{CoDiTE}^2(X)]$. Thus, their test is precisely a comparison of $P_{Y|X,A=0}$ and $P_{Y|X,A=1}$.
> > >
> > > In comparison to this suggested strategy, our approach forms the smoothed SCoDiTE parameter ($:=\psi_P$) and tests $\psi_P = 0$. Our estimator is doubly robust, and our test based on it has provably valid type 1 error control and consistency against fixed alternatives. The suggested approach ([Park et al.](https://arxiv.org/abs/2102.08208)'s KCD) does not come with these guarantees for the same test. Moreover, in the submitted paper's Figs. 2 and 5, we compare our test variants against the KCD baseline. Our results are closer to nominal type I error under propensity misspecification, are more stable under outcome misspecification, and achieve higher power as $n$ grows, especially under outcome misspecification. Further, KCD refits the outcome models in each permutation, whereas our SKCD implementation fits the nuisances once and then bootstraps by reweighting, reducing (especially when $B<n$) the worst-case computational cost from $O(Mn^3)$ to $O(n^3 + Bn^2)$.
> > >
> > > >**...not conviced that the test only considers the "covariate-dependent" effect. For example, if we have completely homogeneous treatment effect (i.e. $Y^{(1)} = Y^{(0)}+c$), the test would still reject null...not sure whether this test is good for testing heterogeneousity causal effect.**
> > >
> > > Thank you for pointing out this subtlety. You are correct that our test does not consider *only* covariate-dependent effects. Our claim, stated more precisely, is "if our test rejects the null, then there exists a distributional treatment effect in *at least one subgroup*". Your example of a homogeneous conditional DTE is a special case of this, as there is a distributional treatment effect (in fact, the same effect) *for every subgroup*. To reflect this subtlety, we will change all mentions of "heterogenous" DTE(s) to "possibly heterogeneous" DTE(s) in the final version of the paper.
> > >
> > > However, we disagree with the comment on the practical (lack of) usefulness of our method for testing covariate-dependent effects. Once our test rejects the null, the provided approach to constructing uniform confidence bands for the estimated witness function helps with a finer-grained subgroup analysis. We already show this utility in Sec. 4.2 of the submitted paper, where our test rejects the null and we show two houshold profiles where the treatment effect differs.
> > >
> > > Finally, we agree that testing whether the conditional DTE genuinely varies with covariate values, rather than being constant across them, is an interesting problem---it is just beyond the scope of the current paper. Our contribution here is to test the global null of no conditional DTE at all; to our knowledge, this is the first test in this setting with the guarantees established in the paper. That said, we believe our framework provides a principled starting point for this narrower problem, since such hypotheses could be formulated through structural restrictions on the conditional witness $U_{P\mid x}$ or its smoothed counterpart $\psi_P$. We leave the development of such tests, with theoretical guarantees, to future work.

---

### Official Review · Reviewer_pV9U · 2026-03-02

**Soundness:** 3
**Presentation:** 3
**Significance:** 3
**Originality:** 4
**Overall Recommendation:** 4
**Confidence:** 4

**Summary:**

The authors propose a novel Hilbert-valued estimand to capture how full outcome distributions vary with covariates across treatment groups. To estimate this, they develop a cross-fitted, doubly robust one-step estimator and use this estimator to construct a permutation-free, kernel-based statistical test for global homogeneity of conditional potential outcome distributions.

**Compliance With Llm Reviewing Policy:**

Affirmed.

**Final Justification:**

After considering the authors’ rebuttal, I decide to maintain my score.

**Key Questions For Authors:**

1.Since SCoDiTE induces smoothing over $x$, how do kernel scale/bandwidth choices affect power and localization, especially for highly localized heterogeneity? Any data-driven selection guidance?

2.Can you provide actionable recommendations to diagnose/mitigate inflation (e.g., more folds, alternative learners, regularization choices)?

3.When is SKCD most advantageous vs KCD? A clearer characterization of alternatives (tail shifts, high-dimensional outcomes, non-pointwise differences) where SKCD should dominate would help.

**Limitations:**

Yes

**Strengths And Weaknesses:**

Strengths：

1.This method meaningfully advances rigorous global testing for conditional distributional effects.

2.The method avoids permutations and does not require refitting nuisance functions across bootstrap replicates.

Weaknesses:

1.Moving from CoDiTE to SCoDiTE smooths the witness over $x$. The paper would benefit from a clearer discussion of how this impacts localization, resolution and sensitivity to the choice of $k(x,x')$.

2.The simulations show inflation in some regimes. More concrete guidance (sample sizes, tuning, diagnostics) would improve usability.

---

> ### Author Rebuttal · Authors · 2026-03-31
>
> Thank you for your thorough review of our paper.
>
> >**... how do kernel scale/bandwidth choices affect power and localization ... Any data-driven selection guidance?**
>
> We developed a data-driven selection approach, SKCDAgg. We give more details in our response to reviewer EdwP and in [Rebuttal](https://limewire.com/d/HWKWi#TpTPWUKqeC) Sec. 2.
>
> For kernel smoothing using $k\_h(x, \cdot)$, we expect that a smaller $h$ preserves local structure but increases variance; larger $h$ borrows strength across covariate neighborhoods but can oversmooth fine-scale effects. A sweep over $h \in \{0.25, 0.5, 1, 2, 4\} \times$ the median heuristic on our MNIST-based simulation (5D $X, Y$; broad heterogeneity; Rebuttal Sec. 2.1, Figs. 2 and 3) shows that for our proposed SKCD tests, median heuristic is a reasonable default; power decreases for smaller bandwidths and increases (but not as markedly) for larger bandwidths---consistent with the expected behavior.
>
> We also conducted a new simulation study (Rebuttal Sec. 3, Fig. 4) comparing broad and localized heterogeneity (matched in RMS effect size). We also investigated the interplay of bandwidth selection and the curse of dimensionality through this study (see our response to reviewer EdwP). As expected, the localized alternative is systematically harder (all power curves lie lower for the localized alternative), because once $h$ exceeds the scale of the effect, the signal is oversmoothed.
>
> Note that despite the kernel-smoothing, the SKCD tests consistently outperform the KCD baseline in power in finite samples (e.g., at $n=1000$ with the median heuristic in both Rebuttal Sec. 3, [Fig. 5](https://limewire.com/d/sFwoX#lzgTgksU0L) and the submitted paper's Fig. 2). We discuss this in more detail in our response to your third question below.
>
> >**... simulations show inflation in some regimes ... actionable recommendations to diagnose/mitigate inflation ...**
>
> Note that inflation in our simulations is worse in the outcome-misspecified regime, which is severe by design. In practice, strategies to mitigate this include (1) use flexible ensemble methods (e.g., superlearner / stacking) that pool across both parsimonious and complex learners; and (2) cross-fitting, which we already employ. Upon your suggestion, we will discuss these recommendations explicitly in the final version of the paper (Sec. 4 and 5). We also note that our new SKCDAgg extension ([Rebuttal](https://limewire.com/d/HWKWi#TpTPWUKqeC) Section 2) removes at least bandwidth-sensitivity as a concern.
>
> We also implemented a causal inference validation method ([Parikh et al, 2022](https://arxiv.org/abs/2202.04208)) modified for distributional nulls. For the 401k study, even under likely outcome misspecification, Rebuttal [Fig. 6](https://limewire.com/d/sFwoX#lzgTgksU0L) shows we don't expect much inflation.
>
> >**When is SKCD most advantageous vs KCD?**
>
> We provide a clearer characterization here. Recall that KCD is the squared-norm of the CoDiTE discrepancy pointwise (KCD $= \mathbb{E}\_{P\_{X}}[\\|U\_{P|X}\\|^2\_{\mathcal{H}\_\mathcal{Y}}]$), yielding a degenerate U-statistic under the null, while SKCD-MMD cross-correlates discrepancies at different covariate locations (recall, $\\|\psi\_P\\|\_{\mathcal{H} }^2 = \mathbb{E}\_{P\_X}\mathbb{E}\_{P\_{X}}[k(X,X')\langle U\_{P|X}, U\_{P|X'}\rangle_{\mathcal{H}\_\mathcal{Y}}]$). When the per-point signal $\\|U\_{P|x}\\|$ is weak but in similar directions for nearby $x$ and $x'$, $\langle U\_{P|x}, U\_{P|x'}\rangle_{\mathcal{H}\_\mathcal{Y}}$ is reliably positive and the $k(x,x')$ weighting only minimally attenuates the signal, while KCD's degenerate U-statistic may be too noisy to exploit this. SKCD-MMD should therefore dominate for:
> (1) spatially 'broad' heterogeneity, where the treatment effect varies smoothly in $x$; (2) tail shifts, where only a small fraction of the outcome distribution changes but the change is similar across covariate values; (3) high-dimensional outcomes where the direction of $U\_{P|x}$ is more informative than its magnitude.
>
> More generally, however, if either the propensity scores are known or the outcome models are well-estimated, SKCD tests using an appropriate bandwidth (obtainable using SKCDAgg) will dominate the KCD test, even under (non-pointwise) localized heterogeneity. This follows directly from our theory, as SKCD is based on a statistically efficient, doubly robust estimator of the distributional effect, which compensates for the loss of signal due to smoothing, even for SKCD-MMD. For SKCD-Wald, the covariance inverse compensates for the attenuation in the signal-to-noise ratio at the population level but requires more data for covariance estimation. Our simulations validate this through the results for power in the MNIST-based study (Figs. 2 & 5 in the submitted paper) and Rebuttal Section 3, Fig. 5, where SKCD consistently outperforms KCD.
>
> We will add this discussion to Sec. 2.2 in the final version of the paper.

---

> > ### Author Rebuttal · Reviewer_pV9U · 2026-04-02
> >
> > I thank the authors for their time.  I still have some questions that haven't been resolved.
> >
> > 1.Does dynamically aggregating multiple bandwidths compromise the Type 1 error guarantees of Theorem 3.3, which strictly assumes a fixed operator $\Omega _{n} $?
> >
> > 2.Does kernel smoothing over covariates make SCoDiTE inherently blind to sharp, discontinuous treatment effects compared to pointwise methods like KCD?
> >
> > 3.Given the $O (Bn^{2} )$ bootstrap complexity, how can the method scale to massive datasets, and are there concrete plans to reducing the associated overhead?

---

> > > ### Author Response · Authors · 2026-04-08
> > >
> > > Links live for 1mo: [Old Rebuttal](https://limewire.com/d/HWKWi#TpTPWUKqeC) -- [Figs](https://limewire.com/d/Pe5Q4#Vo5MH8M3uq)   |   [Updated Rebuttal](https://limewire.com/d/n6kTc#HK0Z9CETF5) -- [Figs](https://limewire.com/d/X6qwt#I4rPjDaOiX).
> > >
> > > >**...aggregating multiple bandwidths compromise the Type 1 error guarantees...assumes a fixed operator $\Omega _{n}$?**
> > >
> > > Thank you for this careful question. We first point out that Theorem 3.3 *does not* require $\Omega_n$ to be fixed in the sense of being non-random; it already allows $\Omega_n$ to be data-dependent. So, aggregation does not compromise test guarantees simply because the operator is estimated from the data. What changes is that, upon aggregation, we are dealing with a finite collection of bandwidth-specific statistics $T_n^m = n \langle\Omega_n^m\bar\psi_n^m, \bar\psi_n^m\rangle_{\mathcal{H}^m}, m\in[M]$ instead of a single statistic.
> > >
> > > To address this, we further extended the theory (see Updated Rebuttal's Sec. 2, Theorem 2.2) so that each bandwidth $m\in[M]$ has a corresponding operator $\Omega_n^m\in\mathscr{W}$ such that $\\|\Omega_n^m-\Omega_\star^m\\|_{op}=o_p(1)$ for each $m$. The stacked asymptotic linearity argument is unchanged. Relative to the MMD case, the only additional step is that after obtaining joint weak convergence for the bandwidth-specific estimators $\bar\psi_n^m$, we use Slutsky's lemma together with the consistency of $\Omega_n^m$ to, under the null, obtain joint convergence of $(\sqrt{n}\bar\psi_n^m, \Omega\_n^m )\_{m=1}^M$, and then apply the continuous mapping theorem to $(h,\Omega)\mapsto \langle \Omega h,h\rangle$ to obtain joint weak convergence of $(T\_n^m)\_{m=1}^M$. The same bootstrap arguments as the previous Rebuttal's Theorem 2.2 then yield asymptotic type 1 error control for the standardized max-aggregated test.
> > >
> > > The only caveat is that the theorem is stated for a pre-specified *fixed and finite* grid of bandwidths. So technically, the median-centered multiplicative grid used in practice is a principled but still heuristic implementation of SKCDAgg.
> > >
> > > >**...make SCoDiTE inherently blind to sharp, discontinuous treatment effects compared to...KCD?**
> > >
> > > Thank you for making this concern more precise. To answer your question, no, kernel smoothing over covariates **does not** make SCoDiTE inherently blind to sharp, discontinuous treatment effects. As shown in Sec. 2.2 of the submitted paper, the SCoDiTE $\psi_P = \mu_{P_{Y(1), X}} - \mu_{P_{Y(0),X}}$, i.e., it identifies the difference between the kernel mean embeddings of the joint potential outcome and (pre-treatment) covariate distributions under standard causal assumptions. Since the product kernel is characteristic, $\psi_P=0$ if and only if $P_{Y(1), X} = P_{Y(0),X}$. By the submitted paper's Prop. 2.1, this is equivalent to $P_{Y{(1)}|X=x} = P_{Y{(0)}|X=x}$ $P_X$-a.e. Therefore, if the conditional potential outcome distributions differ on any set of covariate values with positive $P_X$-measure, even through a sharp discontinuity, then $\psi_P\neq 0$.
> > >
> > > In fact, under conditions on nuisance estimation and the operator $\Omega_n$, the submitted paper's Theorem 3.3 establishes that our SKCD test---based on the cross-fitted one-step estimator of the SCoDiTE---is consistent against *all* fixed alternatives. This means that, given enough data, our SKCD test will have power (tending to $1$ as $n$ grows) to detect any fixed conditional DTE, even if sharp/discontinuous. Of course, since kernel smoothing will "smudge" a sharp conditional DTE and attenuate the signal, a larger number of samples may be required compared to a test based on pointwise methods like the KCD. However, we emphasize again that SKCD is based on a statistically efficient doubly robust estimator of the distributional effect, which can compensate for this in practice.
> > >
> > > >**...how can the method scale to massive datasets...**
> > >
> > > Thank you for raising the issue of scalability. To respond to this, we implemented a faster low-rank version of our method using Nystrom approximations for both the covariate and outcome kernels, inspired by prior work in related contexts. We give the constructions, algorithm, and complexity analysis in Updated Rebuttal's Sec. 4.
> > >
> > > With Nystrom ranks $r_x=r_y = r \ll n$, the complexity is $O(n^2r + Bnr^2)$ for both the MMD and Wald variants. We implement several growth schedules for $r$. With polylogarithmically growing $r$, the algorithm is near-quadratic in $n$ in the pre-computation phase and near-linear in the bootstrap phase.
> > >
> > > In our experiments (see Updated Rebuttal's Figs. 6 and 7), the low-rank variants typically retain good type 1 error and power behavior. We observe large runtime gains for SKCD-Wald (up to 5.4x), where low-rank structure substantially reduces the matrix inversion cost. The gains are modest (up to 1.3x) for SKCD-MMD---not surprising at current sample sizes on GPUs, especially since low-rank approximation removes less overhead than it does for Wald.

---

### Official Review · Reviewer_EdwP · 2026-03-06

**Soundness:** 3
**Presentation:** 3
**Significance:** 3
**Originality:** 3
**Overall Recommendation:** 4
**Confidence:** 4

**Summary:**

This paper introduces a novel framework for analyzing Conditional Distributional Treatment Effects (CoDITE) by mapping potential outcome distributions into a Reproducing Kernel Hilbert Space (RKHS). The authors define the "Smoothed" Conditional Distributional Treatment Effect (SCoDiTE) and develop a cross-fitted, one-step estimator that achieves minimax optimality and double robustness. A primary contribution is the development of the Smoothed Kernel Conditional Discrepancy (SKCD) test, which provides a computationally efficient, permutation-free method for testing global homogeneity using the multiplier bootstrap of the efficient influence function. By deriving closed-form expressions for both MMD-based and Wald-type statistics, the authors significantly reduce the computational complexity compared to existing kernel-based methods while maintaining valid Type 1 error control. The practical utility of the method is demonstrated through its ability to localize treatment effects via uniform confidence bands, as illustrated in both high-dimensional image simulations and an analysis of 401(k) eligibility impacts on household wealth.

**Compliance With Llm Reviewing Policy:**

Affirmed.

**Final Justification:**

the rebuttal addressed my concern, but I keep my original assessment

**Key Questions For Authors:**

see weaknesses

**Limitations:**

see weaknesses

**Strengths And Weaknesses:**

**Strengths**

* **Soundness:** The theoretical foundation is rigorous. The authors establish the asymptotic normality of the SCoDiTE estimator at the root-$n$ rate and provide a formal proof of its double robustness and minimax optimality. The transition from estimation to a permutation-free testing procedure via the multiplier bootstrap is methodologically sound.
* **Presentation:** The paper is well-structured and clearly written. It successfully motivates the need for distributional analysis over mean-based effects and provides a clear mathematical path from the estimand definition to the final test statistic. The inclusion of "effect localization" through uniform confidence bands adds significant practical value.
* **Significance:** The work addresses a critical gap in causal inference by providing a scalable way to test for global homogeneity of conditional distributions. The reduction in computational complexity from $O(Mn^3)$ to $O(n^3 + Bn^2)$ makes kernel-based distributional testing feasible for modern datasets.
* **Originality:** The synthesis of semiparametric efficiency theory with Hilbert-valued parameters is highly creative. Specifically, the introduction of a "smoothed" kernel discrepancy that allows for Wald-type statistics (incorporating the covariance structure) represents a novel advancement over standard MMD-based approaches.

**Weaknesses**

* **Curse of Dimensionality in Joint Embeddings:** A significant concern is that the method relies on comparing mean embeddings of the joint distributions $(Y(1), X)$ and $(Y(0), X)$. In settings where the outcome $Y$ is low-dimensional but the covariate space $X$ is high-dimensional, the test may suffer from a substantial loss of power. This is a well-documented challenge in kernel two-sample testing, where the distance between embeddings can vanish as the dimensionality of the noise-carrying components ($X$) increases. The paper would be much stronger if this were investigated theoretically or through targeted simulations varying the $X/Y$ dimensionality ratio.
* **Sensitivity to Hyperparameters:** The robustness of the test depends on several hyperparameters, most notably the regularization parameter $\epsilon$ introduced in Section 3.2 for the Wald-type statistic. There is currently insufficient analysis regarding how different values of $\epsilon$ impact the trade-off between Type I error control and power. Furthermore, the selection of the smoothing bandwidth $h$ is a known bottleneck; a dedicated stability study or a more principled heuristic for selecting these parameters is needed to ensure the test's reliability across diverse datasets.

---

> ### Author Rebuttal · Authors · 2026-03-31
>
> Thank you for your careful review of our paper. We respond to your comments below.
>
> >**Curse of Dimensionality ... if this were investigated theoretically or through targeted simulations varying the $X/Y$ dimensionality ratio.**
>
> We agree this concern is well-documented in the two-sample testing literature ([Reddi et al., 2015](https://arxiv.org/abs/1406.2083)) and is closely connected to bandwidth selection. As our SCoDiTE is defined using kernel-smoothing over $x$, we investigated these concerns together, also addressing reviewer pV9U's request for a clearer discussion on how bandwidth choices affect power and localization. Upon your suggestion, we designed and ran a targeted simulation study, with supporting analysis ([Rebuttal](https://limewire.com/d/HWKWi#TpTPWUKqeC) Sec. 3, [Fig. 4](https://limewire.com/d/sFwoX#lzgTgksU0L)), that varies the $X/Y$ dimensionality ratio. Specifically, the outcome $Y$ is scalar, the treatment effect depends on a fixed 2D signal subspace of $X$, and we add $q \in$ {$0, 5, 10, 50, 100$} independent noise covariates, growing the $X$-dimension from 2 to 102 while $Y$ stays 1D. We study two alternatives---broad and localized heterogeneity---matched in RMS effect size.
>
> We show the population SKCD-MMD factors, in this idealized setting, as $A_q(h) \cdot B(h)$, where $A_q(h) = (1 + 2/h^2)^{-q/2}$ captures noise-dimension attenuation and $B(h)$ the signal contribution. For small $h$, this decay is exponential in $q$, in the same vein as [Reddi et al., (2015)](https://arxiv.org/abs/1406.2083)'s observation 1. The empirical results confirm this: for fixed $n=1000$, power for our methods collapses to zero by $q = 10$ for small $x$-bandwidths. However, under the median heuristic ($h^2 \propto q$) and larger scalings, the decay is milder, since $A_q$ stabilizes to a constant---but the growing bandwidth oversmooths the signal space ($B(h) \to 0$), which decreases power; this is even more pronounced for localized heterogeneity ([Fig. 5]()). Note that our problem differs structurally from pure two-sample testing though. It requires fitting conditional mean embeddings. Thus, for a low-dimensional outcome, under an appropriate $x$-bandwidth choice, and when the outcome nuisance is well-estimated, the overall degradation is gradual.
>
> To mitigate the curse of dimensionality, whose severity we saw depends on the $x$-bandwidth, we extended our theory and implementation to SKCDAgg ([Rebuttal](https://limewire.com/d/HWKWi#TpTPWUKqeC) Sec. 2, Theorems 1--2), a principled bandwidth aggregation approach in the spirit of [Schrab et al., (2023)](https://arxiv.org/abs/2110.15073). We give more details in our response to your Weakness 2 below and to reviewer pV9U. SKCDAgg still degrades with dimension---the underlying attenuation is unavoidable---but remains competitive across both alternatives and all noise dimensions without requiring knowledge of the signal scale. We will incorporate a discussion of the dimensionality--bandwidth tradeoff into the final version of Section 4 of the paper.
>
> >**Sensitivity to ... regularization parameter $\varepsilon$ [for the] Wald-type statistic ... insufficient analysis...**
>
> Please see our response to reviewer 3yPp's first question. We also provide the details in [Rebuttal](https://limewire.com/d/HWKWi#TpTPWUKqeC) Sec. 1.
>
> >**... selection of the smoothing bandwidth $h$ is a known bottleneck ...**
>
> Our SCoDiTE witness function is a $k\_h (x, \cdot)$-smoothing of the pointwise CoDiTE witness function. The bandwidth $h$ controls a tradeoff: smaller $h$ preserves local structure but increases variance; larger $h$ borrows strength across covariate neighborhoods but can oversmooth fine-scale effects.
>
> In response to the question of bandwidth selection, we developed SKCDAgg, a principled, data-driven approach. We provide all mathematical details in Rebuttal Sec. 2.2. It uses a (standardized) maximum of the SKCD statistics over a finite (fixed) bandwidth grid. We prove size control at level $\alpha$ and consistency against any fixed alternative detected by at least one bandwidth. The theory uses the asymptotic linearity and normality along with bootstrap validity from our main results. In our experiments on the 'neither specified' regime of our MNIST-based simulation study, SKCDAgg (for both MMD and Wald) achieves the best combined type 1 error and power without requiring knowledge of the heterogeneity scale (Rebuttal Sec. 2.1, Figs. 2 and 3). Empirically,  SKCDAgg also stabilizes type 1 error and power for the two interesting $\gamma$ values, $1/3$ and $1/19$. We will include SKCDAgg and this discussion in the final version of the paper.
>
> In light of the new analyses and results provided to resolve the highlighted weaknesses in the paper, we hope you will consider raising your score. We are also happy to answer additional questions/concerns you have during the discussion period!

---

> > ### Author Rebuttal · Reviewer_EdwP · 2026-04-03
> >
> > Thank you for the detailed and thoughtful response. I appreciate the clarifications and the planned revisions, which address my concerns.

---

### Official Review · Reviewer_3yPp · 2026-03-13

**Soundness:** 3
**Presentation:** 3
**Significance:** 3
**Originality:** 3
**Overall Recommendation:** 5
**Confidence:** 2

**Summary:**

This paper proposes a doubly robust, minimax optimal estimator for conditional distributional treatment effects (CDTEs), along with a permutation-free bootstrap test with provable type I error control. Closed-form MMD and Wald-type test statistics are derived, and the method is validated on both simulated and real data.

**Compliance With Llm Reviewing Policy:**

Affirmed.

**Final Justification:**

The authors addressed my questions.

**Key Questions For Authors:**

1. How sensitive are the finite-sample type I error and power to the choice of \gamma?
2. I find the assumptions of theorem 3.1 hard to interpret, espcially assumption iii, is this verified in a wide range of practical scenerios?

**Limitations:**

Yes.

**Strengths And Weaknesses:**

Strengths: In my view, the authors make a clear and well-executed theoretical contribution: a doubly robust, minimax optimal estimator for conditional DTEs with provable type I error control. The permutation-free bootstrap seem to be a very good computational improvement, and the closed-form MMD and Wald-type statistics are practically valuable. I think the experiments are well designed.
Weaknesses: The contributions are largely a natural extension of existing tools (Luedtke & Chung 2024, Park et al. 2021) to the conditional setting, if I am undertanding correctly. I agree that it is non trivial it but it doesn't make the paper a breakthrough. The data-driven selection of the regularization parameter \epsilon remains unresolved, and the strong positivity assumption is usually violated in practice, but this is more of a general causal inference limitation.

---

> ### Author Rebuttal · Authors · 2026-03-31
>
> Thank you for your review of our paper. We address your concerns below.
>
> >**... data-driven selection of the regularization parameter \varepsilon remains unresolved ... How sensitive are the finite-sample type I error and power to the choice of \gamma?**
>
> In response to this concern, we reran the SKCD-Wald tests for the original MNIST-based simulation study, varying $\gamma \in \\{1/19, 1/3, 1, 3, 19\\}$ across all four misspecification regimes. As we found in practice, this corresponded to an $\varepsilon$ ranging from approximately 0.08 to 0.85 ([Rebuttal](https://limewire.com/d/G2L4U#lYA2nm4lID) Sec. 1, [Fig. 1](https://limewire.com/d/sFwoX#lzgTgksU0L)).
>
> Note that the original SKCD-Wald test used $\gamma$. We find that the test is stable over $\gamma \in \\{1/3, 1, 3, 19\\}$: type 1 error and power vary among them only modestly. The smallest value ($\gamma = 1/19$) (least regularization) achieves the highest power at larger $n$, and shows inflated type 1 error only at small $n$, most visibly in the `outcome misspecified' regime.
>
> We also analyze this behavior spectrally and empirically by reparameterizing the regularized covariance proxy to be inverted as $M_n + \gamma \cdot \mathrm{tr}(M_n) \cdot I$, where $M_n$ is the covariance matrix proxy. It then follows that $\gamma$ controls both the condition number ($\leq 1 + 1/\gamma$) and effective dimension ($\leq 1/\gamma$) of the regularized covariance proxy (shown in Rebuttal Sec. 1). We empirically confirm that $\gamma = 1/3$ keeps both quantities small, providing a principled default. The smaller $\gamma = 1/19$ increases both; this trusts more dimensions in the covariate proxy at the cost of well-conditioning, which helps when the covariance is estimated well.
>
> We will update the submitted paper's Fig. 2 in the final version by adding the varying-$\gamma$ results to the existing plot, and will further extend Appendix H.2.3 of the paper by adding this spectral interpretation and practical guidance on $\gamma$ selection.
>
>
> >**I find the assumptions of theorem 3.1 hard to interpret, especially assumption iii, is this verified in a wide range of practical scenarios?**
>
> Conditions (i) and (ii) just mean that the nonparametric propensity score and outcome model estimators (for each data-split) are required to converge to their true values at some positive rate---this is satisfied in practice by essentially any consistent nonparametric estimator. Note that $\tau_r$ is the (per data-split) convergence rate for propensity estimation, and $\gamma_{a,r}$ is the (per data-split and treatment) convergence rate for the outcome model estimation.
>
> Condition (iii), $\tau_r + \min\\{\gamma_{0,r}, \gamma_{1,r}\\} > 1/2$, is the double robustness rate condition. It requires that, for each data-split, the *product* of the propensity and (slower of the two treatment-specific) outcome estimation *errors* vanishes faster than $n^{-1/2}$ (or equivalently, that the *sum* of their convergence *rates* is greater than $1/2$). Concretely, if both estimators converge at rate $n^{-1/4}$ or faster, condition (iii) is satisfied.
>
> This is a standard and relatively mild condition in the semiparametric causal inference literature (e.g., [Chernozhukov et al., 2024](https://arxiv.org/abs/1608.00060v7) Assumption 3.2) and is known to hold for a wide range of modern nonparametric estimators, including those based on sample splitting / cross-fitting as we employ. We will add an interpretive remark to this effect in the final version of the paper (Section 3.1).
>
> Thank you again. If we have addressed your questions and concerns adequately, we hope you will consider increasing your scores. We are happy to answer any further questions during the discussion period!

---

> > ### Author Rebuttal · Reviewer_3yPp · 2026-04-02
> >
> > Thank you for your answers!

---

### Decision · Program_Chairs · 2026-04-30

**Decision:**

Accept (regular)

**Comment:**

This paper proposes a doubly robust, minimax optimal estimator for conditional distributional treatment effects (CDTEs), along with a permutation-free bootstrap test with provable type I error control. The reviewers find that the theoretical foundation is rigorous and that the paper is well-structured and clearly written. The work addresses an important gap in causal inference by providing a scalable approach for testing global homogeneity of conditional distributions.

In particular, the combination of doubly robust and minimax optimal estimation with valid inference guarantees represents a strong and well-executed contribution. The permutation-free bootstrap offers a meaningful computational advantage, and the closed-form test statistics further enhance practical applicability. The experimental evaluation is carefully designed and supports the theoretical findings. Moreover, the authors have addressed the reviewers’ concerns effectively in the rebuttal, and the overall assessments are consistently positive.